



# The stability of present-day Antarctic grounding lines — Part A: No indication of marine ice sheet instability in the current geometry

Benoît Urruty[1]★, Emily A. Hill[2]★, Ronja Reese[2,3]★, Julius Garbe[3,4], Olivier Gagliardini[1], Gael Durand[1], Fabien Gillet-Chaulet[1], G. Hilmar Gudmundsson[2], Ricarda Winkelmann[3,4], Mondher Chekki[1], David Chandler[5], and Petra M. Langebroek[5]

[1]Univ. Grenoble Alpes, CNRS, IRD, Grenoble INP, IGE, 38000 Grenoble, France
[2]Department of Geography and Environmental Sciences, Northumbria University, Newcastle, UK
[3]Potsdam Institute for Climate Impact Research (PIK), Member of the Leibniz Association, Potsdam, Germany
[4]Institute of Physics and Astronomy, University of Potsdam, Potsdam, Germany
[5]NORCE Norwegian Research Centre, Bjerknes Centre for Climate Research, Bergen, Norway
★These authors contributed equally to this work.

**Correspondence:** Olivier Gagliardini (olivier.gagliardini@univ-grenoble-alpes.fr)

**Abstract.**

Theoretical and numerical work has firmly established that grounding lines of marine-type ice sheets can enter phases of irreversible advance and retreat driven by the marine ice sheet instability (MISI). Instances of such irreversible retreat have been found in several simulations of the past and future evolution of the Antarctic Ice Sheet. However, hitherto the stability regime

of Antarctic Ice Sheet grounding lines in their *current position* has not been assessed. Here we conduct a systematic numerical stability analysis of all the grounding lines of the Antarctic Ice Sheet to determine if they are currently undergoing irreversible retreat through MISI. To do this, we initialise three state-of-the-art ice-flow models, Úa, Elmer/Ice, and PISM, to replicate the current geometry of the Antarctic Ice Sheet, and then apply small, but numerically significant, perturbations in ocean-induced ice-shelf melt. We find that the grounding lines around Antarctica migrate slightly away from their initial position

while the perturbation is applied, and then revert to the initial state once the perturbation is removed. There is no indication of irreversible or self-sustaining retreat. This suggests that present-day grounding-line retreat is driven by external climate forcing alone. Hence, if the currently observed mass imbalance were to be removed, the grounding-line retreat would likely stop. However, under present-day climate forcing, further grounding-line retreat is expected, and our accompanying paper (Part B, Reese et al., 2022) shows that this could eventually lead to a collapse of some marine regions of West Antarctica.

## 1 Introduction

Retreat of the Antarctic grounding lines, i.e. the zones where the grounded ice sheet becomes so thin that it floats, could destabilise large marine regions of the ice sheet (Weertman, 1974; Schoof, 2007; Mengel and Levermann, 2014; Feldmann and Levermann, 2015), thereby committing several metres of global sea-level rise over the coming centuries to millennia (e.g., DeConto and Pollard, 2016; Golledge et al., 2015; Ritz et al., 2015; Cornford et al., 2015). Indeed, the potential for widespread

destabilisation and rapid ice discharge is one of the greatest uncertainties in future projections of ice sheet mass loss (Robel





et al., 2019; Pattyn and Morlighem, 2020; IPCC, 2021). Previous studies have made some suggestions that present-day retreat in regions of the West Antarctic Ice Sheet could mean that unstable retreat has begun (Joughin et al., 2014; Favier et al., 2014; Rignot et al., 2014). However, these studies do not provide compelling evidence, and to date there has not yet been a systematic numerical stability analysis to assess whether irreversible retreat of Antarctic grounding lines is already underway.

Marine ice sheet instability (MISI) is the proposed mechanism by which grounding lines are considered unstable, i.e. to undergo self-sustained, irreversible retreat. If ice flux across the grounding line increases with thickness, retreat on a retrograde (inland) sloping bed into deeper water, and thus regions of greater ice thickness, promotes a positive feedback in which retreat continues, unabated, inland. This self-sustaining mechanism was theoretically shown to determine the stability regime of grounding lines of marine, laterally uniform ice sheets. In this case, no stable steady-state grounding lines exist on a retrograde
sloping bed (Weertman, 1974; Schoof, 2007, 2012). However, in the case of laterally confined ice shelves that buttress the inland grounded ice, the MISI mechanism becomes more complex. Indeed, in the presence of buttressing ice shelves, stable steady-state grounding-line positions can exist on a retrograde bed slope (Gudmundsson et al., 2012; Pegler, 2018; Haseloff and Sergienko, 2018). Most ice shelves around Antarctica provide such buttressing, and have an important impact on inland ice dynamics (Fürst et al., 2016; Reese et al., 2018b). Crucially, this means that the stability of Antarctic grounding lines cannot
be concluded from the bed-slope direction alone, and ice sheet model simulations are required.

      The existence of MISI means that a shift in the position of the grounding line can cause it to cross a critical threshold (or 'tipping point'), beyond which the MISI mechanism drives the system towards a different steady state. The resulting retreat is considered irreversible, because once initiated, reversing the perturbation to pre-threshold conditions is not sufficient to halt or reverse the retreat. Instead, the forcing has to be taken past its initial value to recover the initial state (Rosier et al., 2021). As an
example, in their simulations of the whole Antarctic Ice Sheet, Garbe et al. (2020) find that retreat of West Antarctic grounding lines could be initiated by around $1 - 2\,°C$ of global warming above pre-industrial, while the recovery of these grounding lines to their modern positions requires temperatures that are at least $-1\,°C$ below the pre-industrial average.

      Large parts of the Antarctic Ice Sheet have been identified as susceptible to MISI due to their deep inland sloping topography (see Fig. S60 in Morlighem et al., 2020). There is some evidence for past extensive loss of sectors of the Antarctic Ice Sheet
during warm periods of the Pleistocene and Pliocene (e.g. Scherer et al., 1998; Reinardy et al., 2015; Wilson et al., 2018; Golledge et al., 2021). In the future, several numerical simulations have shown potential for substantial inland retreat under both present-day (Golledge et al., 2021) and future scenarios of climate forcing (e.g., DeConto and Pollard, 2016; Golledge et al., 2015). The Amundsen Sea Embayment (ASE) sector in West Antarctica is of particular concern as glaciers in this region have been accelerating and thinning, and the grounding lines are retreating (Rignot et al., 2019; Milillo et al., 2022). In addition,
recent work on Pine Island Glacier (Rosier et al., 2021) has shown the existence of three tipping points; two smaller tipping points and one third, larger event, that could eventually induce the collapse of the West Antarctic Ice Sheet.

      Parts of the East Antarctic Ice Sheet could also be susceptible to MISI due to their deep marine basins e.g. Robin subglacial basin upstream of the Filchner-Ronne ice shelf (basin 1 in Fig. 2), Recovery subglacial basin feeding the Filchner Ice Shelf (basin 2), Wilkes subglacial basin (basin 14 and upstream, Mengel and Levermann, 2014) and Aurora subglacial basin up-





stream of Totten Ice Shelf in East Antarctica (basin 13, see Sun et al., 2020). However, with the exception of Totten Glacier
(Rignot et al., 2019), these regions have not yet shown substantial acceleration, thinning or retreat of the grounding line.

The aim of this paper is to determine if stable grounding-line positions exist in the current geometry of the ice sheet. To do
this we perform a numerical stability analysis using three state-of-the-art ice sheet models, Elmer/Ice (Gagliardini et al., 2013),
Úa (Gudmundsson, 2020), and the Parallel Ice Sheet Model (PISM; Bueler and Brown, 2009; Winkelmann et al., 2011). We

initialise Elmer/Ice and Úa to be in steady state and closely replicate the present-day geometry of the ice sheet. While it is
not realistic to assume the Antarctic Ice Sheet is in a steady state today, it is necessary for such stability analysis to remove
the influence external forcing may have on the ice sheet. To assess whether this steady state ice sheet geometry is stable, we
can apply a small-amplitude perturbation to a control parameter (in this case sub-shelf melt) that satisfies the steady state
condition. This perturbation is designed to be large enough that it is numerically significant, but small enough that the state of

the system is not fundamentally altered. Applying a small perturbation to a steady state solution is a common mathematical
stability approach (e.g., Liao et al., 2007) and has been previously used in numerical studies to assess the stability of grounding
lines in idealised set-ups (Schoof, 2007; Vieli and Payne, 2005; Pattyn et al., 2006; Gudmundsson et al., 2012). If the steady
state is stable, the system will evolve back to its initial state after the perturbation is removed. Conversely, if the steady state is
unstable, a small perturbation will be enhanced and the system evolves away from its initial steady state. If we can find stable

steady states in the current geometry of the ice sheet, then we can conclude that the present-day location of Antarctic grounding
lines are stable and likely have not crossed an internal instability threshold. The existence of such stable steady states is also
strong indication that the *currently* observed retreat of Antarctic grounding lines is purely driven by changes in the external
drivers such as oceanic forcing.

To corroborate our analysis, we repeat these perturbation experiments using a transient ice sheet state in PISM that has been

forced using ocean and atmospheric conditions from 1850 to 2015. Although stability cannot strictly be defined for non-steady
states, we find that even with transient forcing included, the current grounding-line positions do not show self-sustained retreat.
Instead, after removing the perturbation, the grounding lines either reverse towards their initial position or the same position
obtained in a control simulation. Note that in this manuscript we refer to grounding lines as 'unstable' if they are engaged in
MISI-driven retreat, and 'stable' otherwise, even if the grounding lines are not in steady state.

While the approach outlined above answers the question of whether present-day grounding lines have or have not yet crossed
a critical threshold with respect to their position, the problem of their stability could also be approached from another angle.
Similar to the critical threshold in the position, the system's tipping points are defined through a critical value in the forcing, e.g.
the climatic conditions driving the system, beyond which grounding lines are committed to eventually cross a critical threshold
and undergo irreversible retreat. The latter question is specifically addressed in our accompanying paper (Part B, Reese et al.,

2022), where long-term model simulations are used to assess whether present-day climate forcing has the potential to eventually
lead to a collapse of major marine basins.

The paper is structured as follows: In the following Sect. 2.1 we present the common datasets and approaches used in our
model initialisation. We then present details of the ice sheet model set-ups in Sect. 2.2, and in Sect. 2.3 we present the initial





model states used in the perturbation experiments described in Sect. 2.4. The results are presented for the entire ice sheet and

individual drainage basins in Sect. 3, and discussed further in Sect. 4.

## 2   Methods

Here, we perform numerical simulations using three ice sheet models, to explore the stability of the current positions of
Antarctic grounding lines. A summary of our entire methodology is shown in Fig. 1. We first initialise all three models using
as many common aspects of our models as possible (Sect. 2.1) to create initial states that replicate the observed geometry of

the Antarctic Ice Sheet (see Sect. 2.3). Secondly, we use these initial states to assess the sensitivity of current grounding-line
positions to perturbations in ice shelf buttressing (Sect. 2.4). All three ice sheet models are initialised using slightly different
procedures, the details of which are outlined in Sect. 2.2.1 and 2.2.2.

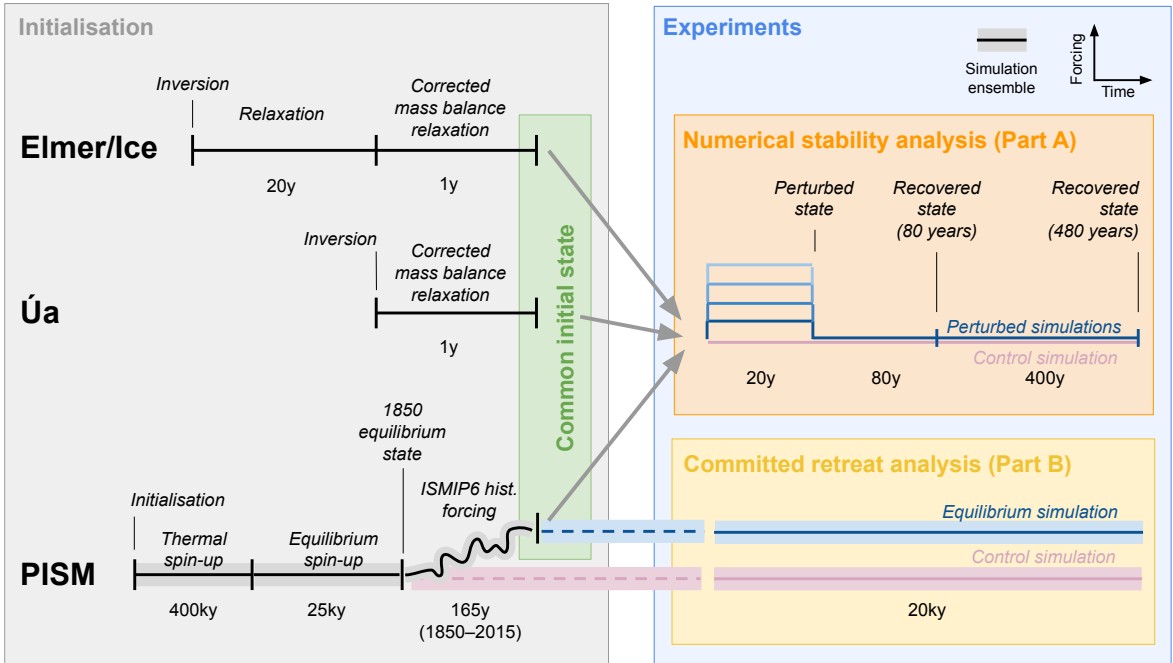

**Figure 1. Overview of experimental setup.** Schematic summarizing the experimental setup used in the stability analysis of the present-day
Antarctic grounding lines. The setup comprises the different model initialisation procedures (grey box) as well as the two different types of
prognostic model experiments (blue box): the numerical stability analysis presented in the current manuscript (Part A; orange box) and the
committed retreat analysis presented in Reese et al. (2022) (Part B; yellow box). The three different model initialisation procedures yield
three comparable initial states ("common initial state"; green box), from which all experiments are started. See text for more details.



## 2.1 Common Approach

To minimise the differences between the ice sheet initial states, we apply as many common features as possible in the model
set-ups (Table 1), either in terms of model physics, input datasets or parameterisations. In principle, any remaining differences
between the ice sheet reference states will be solely due to their individual initialisation procedures and implementation of
model physics. A comparison of all three models is given in Table 1.

PISM is a finite difference model with a regular 8-km grid, and is capable of simulating large ensembles on multi-millennial
time-scales, whereas Elmer/Ice and Úa are finite element models with unstructured grids and are designed for decadal to
century scale simulations. All models use ice density $\rho_i = 917 \ \mathrm{kg\,m^{-3}}$, sea water density $\rho_w = 1027 \ \mathrm{kg\,m^{-3}}$, and gravity
$g = 9.81 \ \mathrm{m\,s^{-2}}$. To use the most physics-based friction law all models impose a pressure-dependent friction law, which usually
leads to more physically representative sliding close to the grounding line as compared to the Weertman sliding law (Brondex
et al., 2019). The exact formulations of each law used are given in Appendices A1-A3.

To initialise the models to present day conditions, we require a number of observational datasets. We use bedrock topography,
ice surface elevation, and ice thickness from BedMachine Antarctica Version 2 (Morlighem et al., 2020). To replicate the current
ice flow, we take surface velocities from a recent snapshot in time (2015/16) from the MEaSUREs Annual Ice Velocity Maps
Version 1 dataset (Mouginot et al., 2019), which has a resolution of 1 km and good coverage across the entire ice sheet. In the
case of Úa and Elmer/Ice these surface velocities are used directly in the inversion algorithm. Across the surface of the ice sheet,
all models apply a constant-in-time surface mass balance, which is the output from the regional climate model RACMO2.3p2
averaged from 01/1995 to 12/2014 (van Wessem et al., 2018). Melting at the bottom of ice shelves is a key control on the
dynamics of the Antarctic Ice Sheet, and is the focus of the perturbation experiments. To ensure that the prescription of melt
rates does not affect the results of the perturbation experiments, all three models parameterise sub-shelf melt rates using their
respective implementations of the Potsdam Ice shelf Cavity mOdel (PICO; Reese et al., 2018a). We ensure that the PICO
geometry is the same as that in Reese et al. (2018a). PICO parameters were selected to reflect the sensitivity of sub-shelf melt
rates in the ASE sector and Filchner-Ronne Ice Shelf to ocean temperature changes. Further details on this tuning of PICO
parameters can be found in Reese et al. (2022). This results in the parameter for vertical heat exchange $\gamma_T^* = 5.5 \times 10^{-5} \ \mathrm{m\,s^{-1}}$
and the parameter for the overturning strength $C = 2 \ \mathrm{Sv\,m^3\,kg^{-1}}$. Temperature corrections were applied to individual PICO
basins summarising the far-field ocean conditions such that present-day melt rates (Adusumilli et al., 2020) are obtained for
present-day ocean forcing from Schmidtko et al. (2014), averaged over the time period 1975 to 2012..



| | Elmer/Ice | PISM | Úa |
|---|---|---|---|
| Numerical method | Finite element | Finite difference | Finite element |
| Stress Balance | SSA | SSA+SIA | SSA |
| Grid Resolution | Unstructured grid 1–50 km | 8 km | Unstructured grid 1–200 km |
| Rheology | Glen's flow law | Glen–Paterson–Budd–Lliboutry–Duval flow law | Glen's flow law |
| Friction law | Regularized Coulomb (Joughin et al., 2019) | Power-law with Mohr–Coulomb (Schoof and Hindmarsh, 2010; Bueler and van Pelt, 2015) | Regularized Coulomb (Asay-Davis et al., 2016) |
| Initialisation method | Data assimilation with relaxation | Spin-up | Data assimilation with relaxation |

**Table 1.** Comparison of the models and the physics of the models which are detailed further in Sect. 2.1 and Appendices A1-A3. SSA: Shallow-shelf approximation, SIA: Shallow-ice approximation.

## 125   2.2   Model Initialisation

### 2.2.1   Elmer/Ice and Úa

In this section, due to their similar initialisation procedures (see Fig. 1), we present the common approach used by Elmer/Ice and Úa to create initial model states for the Antarctic Ice Sheet. Specific details related to each individual model are presented in Appendices A1 and A2. These models have both been used extensively to solve ice flow problems in Antarctica, and have 130 participated in a number of model intercomparison experiments, e.g. Pattyn et al. (2012); Cornford et al. (2020). Both ice sheet codes are finite element models that solve the vertically integrated ice dynamics equations using the Shallow Shelf Approximation (SSA; MacAyeal, 1989). An unstructured finite element mesh was created for each model, where element sizes were refined in fast flowing regions. In both cases, mesh resolution is 1 km close to the grounding line. Observations of ice sheet topography were then linearly interpolated onto these meshes. Throughout the forward-in-time experiments, the extent of 135 the ice shelves remains unchanged; however, both models impose a minimum ice shelf thickness, which is sufficiently thin to represent ice that has been removed and provides no buttressing. The grounding-line position is calculated using the flotation criterion.

Here, we use a two step approach to initialise Elmer/Ice and Úa. First, we used a data assimilation approach in which we perform a model inversion to estimate the basal friction and viscosity parameters using the adjoint method (MacAyeal, 1993), 140 to replicate the observed present-day ice sheet velocities. The optimal fields for these parameters are found by minimising a cost function which is the sum of misfit and regularisation terms. The main misfit term is the difference between observed and modelled velocities. Both models also apply an additional penalty on the rates of thickness change, to reduce nonphysical ice



flux divergence anomalies. We regularise the inverse solutions using Tikhonov regularisation terms that enforce smoothness of the inferred parameters (friction and viscosity). The regularisation weights are determined using an $L$-curve analysis. By design, at the end of the inversion the surface velocities are in very good agreement with observations. See Appendices A1 and A2 for details on the inversion.

Second, given that the goal of this paper is to assess the stability of current grounding lines with respect to small-amplitude perturbations, we need to obtain an ice sheet initial state that is in steady state. We define steady state as a state for which ice volume changes through time are as close as possible to zero. While the data assimilation approach is able to replicate present day ice velocities and grounding-line positions, once we run the models forward in time, we quickly drift away from these initial conditions, due to inconsistencies in the input datasets and uncertainties in model parameters. Similarly to Price et al. (2011) and Goelzer et al. (2013), and to overcome the challenges of model drift under control conditions, we apply a correction to the mass balance term ($\dot{m}$) in the form

$$\dot{m} = \dot{b}_{\mathrm{RACMO}} - \dot{b}_{\mathrm{PICO}} - \frac{dh}{dt}\bigg|_{t_{\mathrm{relax}}}. \tag{1}$$

This seeks to bring rates of volume change as close as numerically possible to zero (steady state), by subtracting rates of ice sheet thickness change ($dh/dt$) needed to keep the model in balance. There are subtle differences as to how $\dot{m}$ is calculated in each model. Elmer/Ice begins with a 20-year relaxation period with $\dot{m} = \dot{b}_{\mathrm{RACMO}} - \dot{b}_{\mathrm{PICO}}$, where $\dot{b}_{\mathrm{RACMO}}$ denotes the 1995–2014 averaged surface mass balance provided by RACMO2.3p2 (van Wessem et al., 2018) and $\dot{b}_{\mathrm{PICO}}$ the sub-shelf melt rates provided by PICO using 1975–2012 averaged ocean forcing from Schmidtko et al. (2014) (see Appendix A1). In a second step, rates of thickness change $dh/dt$ were subtracted from the RACMO surface mass balance field and sub-shelf melt rates from PICO (Eq. 1). This $\dot{m}$ field was then put back into Elmer/Ice for a further 1-year relaxation period. Úa uses a semi-iterative approach detailed in Appendix A2. This takes $\frac{dh}{dt}\big|_{\mathrm{inv}}$ calculated after the inversion as input to Eq. (1) for a 1-year relaxation period, $t_{\mathrm{relax}}$. The $\dot{m}$ field is then recalculated using $\frac{dh}{dt}\big|_{t_{\mathrm{relax}}}$ at 1 year. In both models $\dot{m}$ is held constant for all remaining simulations. We find that applying this corrected mass balance term over the entire ice sheet (both grounded and floating portions) brings the ice sheet models to a steady state.

### 2.2.2 PISM

In contrast to Úa and Elmer/Ice, PISM simulations use a finite difference scheme to solve the momentum balance on a regular grid of 8 km horizontal resolution. Ice flow velocities are obtained from a superposition of the SSA and the shallow ice approximation (SIA) velocity fields. PISM is thermo-mechanically coupled and solves the enthalpy evolution on a three-dimensional grid. The ice rheology / rate factor is calculated from the ice enthalpy. A pseudo-plastic power-law relationship relates the basal shear stress to the SSA basal sliding velocities. Thereby, the Mohr-Coulomb criterion is used to compute the basal yield stress from parameterised till material properties (the till friction angle, a heuristic, piece-wise linear function of bedrock topography following the assumption that subglacial material with a marine history should be weaker) and the effective pressure on the saturated till (Bueler and van Pelt, 2015). The amount of water in the till is determined by the enthalpy model and a constant





till water decay rate. For consistency with the other models, glacial isostatic adjustment is not considered and we only calve floating ice that extends beyond the present-day extent of Antarctic ice shelves.

The strategy adopted to build an initial state with PISM relies on spin-up methods, with the approach taken here detailed in Reese et al. (2022) and summarised in Fig. 1: we first create a thermal equilibrium with constant geometry, followed by an ensemble of equilibrium simulations with full dynamics run for 1850 climate conditions; starting from these, historic

simulations are run from 1850 to 2015. We use ISMIP6 historic forcing for the atmosphere and the ocean from 1850 to 2015, which is a model result and not fully representative of the past climate history. Climatologies for the equilibrium state were created such that when adding the historic anomalies the atmosphere and ocean forcings between 1995 and 2014 match the present-day observations. Observed present-day velocities do not directly enter the initialisation, but instead are used, amongst other criteria, indirectly to determine optimal parameters in the initial state ensemble, which is spanned over various uncertain

parameters related to basal sliding and ice flow. From this ensemble of initial states, we select the state that best replicates present-day ice thickness, grounding-line position, mass loss, and ice surface velocities. We do so by using a scoring scheme that tests the (root mean square) deviations from observations in each individual variable by normalising and multiplying them. We lay a specific focus on the Amundsen, Weddell, and Ross seas by including the regional scores in addition to the continental values. We then use the best scoring run for the stability experiments ("ANT1" in Reese et al., 2022). Note that this state is

slightly gaining mass over the historic period. Further details on PISM are given in Appendix A3.

## 2.3   Ice sheet initial states

We have generated three ice sheet model initial states which closely replicate the current geometry, surface velocity, and grounding-line positions of the Antarctic Ice Sheet. We interpolate these initial states onto a regular 2-km grid to evaluate the performance of the ice sheet initial states with respect to observations. All indicators show a good agreement between each

of the ice sheet models and the observations (Table 2). Despite different initialisation procedures, all three models are able to closely replicate the observed ice thickness and surface velocities, particularly in the location of fast flowing ice streams (see Supplementary Figs. S1 and S2). It is not surprising that Elmer/Ice and Úa have good agreement with observations, given that they impose the surface topography from BedMachine and use observed surface velocities in the inversion. However, PISM, is also capable of locating most, if not all ice streams in their correct positions, and accurately replicating the observed ice

thickness and surface velocities after the spin-up.

To assess the sensitivity of Antarctic grounding lines to small changes in their position, we need to closely replicate the current observed position of the grounding line. Figure 2 shows that the initial grounding-line positions of all three models are in most parts in very good agreement with the observed position of the grounding line in the BedMachine dataset (Morlighem et al., 2020). We calculate the error in the initial grounding-line positions as the difference between simulated and observed

grounded ice areas, divided by the simulated grounding-line length. The resulting grounding-line position errors are 90 m for Elmer/Ice, 562 m for Úa and 12.3 km for PISM, which is below, or very close to, the minimum grid size used in each model. In addition, the good agreement of observed and modelled grounding-line positions can be seen in our individual glacier profile figures presented in the results. As expected, there is a smaller deviation between the observed and modelled grounding lines



| Indicator | Unit | Elmer/Ice | PISM | Úa | Reference | Source |
|---|---|---|---|---|---|---|
| Ice extent (total) | $10^6$ km$^2$ | 13.48 | 13.59 | 13.57 | 13.52 | Morlighem et al. (2020) |
| Ice extent (floating) | $10^6$ km$^2$ | 1.45 | 1.07 | 1.51 | 1.50 | Morlighem et al. (2020) |
| Ice extent (grounded) | $10^6$ km$^2$ | 12.03 | 12.52 | 12.05 | 12.03 | Morlighem et al. (2020) |
| Ice mass | $10^7$ Gt | 2.39 | 2.39 | 2.41 | $2.38 \pm 0.04$ | Morlighem et al. (2020) |
| Ice mass above flotation | $10^7$ Gt | 2.05 | 2.05 | 2.03 | $2.09 \pm 0.04$ | Morlighem et al. (2020) |
| Ice mass above flotation | m SLE | 56.7 | 56.47 | 56.06 | $57.9 \pm 0.9$ | Morlighem et al. (2020) |
| Ice flux across the grounding line | Gt yr$^{-1}$ | 1624 | 2124 | 1727 | $1929 \pm 40$ | Gardner et al. (2018) |
| Surface mass balance (grounded) | Gt yr$^{-1}$ | 1868 | 2203 | 1873 | 1792 | van Wessem et al. (2018) |

**Table 2.** Comparison of the initial state indicators for the initial states created using Elmer/Ice, Úa and PISM. All variables, except for flux across the grounding line, were calculated across the same 2 km resolution grid. Reference values for ice extents (total, floating, and grounded) were calculated using the BedMachine v2 dataset (Morlighem et al., 2020), whereas ice mass values were taken from Table S3 in the dataset paper (Morlighem et al., 2020) (m SLE, metres sea-level equivalent). The total flux across the grounding line was calculated in each respective model. Observed grounding-line flux was taken from Gardner et al. (2018) and total surface mass balance is from RACMO (van Wessem et al., 2018).

in Elmer/Ice and Úa, due to their inversion and relaxation initialisation procedure, in which the grounding lines have had little
opportunity to migrate away from their prescribed initial (observed) position. Due to the spin-up procedure and the coarser grid resolution of PISM there are some areas with greater deviation in the initial grounding-line position. Nonetheless there is overall good agreement, including in regions of particular interest, such as Thwaites Glacier.

By design, initial states created by Úa and Elmer/Ice are as close as possible to steady state. During a steady-state control simulation for both models there is less than 4.2 mm of change in sea-level equivalent volume over 100 years. Furthermore,
there is very little drift in the position of the grounding line from the initial location, deviating by only 12.6 (Elmer/Ice) and 2.1 m (Úa) over 100 years, which is very small with respect to grid resolution. To support the findings of this study further, we generate an initial state using PISM, that includes the recent history of the ice sheet, and therefore is not in steady state (see Sect. 2.2.2). This allows us to examine if current grounding-line positions are also reversible when transient forcing is included, and is therefore complementary to the steady-state perturbations in the other models. We account for discrepancies
between individual models by running control simulations alongside the perturbation experiments, and the results presented are then with respect to these control runs.

## 2.4 Experimental design

We take the initial ice sheet states presented in the previous section and apply a small-amplitude perturbation to assess the stability of the current grounding-line positions. This perturbation is designed to be as small as possible, i.e. we want to impose
a small deviation in the current grounding-line position such that the signal in the position of the grounding line is clearly





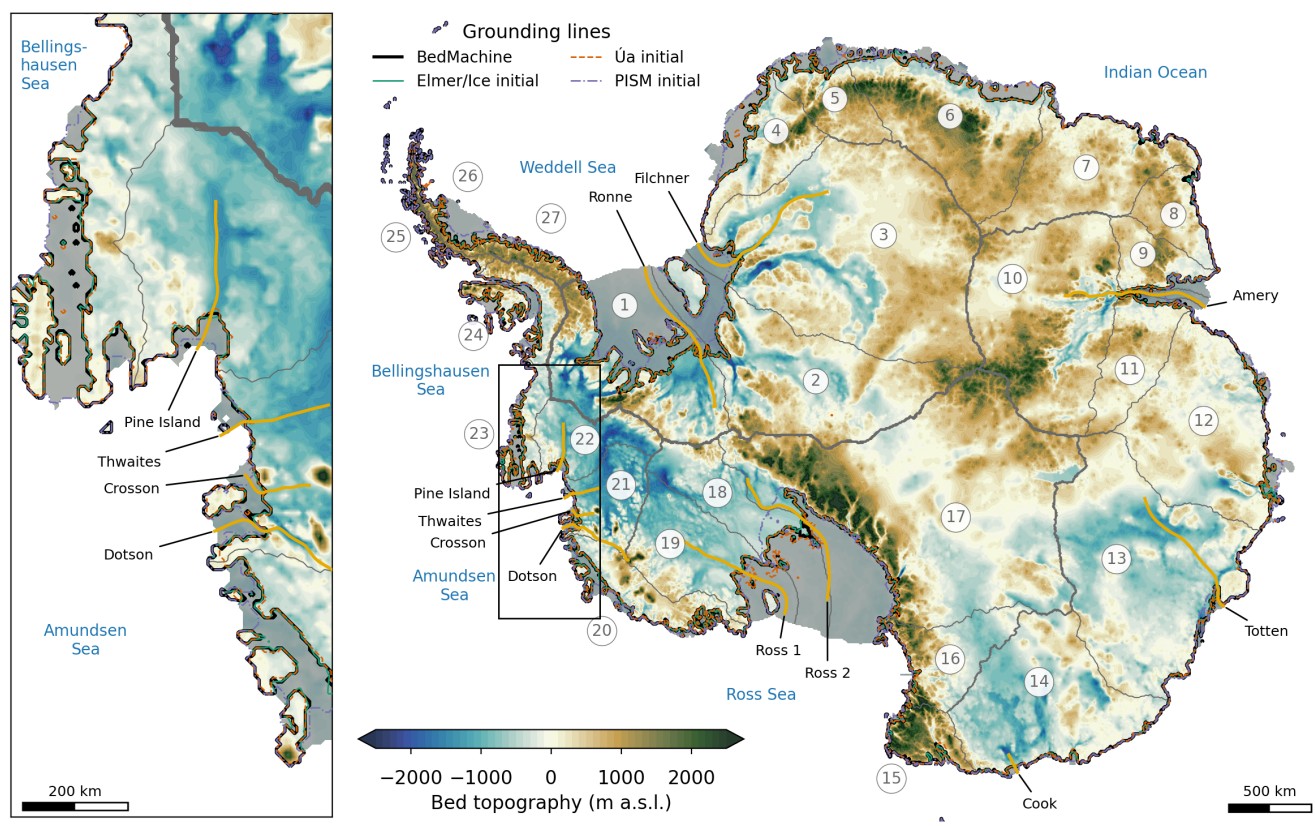

**Figure 2. Modelled and observed present-day Antarctic grounding-line locations.** Present-day Antarctic bed topography (BedMachine; Morlighem et al., 2020) showing regions where the bed topography is below (blue shading) and above (brown shading) sea level in meters above sea level (m a.s.l.). Modelled initial grounding-line positions are shown as colored lines, observed grounding-line position is shown in black. Ice shelves are indicated by grey shading. Golden lines denote the locations of the transects shown in Fig. 4. Grey lines mark the boundaries of the IMBIE basins (Zwally et al., 2012). The inset shows a zoom into the Amundsen Sea Embayment sector of West Antarctica.

visible, and distinctly emerging from the noise, but we do not want to force the grounding lines significantly into a different state. In general, it would be possible to perturb the system using a number of different control parameters in our models. Given the important role that ice shelves play on inland ice sheet dynamics and grounding-line position, via buttressing forces, we here choose to perturb the system by applying a shift in ice shelf melt rates.

We apply this perturbation in all three models by increasing the far field ocean temperature that drives the melt rates calculated by PICO. We increase the input ocean temperature, which is assumed to be representative of the conditions at depth on the continental shelf, by $[+1; +3; +5]$ °C all around the Antarctic Ice Sheet. This perturbation in temperature is applied for 20 years to create a numerically significant grounding-line retreat. By applying different increases in ocean temperature, we are able to test the robustness of this small perturbation and found that a 5 °C perturbation over 20 years remained small, i.e. it

resulted in a small but obvious deviation in the position of the grounding line. After 20 years we remove the perturbation and



allow a recovery phase of the simulation for a further 80 years (Fig. 1). We extend some simulations by 400 years to test the robustness of the grounding-line evolution over longer timescales. In parallel, we run a reference simulation (control run) with no melt perturbation.

There are subtle differences in the application of the melt perturbation between models, due to their initialisation. As Elmer/Ice and Úa already include a correction to the mass balance, we apply ice shelf melt anomalies as $\Delta \dot{b}_{\mathrm{PICO}}(t) = \dot{b}_{\mathrm{PICO}}^{\mathrm{pert}}(t) - \dot{b}_{\mathrm{PICO}}^{\mathrm{ref}}(t)$ where $\dot{b}_{\mathrm{PICO}}^{\mathrm{pert}}$ and $\dot{b}_{\mathrm{PICO}}^{\mathrm{ref}}$ are the perturbed and non-perturbed melt rates, respectively. Both $\dot{b}_{\mathrm{PICO}}^{\mathrm{pert}}$ and $\dot{b}_{\mathrm{PICO}}^{\mathrm{ref}}$ vary with time, to account for cavity geometry changes that influence the calculation of melt rates in PICO. In the control simulations and during the 480-year recovery phase, $\Delta \dot{b}_{\mathrm{PICO}} = 0$. During the 20-year perturbation phase, $\Delta \dot{b}_{\mathrm{PICO}}$ is then subtracted from the synthetic mass balance term $\dot{m}$ to give a total perturbed mass balance $\dot{m}_{pert} = \dot{m} - \Delta \dot{b}_{\mathrm{PICO}}$. In PISM, no such correction to the mass balance has been applied, instead, the initial temperatures in PICO are perturbed directly. We compare the basal mass balance perturbation applied in all three models and find that it is comparable for all experiments (see Fig. S6).

Here, we use the integrated ice flux across the grounding line as the metric of the system state, and to determine the reversibility of the grounding-line position. This is because the ice flux is inherently linked to the MISI hypothesis, where retreat down a sloping bed increases ice flux, leading to thinning, flotation, and further grounding-line retreat. Ice flux is also found to recover faster to a perturbation than grounded area or volume, due to the long timescales needed for the ice to thicken and re-advance. In the steady state initial states of Úa and Elmer/Ice, by design, the ice flux across the grounding line balances the surface accumulation upstream. An increase in ice shelf melt, and thus reduced buttressing, will lead to an increased ice flux. If the flux across the grounding line returns to its initial value after the perturbation is removed, this indicates that the ice sheet reverts to a steady state with a balance between surface accumulation in the grounded regions and grounding-line flux (note that surface accumulation is altered a little by the grounding-line movement). Hence, it is assumed that a return of the grounding-line flux indicates the grounding line has either reverted back to its initial position or has begun to re-advance towards its former position. When the grounding line does not retreat further, it means that it has found a new stable position very close of the previous one. If the flux were to increase away from its initial value, the grounding line is unstable. To support this we also examine the trend in grounded line position after the perturbation is removed, which is calculated as the change in grounded area for a constant grounding-line length.

We also analyse the recovery of the ice flux by calculating the $e$-folding relaxation time, i.e. the time taken for the flux to decrease by a factor of $e$ (Euler's number; $\approx 2.17$). To do this we fit an exponential decay function in the form

$$\Delta Q(t) = \Delta Q_{\mathrm{pert}} e^{-t/\tau} \tag{2}$$

to the change in flux during the recovery period of 80 years $\Delta Q(t)$, where $\Delta Q_{\mathrm{pert}}$ is the change in flux at the end of the 20-year perturbation relative to the initial, unperturbed flux, $t$ is time after the perturbation, and $\tau$ is the recovery timescale. We repeat this for all three models and perturbation experiments, and these exponential curves can be seen in Supplementary Figs. S3 to S5.



## 3 Results

In the following sections we present the results of our perturbation experiments of present-day Antarctic grounding lines as described in Sect. 2.4. Figure 3 shows the integrated ice flux across the entire Antarctic grounding lines in each model during the perturbation experiments. We also present the results integrated across the 27 basins from IMBIE (Fig. 2; Zwally et al., 2012), excluding basins 7, 8, and 25 which contain only small ice shelves. Additional figures showing grounding-line position change and volume above flotation can be found in the Supplement.

### 3.1 Antarctic Ice Sheet

On the Antarctic-wide scale, all models and all perturbations show a similar trend; a strong increase in ice flux, reaching a maximum at the end of the 20-year perturbation period, followed by an exponential decrease for the remaining 80 years (Fig. 3). The magnitude of the flux response to the melt perturbations is comparable between all ice sheet models, in particular for the 1 °C temperature experiment, in which ice flux increased by approximately 300 $\mathrm{Gt\,yr^{-1}}$. In the higher temperature

scenarios (3 °C and 5 °C), the flux responses diverge slightly from one another, PISM and Úa remain similar, while Elmer/Ice shows a stronger increase. This is likely due to subtle differences in the imposed basal melt perturbation in each model, which also diverge with the magnitude of the perturbation (see Fig. S6).

After the perturbation in ice shelf melt rates is removed, Antarctic-wide ice flux in all ice sheet models decreases exponentially, tending towards the initial value. The calculated $e$-folding flux response time reveals that the Antarctic wide recovery

time is in good agreement for all models, ranging between 9 and 20 years, and is largely independent of the magnitude of the perturbation (Fig. 3, bottom panel). During the 80-year recovery period, the flux decreases rapidly, but is not fully recovered. To check if it is able to recover eventually, we extended the relaxation period in the 5 °C simulations in all models to 500 years and found that the Antarctic-wide flux returns to within 3.5 % of its original value (see Supplementary Fig. S9). Alongside the ice flux evolution, the total retreat of the grounding line, and ice volume show similar trends: rapid retreat and reduction

in ice volume during the perturbation, after which retreat rates subside and grounding lines begin a slow recovery. While they do not fully recover within 80 years, due to the aforementioned slower response timescales, in the Antarctic wide signal there is no indication of accelerated retreat (see Supplementary Fig. S6). The recovery of the ice flux, alongside a short, two-decade $e$-folding time, strongly indicates that the majority of Antarctic grounding lines are stable with respect to current geometry, when perturbed by increased ice shelf melt rates. If large parts of the Antarctic grounding line were unstable, i.e. ice flux were

to increase away from the initial value, we would expect a slower or non-existent recovery for the total Antarctic ice flux. Indeed, ice flux evolution for individual basins (in all models) appears to show an exponential decrease after the perturbation is removed (Fig. 3). However, some basins recover quicker than others. In the remainder of this section we explore the response of individual basins in more detail.

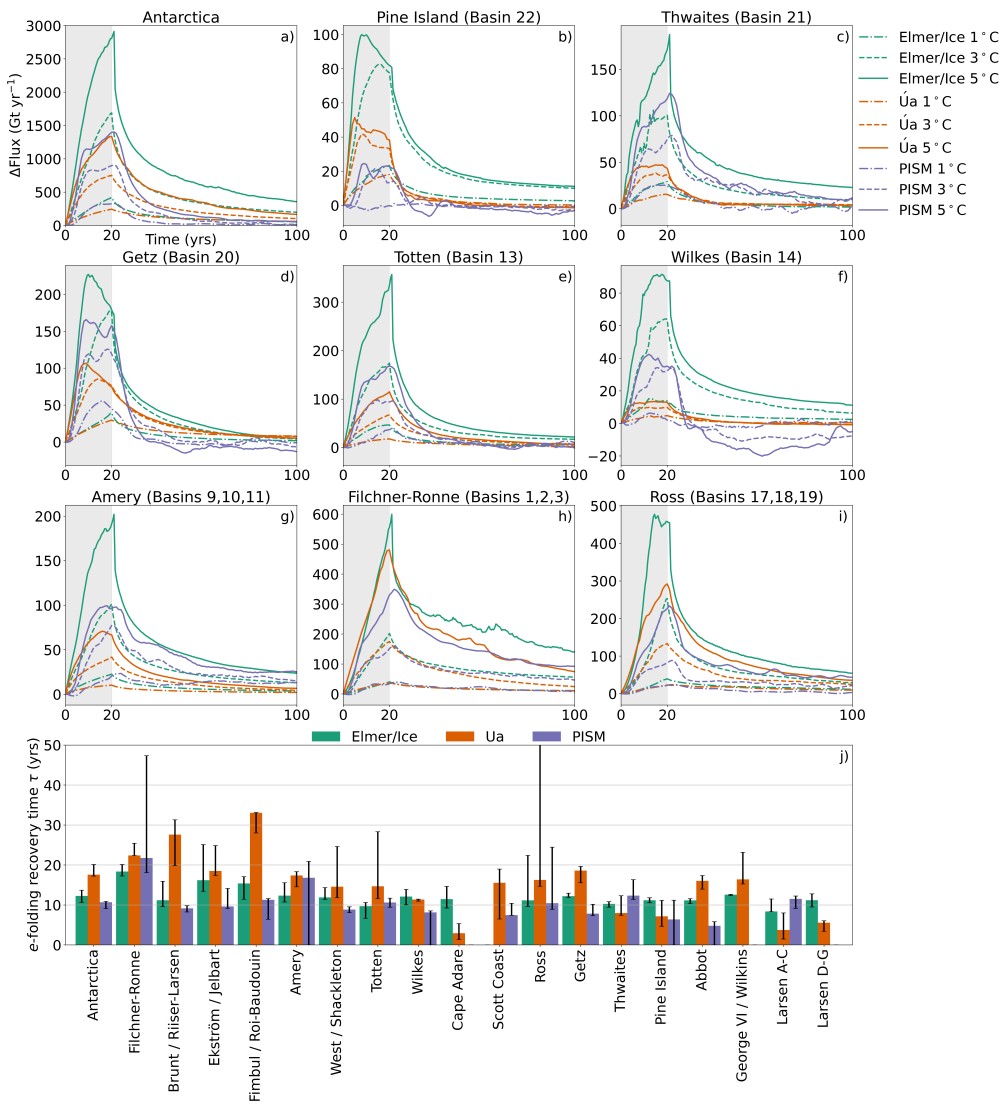

**Figure 3. Reversibility experiments and recovery time scales.** Change in integrated ice flux across the grounding line for perturbed simulations using three ice sheet models. All three temperature perturbation experiments (1 °C, 3 °C, 5 °C) are shown for selected individual basins. For ease we merge the results for basins flowing into the Amery, Filchner-Ronne and Ross ice shelves. See Supplementary Fig. S8 for additional basins. PISM fluxes are smoothed using a running mean filter of 5 years. Grey shading shows the perturbation period. Bar plot shows the $e$-folding recovery time. Each bar shows the median response time from all three experiments (1 °C, 3 °C, 5 °C) for each model, and error bars show the range. Bars are not shown for individual models for some basins (e.g. Cape Adare for PISM) where the exponential fit to the change in ice flux was deemed poor and the $R^2$ value is less than 0.8. There are four individual experiments for PISM where the flux results are noisy (due to coarser grid resolution) such that the exponential fit for that individual temperature perturbation has $R^2 < 0.8$ (see Fig. S4). We set these values to zero as the lower end of the error bars.





**Figure 4. Profiles of the Amundsen Sea Embayment (ASE) sector for the three models at different time intervals during the 5 °C experiment.** Shown are the ice-sheet geometries of the initial state (dotted line filled with light blue), at the end of the perturbation at 20 years (red line), after 80 years of recovery at 100 years (orange line), and after 500 years (cyan line). Observed (BedMachine) grounding-line positions are indicated by a black dots. The small panels show a zoom into the region marked by the black squares. The resolution of each profile depends on the model resolution. Profile locations are shown in Fig. 2.





## 3.2 Amundsen Sea Embayment sector

The Amundsen Sea Embayment (ASE) sector encompasses the Pine Island, Thwaites, and Getz drainage basins (basin numbers 20 to 22 in Fig. 2) and is of particular interest as it drains ice from some of the marine regions of the West Antarctic Ice Sheet. For all three of these basins, our results show a rapid exponential decay in the ice flux after the 20-year perturbation. This rapid recovery is also reflected in the median $e$-folding times for all three models and all three basins in the ASE sector, which are less than 20 years. In addition to the basin-wide flux responses shown in Fig. 3, we extract profiles along four marine glaciers 305 in the ASE sector (Fig. 4) to show the response of a vertical transect along the ice shelves at four critical points of the 5 °C experiment. These flow-lines were directly interpolated from the original model grids. We note that there are some differences in the bed geometry due to the different resolution and interpolation methods used by each model.

At Pine Island Glacier, the initial grounding-line position in Úa is close to observations, whereas in Elmer/Ice and PISM the initial grounding-line positions are located downstream at a topographic ridge (Fig. 4a). Despite their different starting 310 locations, Elmer/Ice and Úa grounding lines retreat (approximately 3-7 km; Fig. S7) across sections of retrograde sloping bed topography during the perturbation. The grounding line in PISM retreats along a section of pro-grade slope to the top of the ridge. After the perturbation is removed, all models show the grounding lines to advance back to their initial positions. For all three models, the 5 °C perturbation causes the ice shelf to progressively thin until it disappears entirely and then grows back to, at least, its initial thickness. Grounding-line flux (Fig. 3) shows a clear V-shape (increase to 20 years, and decrease after), 315 where Úa and PISM show the flux to have returned to either below the control simulation (PISM) or within 0.7 % (Úa) of the initial flux in just 30 years after the perturbation is removed (year 50). However, the recovery is slower in Elmer/Ice, which is reflected in the longer recovery timescale (approximately 11 years) compared with PISM and Úa and the flux remains 10 % higher than the initial at 100 years. This is also reflected in the median response times which are 6 and 7 years respectively for Úa and PISM, and 11 years for Elmer/Ice.

Our results for Thwaites Glacier are similar to Pine Island (Fig. 4b). During the perturbation the grounding lines in the entire Thwaites basin (including Dotson and Crosson ice shelves) retreat by approximately 4 to 9 km (Fig. S7) and the ice flux across the grounding line increases between 50 and 170 $\mathrm{Gt\,yr^{-1}}$. After the perturbation is removed the grounding-line fluxes are recovered in all three models and the response times are within 8-15 years (Fig. 3). At Thwaites Glacier in particular, the initial locations of the grounding lines in all three models are in close agreement with one another approximately 55 km along the 325 profile (Fig. 4b). This is slightly downstream of the observed grounding-line position at a topographic ridge. The retreat during the perturbation is not across a section of reverse bed slope, but instead remains downstream of a second topographic ridge (located 70 km along the profile). Interestingly, all models retreat to, or very close to, the observed position of the grounding line. For PISM and Úa, the ice shelf thins strongly during the perturbation, where as in Elmer/Ice, the ice shelf is relatively thick in the initial state and does not disappear during perturbation. After the end of the perturbation, the ice shelf grows back 330 to its initial shape in all models. In PISM and Úa the grounding lines re-advance fully to their initial positions (within 100 years for Úa and within 500 years for PISM). The recovery in Elmer/Ice is slower than Úa which corresponds to the longer recovery timescale for the entire Thwaites basin (Fig. 3).



Similar behaviour is also observed for Crosson and Dotson ice shelves for Elmer/Ice and Úa (Figs. 4c-d). The ice shelf thins and the grounding line retreats across a prograde slope during the perturbation. The grounding line advances after the perturbation is removed, and in Úa returns fully to its former position after 500 years. The reversing trend after 100 years indicates similar behaviour with Elmer/Ice. In contrast, the Crosson and Dotson grounding lines in PISM show some signs of instability, i.e. after the perturbation is removed, the grounding line continues to retreat and the ice surface lowers. This additional retreat is also reflected in the longer recovery time for Thwaites basin for PISM (Fig. 3), which includes the Crosson and Dotson ice shelves. In addition, we present spatial plots of the ice thickness, rates of ice thickness change and grounding-line positions for perturbation experiments in PISM in Supplementary Fig. S14. In the smaller temperature perturbations ($< 3\ °C$), thinning and retreat continue after the perturbation is removed. However, in the strongest perturbations ($4 - 5\ °C$) the signal is weaker and the grounding line appears to have reached a stable position. We suspect the grounding lines of Crosson and Dotson ice shelves in the PISM reference state are undergoing an accelerated retreat after crossing a local critical threshold. Crucially, however, the grounding lines in PISM start advanced of those of observations and the two other models. The grounding line in Crosson eventually reaches a position a few tens of kilometers upstream which is a very similar position to the observed grounding line, suggesting the observed grounding-line position is indeed stable (Fig. 4c). Long-term (15,000-year) simulations under present-day climate forcing also show that the Crosson and Dotson grounding lines will remain (in the absence of additional forcing) at this position in the future (Fig. S14).

For additional basins in this region of West Antarctica, the Getz (Fig. 3) and those in the Bellingshausen Sea sector (basins 23 and 24 in Supplementary Fig. S8), the results are similar, i.e. the ice flux tends toward its initial value, the grounding lines re-advance to their former positions (Fig. S7), and the response times are less than 20 years (Fig. 3). In summary, despite strong thinning of the buttressing ice shelves and associated retreat of the grounding lines in the ASE sector of the ice sheet during our perturbation experiments, we find that the grounding lines return to their initial positions after the perturbation is removed and the buttressing is reapplied.

## 3.3 East Antarctica

Alongside the strong signal of recovery seen in West Antarctica, a number of basins in East Antarctica show a similar response to melt rate perturbations. In particular, Wilkes, Totten and Amery basins all display a rapid exponential decay in flux after the perturbation is removed, which is reflected in their short response timescales, in most cases $< 20$ years (Fig. 3). Within 100 years, the flux in these basins appears to be reversible in Úa and PISM, whereas in Elmer/Ice the flux is reversible within 500 years (Fig. S11). Alongside this, these basins show a strong signal of re-advance of the grounding line after the perturbation is removed (Fig. S7). This is also evident in selected glacier profiles at Totten and Cook Glaciers (Fig. S13b-c). At both glaciers, the strongest perturbation ($5\ °C$) results in the entire loss of the ice shelves. Despite this, after 100 years the ice shelves have almost recovered their former ice thickness and grounding-line positions. At Cook Glacier the grounding lines have fully returned to their initial positions after 500 years in Úa and PISM and in Elmer/Ice is tending towards the initial position. We note that PISM shows retreat and subsequent re-advance along a section of reverse bed slope at Cook Glacier (Mengel and Levermann, 2014), but this retreat takes the grounding line to the same starting location as that in Úa and Elmer/Ice. At





Totten Glacier, all models start at the observed grounding-line position, and retreat approx 3-6 km inland (Fig. S7). After the perturbation is removed Úa fully recovers by 500 years and Elmer/Ice has regrounded at the observed position (Fig. S13). The ice shelf in PISM has thickened, but the grounding line has not reverted to its initial position after 500 years. However, the

grounding line has approached the same location after 500 years in the control simulation and can therefore still be considered reversible.

At Amery ice shelf, the ice flux has almost fully recovered within 100 years, but the grounding line has not returned to its former position. However, it appears to have found a new stable grounding-line position within a short (approx. < 10 km) distance inland, and does not experience accelerated retreat thereafter (Fig. S13a). We note that a number of smaller basins

surrounding Totten, Wilkes and Amery (basins 12, 15, and 16) show a similar recovery of the ice flux within approximately < 20 years and re-advance of the grounding lines.

In the Dronning Maud Land region of East Antarctica (basins 4, 5 and 6), the signal of recovery is similar; all three ice sheet models show an increase in ice flux and retreat of the grounding line during the perturbation, and a decreasing trend in ice flux after the perturbation is removed (Supplementary Figs. S8 and S7). In general, the recovery of the grounding-line position

is slower in these basins, as reflected in their recovery timescales which are often > 15 years, in particular in Úa (Fig. 3). However, the extended relaxation period shows that the flux in the 5 °C experiment recovers to within 3 % of its initial value for all models and all three basins (4-6). At the same time, all grounding lines are slowly re-advancing to their former positions, and show no signs of accelerated retreat. We note that these basins are not considered to be vulnerable to MISI (Morlighem et al., 2020), and currently show mass gain rather than loss (Rignot et al., 2019).

## 3.4 Filchner-Ronne and Ross Ice Shelf sectors

Elsewhere, the large basins draining into the Filchner-Ronne and Ross ice shelves show more complicated behaviour. In general, we do not see any strong signs of instability, i.e. the flux decays towards its initial value after the perturbation is removed, similar to the previously discussed basins. However, for the Filchner-Ronne basin in particular, the ice flux in all models remains $20-40$ % away from its initial value after the 80-year relaxation period in the 5 °C experiment, and all

models show response times of approximately 20 years (Fig. 3). In addition, all models show some signs of further retreat after the perturbation is removed (between 20-100 years in Fig. S7), but at a reduced rate. However, we find that in our extended simulations (for 5 °C) the grounding lines either (1) tend to re-advance towards their initial positions, (2) show a re-advance of a few to several kilometers before appearing to reach a new steady position (Fig. S10), or (3) settle on a slightly retreated position a short distance inland. In no cases do we see further retreat inland by the end of the simulations after 500 years. For

both the Filchner-Ronne and Ross basins the flux decreases to within 10 % (Elmer/Ice), 4.5 % (PISM), or below (Úa) its initial value after 500 years.

We extract profiles at major outlets feeding into the large Filchner-Ronne and Ross ice shelves (see Fig. S12). For both profiles feeding the Ross Ice Shelf and the Ronne Ice Shelf, all models show signs of thinning and some additional retreat after the perturbation is removed (between 20 and 100 years). However, during the extended relaxation period (100 to 500 years)

the ice shelves have begun to thicken again and the grounding lines do not retreat further inland. In the Filchner profile we



find similar behaviour, with some additional retreat after the end of the perturbation (20 years). The grounding lines in PISM and Elmer/Ice do not retreat further after 100 years, but remain in the same (slightly inland) position. In Úa the grounding line begins to re-advance and has almost returned to its initial (and full) ice shelf extent after 500 years. Overall, there are no signs of accelerated retreat in these basins, and instead the grounding lines of the larger ice shelves may need longer to recover from

a perturbation.

## 4 Discussion

Here, using three ice sheet models, we have applied small-amplitude perturbations to the current position of the Antarctic grounding lines to assess their current stability regime. Our results show that most grounding lines, including those of major marine basins in West Antarctica, are able to recover their former positions when the perturbation in ocean-induced melt is

reversed, and therefore can be considered stable. This is clearly shown by the rapid exponential decay ($e$-folding time of $< 20$ years) of the ice flux (our metric of system state) towards its initial value after the perturbation was removed, both for the entire ice sheet and for most individual basins. Our results have also shown that individual grounding lines can return towards their former positions (Fig. 4) within 100 years, particularly at the smaller basins/ice shelves, whereas the larger basins re-advance at a slower rate. Importantly, there are no signs of accelerated retreat. This conclusion is further supported by the

PISM simulations. The spin-up procedure of PISM, by definition, when it arrives at a steady state, that state will be stable. Hence, we would not need to additionally perturb such a state to determine if it is stable or unstable. Instead, if we can find a stable steady state (given a set of model parameters used in the initialisation) that is close to the present-day grounding-line positions, that in itself supports the conclusion that present-day grounding lines are compatible with a stable steady state in their current positions. Indeed, PISM is able to find such a steady state, that closely replicates the present-day grounding lines

under present-day climate conditions, which supports the conclusion that they are stable (see Appendix section A3.2). Overall, we have shown that the present-day grounding-line positions are stable in their current geometry, in the absence of any present day external forcing. However, it is imperative that this finding is understood in the context of our steady ice sheet model states.

The current state of the Antarctic Ice Sheet is not in steady state; areas in the ASE sector in particular are thinning at a rate of several metres per year (Rignot et al., 2019). To assess the stability of the grounding lines in their current geometry,

we required a steady state model set-up. As aforementioned, this is to exclude the role of external forcing (i.e. climate/ocean warming) and allow us to evaluate whether present-day grounding lines have already crossed an internal instability threshold. To bring Elmer/Ice and Úa into steady state, a small correction to the mass balance imposed in these models has to be made. This amounts to 76 $\mathrm{Gt\,yr^{-1}}$ in Elmer/Ice and 81 $\mathrm{Gt\,yr^{-1}}$ in Úa which represents 4.2 % and 4.5 % of the total surface mass balance respectively (Table 2). Compared to the impact the perturbation has on the flux, this imbalance only accounts for

$3-5$ % in the 5 °C experiments (Fig. 3). For the smaller perturbations the ratio is larger, but lower than $15-30$ %. While correcting the mass balance in this manner allows our modelled grounding lines and geometry to remain close to observations, the compromise is an ice sheet that does not resemble the actual balance between ice flux across the grounding line and the integrated surface mass balance upstream. This imbalance in the present-day ice fluxes (i.e. the integrated accumulation over





the grounded area is lower than the integrated flux across the grounded line) clearly highlights that present-day grounding

lines are, and likely will continue, to retreat over the coming decades. Our results do not suggest the contrary, but only that present-day retreat is externally driven rather than due to an internal ice sheet instability. In this case we can conclude that the grounding line has not yet crossed a critical system state threshold. Note that to draw this conclusion about no ongoing internally-driven retreat, our methodology requires that the critical thresholds for the real ice sheet (1) are correctly represented in our initial configurations and (2) have not been altered by modifications to any of our model parameters to obtain a steady

state. We have achieved both of these by firstly creating initial states as close as possible to present-day observations, and second, only applying a small correction to the surface mass balance in order to bring Úa and Elmer/Ice into steady state.

Given the ongoing impact of changes in external climate forcing on the Antarctic Ice Sheet, it is important to address whether present-day climatic conditions are sufficient to force grounding lines to eventually cross a critical system state and embark on irreversible retreat. We address this question in our accompanying paper (Reese et al., 2022), by conducting multi-

millennia scale simulations using PISM to assess whether present-day Antarctic grounding lines are committed to be unstable under current climate forcing. Indeed, we show that when present-day external climatic drivers are present, current grounding lines can eventually be forced into regions of the bed topography in which the grounding lines then become unstable. Hence, some grounding lines are committed to become unstable unless present-day climate forcing were to be reversed. This supports previous suggestions that present-day climate conditions may be sufficient to trigger rapid grounding-line retreat in West

Antarctica in the long term (Golledge et al., 2021; Garbe et al., 2020; Joughin et al., 2014).

It is perhaps surprising that the results of this paper reveal that the ASE sector of West Antarctica has reversible grounding-line positions in response to a small deviation in their position, and are therefore stable in their current geometry. This is clearly replicated in all three ice flow models. Alongside this, Pine Island and Thwaites glaciers have fast response timescales, consistent with the results of Levermann and Feldmann (2019). This reversibility is evident even under the strongest perturbation

in which a number of the smaller ice shelves disappear entirely during the perturbation period. A number of previous studies have suggested that the retrograde bed slopes of Pine Island and Thwaites may mean they have already begun internally-driven (MISI) retreat (Rignot et al., 2014; Joughin et al., 2014; Favier et al., 2014; Joughin et al., 2010; Seroussi et al., 2017; Mouginot et al., 2014; Milillo et al., 2022). Our experiments show that, under steady state conditions, i.e., the absence of an externally driven imbalance in the ice flux (albeit an unlikely scenario in present day), the Pine Island and Thwaites Glacier grounding

lines are stable with respect to their current geometrical positions. Previous work has shown that the bed topography could cause tipping points to be crossed at Pine Island Glacier (Rosier et al., 2021). Importantly, our steady state simulations do not contradict this finding, but show that the current retreat of the grounding line at Pine Island is not (yet) driven by MISI, but is instead externally driven. This is also supported by observations that show grounding-line retreat to have recently stagnated, suggesting the current position is indeed stable (Konrad et al., 2018). When the grounding line retreats further towards the

critical regions identified in Rosier et al. (2021), internal instability is likely to dominate over external forcing.

At Thwaites Glacier, all models have shown that a small perturbation in the current grounding-line position (in the absence of external forcing) is not sufficient to initiate MISI retreat. We note that this still holds when present-day climate forcing is included (PISM experiments); the grounding line and ice shelf thickness return close to observations as shown by the steady



state simulations of the other models (see Fig. 4). Crucially, entire removal of the ice shelf in our strongest perturbation is

not sufficient to destabilise the current grounding-line position, as has been suggested (Wild et al., 2022), due to the limited buttressing it provides. While present-day observed retreat at Thwaites is unlikely to be due to MISI, if a large-amplitude change in the grounding-line position were to occur (e.g. due to future changes in ocean conditions) it is possible that the grounding line will eventually become unstable. Indeed, a number of modelling studies have shown accelerated retreat under future climate forcing is possible when the grounding line retreats further inland (Joughin et al., 2014; Seroussi et al., 2017)

although associated with large uncertainties (Nias et al., 2019; Robel et al., 2019).

Elsewhere, in George V Coast and Wilkes Land, regions with deep marine subglacial basins (see Basins 13 and 14 in Fig. 2) that are considered prone to MISI (Morlighem et al., 2020), we find no signs of unstable grounding lines in their current position. Instead, these basins show a rapid recovery of the ice flux to its initial value, and re-advance of the grounding line to its former position within 500 years (see Basins 13 and 14 in Fig. S10). Cook and Ninnis glaciers in particular, have their

current grounding-line positions seaward of the 'ice plug' region identified in Mengel and Levermann (2014). However, once the grounding lines retreat beyond this stabilising region, there is potential for rapid MISI-driven retreat into the deep Wilkes subglacial basin. A deep subglacial trough also exists at Totten Glacier, but again we find the grounding line to be stable in its current location. Hence, present day (Li et al., 2015; Konrad et al., 2018) and near future retreat (Pelle et al., 2021) of Totten Glacier's grounding line is likely to be driven by external drivers rather than MISI.

By repeating our perturbation experiments using three models we have captured structural uncertainties associated with individual ice sheet models. Despite differences in initialisation procedure, grid resolution, and in some cases the initial position of the grounding lines, no model shows any signs of unstable grounding lines (with the exception of Dotson/Crosson in PISM discussed in Sect. 3.2). However, we note several potential sources of uncertainty. Firstly, we chose to perturb the grounding lines by increasing sub-shelf melt in the PICO box model. Any parameterisation of ice shelf melting has associated uncer-

tainties, and may not capture some important physical processes in sub-shelf cavities. However, the ice shelf melt distribution is not considered to greatly influence our results for two main reasons. First, our results are consistent across a wide range of temperature perturbations, including the unrealistic scenario (+5 °C) in which several ice shelves are removed entirely. Second, different implementations of PICO in each model will lead to subtly different melt rate distributions. However, our results are consistent despite this. In addition, we could have perturbed the grounding line using different model parameters,

e.g. relating to ice viscosity or basal sliding. Indeed, we modified the basal sliding using Úa and the results were consistent with those presented here; grounding lines re-advanced after the perturbation was removed.

The goal of our experiments is to assess the stability of the observed grounding lines in their current geometry. Hence, the results are dependent not only on replicating current geometry in our ice flow models, but also upon observations of the bedrock topography. While our understanding of the subglacial topography of the Antarctic Ice Sheet has improved greatly in

the last two decades, uncertainties remain, particularly in poorly surveyed regions of the ice sheet. One approach to quantify these uncertainties would be to conduct additional perturbation experiments, with modifications to the topography. In part, we have achieved this by conducting simulations using different ice sheet models and their respective grid resolutions, where in particular the bed topography in PISM is smoother than the other models (see Fig. 4). Grid resolution itself is a source of





uncertainty in ice sheet model simulations, in particular around the grounding line (Durand et al., 2009; Pattyn et al., 2013).
Again, we have partly captured uncertainties in grid resolution by conducting experiments using different models. In addition, we repeated some experiments in Úa in which we modified the mesh resolution around the grounding line, and found that our results were not affected by a finer resolution (500 m) at the grounding line.

While our models replicate the current geometry of the ice sheet, it remains challenging to accurately represent all the processes and dynamics of the ice sheet. One particular physical process that is currently challenging to model is iceberg
calving. In Elmer/Ice and Úa the ice front is fixed and the ice shelves maintain a numerically required minimum thickness, even in the case of complete melting of the ice shelf. This is a common approach in a number of ice sheet models due to the numerical challenges of removing ice entirely from the domain. It is possible that this thin layer of ice promotes ice shelf regrowth at a quicker rate than a true re-advance of the ice front, due to the external forcing, surface and basal mass balance, that are still applied across areas that have reached the minimum ice thickness. However, we note that the experiments conducted
with PISM show similar recovery time scales, where in this case ice shelf 'cells' are converted to ocean 'cells' when the ice thickness decreases to zero. Alongside calving, ice shelf damage can have an important impact on the magnitude of grounding-line retreat due to weakening in the shear zones (Lhermitte et al., 2020). Hence, it is possible that with damage included, the retreat of the grounding line in our perturbation experiments would be larger. However, damage is not yet sufficiently well parameterised for inclusion in our models, and implementing time-dependent damage in particular, is an ongoing challenge.

## 5   Conclusions

Here we show that the grounding lines of Antarctica, including Pine Island and Thwaites glaciers, exhibit no indication that marine ice sheet instability (MISI) is already underway in the *current* geometry of the ice sheet. We initialised Úa and Elmer/Ice to be in steady state close to the present-day geometry of the ice sheet, and then performed a numerical stability analysis in which we applied a small-amplitude perturbation in ocean-induced sub-shelf melt. In all cases we found that the grounding
lines retreated slightly away from their initial position while the perturbation was applied, and then returned towards their initial position after the perturbation was removed. We repeated these experiments using an ice sheet state in PISM which included the present-day trend in mass loss and found that this does not affect the reversibility of the grounding lines in the perturbation experiments. Taken together, this indicates that the Antarctic grounding lines are not currently engaged in irreversible retreat caused by internal (MISI) dynamics. Instead, it is likely that present-day observed retreat is purely driven by external forcing.
While the experiments presented in this paper do not suggest that an internal instability threshold has already been crossed in Antarctica, crucially, these experiments do not replicate the present-day mass imbalance of the ice sheet. Given that it appears highly unlikely that present-day climatic changes can be reversed, future retreat of the grounding lines driven by changes in atmospheric/oceanic conditions could force grounding lines to reach regions of the bed topography at which the ice sheet crosses a tipping point, and retreat becomes driven by the MISI. Future work is needed on an Antarctic-wide scale to quantify
the amplitude and duration of forcing required for present-day grounding lines to transgress into an irreversible retreat that would cause grounding lines to retreat over hundreds of kilometers, thereby committing several metres of sea-level rise.



## Appendix A: Ice sheet models

### A1 Elmer/Ice

#### A1.1 Model description

Elmer/Ice (https://elmerice.elmerfem.org/) is an open-source finite-element software for ice-sheets, glaciers and ice flow modelling built on the multi-physics finite-element-model suite Elmer (http://www.elmerfem.org/blog/). Elmer/Ice is a very general and flexible tool and as such has been used for a large diversity of applications (181 publications since 2004). The main features and capabilities of Elmer/Ice have been described in Gagliardini et al. (2013) and in the associated publications (https://elmerice.elmerfem.org/publications). Here, only the main characteristics relevant for our analysis will be presented.

Regarding the physics included in Elmer/Ice, the ice flow velocity is computed solving the Shallow Shelf Approximation (SSA) assuming an isotropic rheology following Glen's flow law. The initial viscosity field is computed using the 3D ice temperature field given by Liefferinge and Pattyn (2013) and using the values given in Cuffey and Paterson (2010) for the activation energies and prefactors used to compute the temperature-dependent Glen's rate-factor $A$. This initial viscosity is then modified using inverse methods. Regarding the boundary conditions, the ice front of ice-shelves is assumed to be fix (i.e.,

the sum of calving flux and frontal melt flux equals the ice flux at the front). For the grounded part, the friction parameter field is first inferred using inverse methods and a regularized friction law is then used for the relaxation phase (see Sect. 2.2.1). For the floating part of the basal boundary, basal melt from the ocean is applied using the PICO box model (Reese et al., 2018a) and no melt is applied to partially floating mesh elements. The grounding line is determined using a flotation criterion and a sub-grid scheme is applied for the friction in partially floating elements; SEP3 in Seroussi et al. (2014). Regarding the mesh,

we use an anisotropic mesh adaptation scheme that uses the observed ice velocities and thickness. The mesh is preferentially refined along the directions of highest curvature of these two fields with an additional criterion function of the distance to the grounding line. The resulting mesh contains 545,837 nodes and 1,070,444 linear elements and the size varies from 1 km close to the grounding line to 50 km in the interior. The mesh is held constant during the transient simulations.

Details concerning the model initialisation, relaxation and the friction law used in the transient simulations are given below.

#### A1.2 Model inversion

To initialise the model, we use an inverse method to estimate model parameters that control the friction and ice viscosity by minimising the misfit between observed ($\boldsymbol{u}_{\mathrm{obs}}$) and modelled ($\boldsymbol{u}_{\mathrm{mod}}$) velocities. For the friction, we optimise $\beta$ in a linear friction law that relates the basal shear stress, $\boldsymbol{t}_b$, to the basal sliding velocity, $\boldsymbol{v}_b$:

$$\boldsymbol{t}_b = -10^\beta \boldsymbol{v}_b = -C_{eff}\boldsymbol{v}_b, \tag{A1}$$





This ensures that the effective friction coefficient $C_{eff} = 10^{\beta}$ remains positive. For the ice rheology, the vertically averaged effective viscosity used in the SSA is written as

$$\bar{\mu} = \eta^2 \bar{\mu}_{ini} I_D^{(1-n)/n}, \qquad \text{with} \quad \bar{\mu}_{ini} = 1/H \int\limits_{z_b}^{z_s} (2A)^{-1/n} dz, \tag{A2}$$

where $n$ is the Glen exponent, $I_D$ is the second invariant of the strain-rate tensor $\boldsymbol{D}$ defined as $I_D^2 = 2\boldsymbol{D} : \boldsymbol{D}$, $\bar{\mu}_{ini}$ is the initial vertically averaged ice rigidity computed using the 3D ice temperature field given by Liefferinge and Pattyn (2013) as

explained above, and, to ensure that the viscosity remains positive, we optimise the parameter $\eta$ starting from an initial guess of one (i.e. no modification from the viscosity initially predicted from the temperature field).

   We apply a standard inverse methodology in which a cost function $J(\beta, \eta)$, which is the sum of misfit ($I$) and regularisation ($R$) terms, is minimized. The gradients of $J$ with respect to $\beta$ and $\eta$ are determined in a computationally efficient way using the adjoint method. The misfit ($I$) and regularisation ($R$) terms are defined as:

$$I = \frac{1}{2} \sum_{i=1}^{N_{\text{obs}}} (\|\boldsymbol{u}_{\text{mod}}^i - \boldsymbol{u}_{\text{obs}}^i\|)^2 \tag{A3}$$

$$R = \lambda_1 \frac{1}{2} \int\limits_{\Omega} (\boldsymbol{\nabla} \cdot (\boldsymbol{u}_{\text{mod}} h) - \dot{a})^2 d\Omega + \lambda_2 \frac{1}{2} \int\limits_{\Omega} \|\boldsymbol{\nabla}\beta\|^2 d\Omega + \lambda_3 \frac{1}{2} \int\limits_{\Omega} \|\boldsymbol{\nabla}\eta\|^2 d\Omega \tag{A4}$$

   As the velocity observation grids might be incomplete or have a better spatial resolution than the finite-element mesh in the ice-sheet interior, the difference between the model and the observations in $I$ (Eq. A3) is evaluated at the $N_{\text{obs}}$ locations where

observations are present. The model velocities are interpolated using the natural finite-element interpolation functions. For the regularisation term $R$ (Eq. A4), the first term computes the misfit between the modelled flux divergence $\boldsymbol{\nabla} \cdot (\boldsymbol{u}_{\text{mod}} h)$ and the apparent point mass balance $\dot{a}$ integrated over the model domain $\Omega$. Due to uncertainties in other model parameters that are not controlled during the inversion (e.g. the bed elevation), it is not possible in general to match both the velocities and the apparent point mass balance. So this term acts more as a regularisation terms that penalizes the highest ice flux divergence

anomalies. The remaining anomalies are then dampened during a relaxation period, see next section. Here, for the apparent point mass balance we use the parametrization by DeConto and Pollard (2016) for the basal mass balance and RACMO for the surface mass balance and neglect the thickness rate of changes. The two other terms impose a smoothness constraint for the two inferred parameters $\beta$ and $\eta$.

   The regularisation weights used in this study are $\lambda_1 = 3.162 \times 10^{-5}$, $\lambda_2 = 1.259 \times 10^2$ and $\lambda_3 = 7.943 \times 10^4$. Following the

principle of the $L$-curve analysis, they have been empirically chosen from a large set of inversions, as those that give a good compromise with a low misfit and small regularisation terms.

   As the velocity data set used for the common initial state is spatially incomplete, the inversion is first performed with a mosaic that aggregates observations from 2007 to 2018 (Mouginot et al., 2019) and thus has a nearly complete spatial coverage but comes at the expense of the accuracy in areas where velocities have largely changed. The minimisation is then continued





for 100 iterations using the 2015-2016 dataset to get closer to those observations while staying close to the initially inverted

values in areas where observations are missing.

### A1.3 Relaxation

The role of the relaxation is to reduce the inconsistency between input data and inverted data when we switch from a diagnostic

to a prognostic simulation (Gillet-Chaulet et al., 2012), which results in unreasonably high surface elevation rate of change.

The model relaxation in Elmer/Ice is divided into three different steps. During the first 5 years, the relaxation is applied with

the linear sliding law used for the inversion and the inverted friction coefficient. In floating areas, the friction parameter cannot

be inverted, but is set to a fixed value of $C = 1\,\mathrm{Pa\,m^{-1}\,a}$ to allow some friction if the grounding line was to advance. Then, to

use a more realistic friction law, the regularized law from Joughin et al. (2019) (see section below) is applied for the following

15 years of relaxation. The conversion from the linear friction law to the regularized one is also described in Sect. A1.4. The

last part of the relaxation is applying the mass balance correction as explained in Sect. 2.2.1. It consists of 1 year of balanced

flux relaxation, in which the mass balance term is defined from Eq. (1) and the correction term $\frac{dh}{dt}\big|_{t_{\mathrm{relax}}}$ is defined as the

thickness rate of change from the last time step of the previous relaxation step. This step permits to reach a near-steady state.

### A1.4 Regularized Coulomb sliding law

For basal sliding, we adopt the regularized Coulomb sliding law proposed by Joughin et al. (2019):

$$\boldsymbol{t}_b = -\lambda C_{s,m} \left( \frac{\|\boldsymbol{v}_b\|}{\|\boldsymbol{v}_b\| + u_0} \right)^{1/m} \frac{\boldsymbol{v}_b}{\|\boldsymbol{v}_b\|}, \tag{A5}$$

that depends on the two parameters $C_{s,m}$ and $u_0$ and where

$$\lambda = \begin{cases} 1, & \text{for } h_{af} \geq h_T \\ \dfrac{h_{af}}{h_T}, & \text{otherwise} \end{cases} \tag{A6}$$

with $h_{af}$ the height of ice above flotation and $h_T$ a threshold height. Following Joughin et al. (2019), we adopt $m = 3$,

$u_0 = 300\,\mathrm{m\,a^{-1}}$ and $h_T = 75\,\mathrm{m}$.

This friction law (Eq. A5) exhibits two asymptotic behaviours, a Weertman regime $\boldsymbol{t}_b = -C_{s,m}/u_0^{1/m}\|\boldsymbol{v}_b\|^{1/m-1}\boldsymbol{v}_b$ for

$\|\boldsymbol{v}_b\| \ll u_0$ and a Coulomb regime $\boldsymbol{t}_b = -\lambda C_{s,m}\boldsymbol{v}_b/\|\boldsymbol{v}_b\|$ for $\|\boldsymbol{v}_b\| \gg u_0$. It does not include a direct dependency on effective

pressure $N$ which role is subsumed in the model parameters. In pressure-dependant friction laws like that used in Úa (Eq. A10),

$u_0$ depends on the effective pressure. However, assuming a perfect hydrological connection between the sub-glacial drainage

system and the ocean to compute $N$ usually restricts the Coulomb regime to a small area close to the grounding line where

the ice is close to flotation. Note that in this particular case, we have $N = \rho g h_{af} = \rho g h_T \lambda$, such that both Eqs. A5 and A10

have the same dependency to water pressure in the vicinity of the grounding line. As the friction parameter $C_{s,m}$ is determined

through an inversion, it should include the dependency to $N$; so that keeping it constant through time implicitly assumes that

$N$ does not change. Because this assumption is certainly not valid as the ice column approach flotation, the factor $\lambda$ imposes a



linear correction to the friction when $h_{af}$ drops below the threshold $h_T$, so that the friction decreases smoothly toward zero at
the grounding line.

The inversion being done assuming a linear Weertman sliding law (Eq. A1), $C_{s,m}$ is inferred from the inverted effective
friction coefficient $C_{eff}$, such that

$$C_{s,m} = C_{eff} \|\boldsymbol{v}_b\| \left( \frac{\|\boldsymbol{v}_b\| + u_0}{\|\boldsymbol{v}_b\|} \right)^{1/m} / \lambda, \tag{A7}$$

In floating areas, where the friction was not constrained by the inversion, we set a constant value of $C_{s,m} = 10$ kPa.

## A2   Úa

### A2.1   Model description

Úa is an open source finite-element ice flow model (Gudmundsson et al., 2012; Gudmundsson, 2020). The model is based on
a vertically integrated formulation of the momentum equations and can be used to simulate the flow of large ice sheets such
as the Antarctic and Greenland ice sheets, ice caps and mountain glaciers. Úa solves the ice dynamics equations using the
shallow-shelf approximation (SSA) (MacAyeal, 1989) and Glen's flow law (Glen, 1955). The location of the grounding line is
determined using the flotation condition. During forward transient experiments Úa allows for fully implicit time integration,
and the non-linear system is solved using the Newton-Raphson method. A minimum ice thickness constraint is used to ensure
that ice thicknesses remain positive.

To initialise the Antarctic-wide model we take the ice extent from BedMachine Antarctica v2 (Morlighem et al., 2020) and
within this boundary we generated a finite-element mesh with 194,193 nodes and 385,097 linear elements using the Mesh2D
Delaunay-based unstructured mesh generator (Engwirda, 2015). Element sizes were refined based on effective strain rates and
distance to the grounding line, and have a maximum size of 226 km in the very interior of the ice sheet, a mean size of 3.8 km,
a median size of 1.57 km, and a minimum size of 0.68 km. Element sizes close to the grounding line are 1 km in size. We then
linearly interpolated ice surface, thickness and bed topography from BedMachine Antarctica v2 (Morlighem et al., 2020) onto
the model mesh. The surface boundary condition is stress-free, allowing the surface to respond freely to changes in surface
velocity and surface mass balance. Surface mass balance is initially prescribed using output from RACMO (van Wessem et al.,
2018), and sub-shelf melt is parameterised using an implementation of the PICO box model (Reese et al., 2018a). Parameters
of the basal slipperiness coefficient $C$ in the sliding law and ice rate factor $A$ in the flow law are determined using an inverse
approach described in the following section.

### A2.2   Model inversion

To initialise the model we use a data assimilation approach in which we estimate unknown parameters $C$ and $A$ by minimising
the misfit between observed ($u_{obs}$) and modelled ($u_{mod}$) velocities. Observed ice velocities (Mouginot et al., 2019) were
linearly interpolated onto the model mesh. Úa uses a standard inverse methodology in which a cost function $J$, which is the
sum of a misfit ($I$) and regularisation ($R$) term, is minimized. The gradients of $J$ with respect to $A$ and $C$ are determined in





a computationally efficient way using the adjoint method and Tikhonov-type regularisation. The misfit ($I$) and regularisation ($R$) terms are defined as:

$$I = \frac{1}{2\mathcal{A}} \int (u_{\mathrm{mod}} - u_{\mathrm{obs}})^2 / \epsilon_{\mathrm{obs}}^2 \, d\mathcal{A} \tag{A8}$$

$$R = \frac{1}{2\mathcal{A}} \int \left( \gamma_s^2 (\nabla \log_{10}(p/\hat{p}))^2 + \gamma_a^2 (\nabla \log_{10}(p/\hat{p}))^2 \right) d\mathcal{A} \tag{A9}$$

where $\mathcal{A} = \int d\mathcal{A}$ is the area of the model domain, $\epsilon_{\mathrm{obs}}$ are measurement errors, and $\hat{p}$ are the prior values for model parameters ($\hat{A}$ and $\hat{C}$). We use a uniform prior $\hat{A} \approx 1.15 \times 10^{-8}$ kPa$^{-3}$ yr$^{-1}$ equivalent to an ice temperature of approx. $-10°$C using an Arrhenius temperature relation. For $\hat{C}$ we estimate the prior as $\hat{C} = u_{\mathrm{obs}}/\tau_b^m$ with and $\tau_b = 80$ kPa and $m = 3$. The value of $C$ beneath the ice shelves do not deviate from this prior value. Tikhonov regularisation parameters $\gamma_s$ and $\gamma_a$ control the slope and amplitude of the gradients in $A$ and $C$. Optimum values were determined using $L$-curve analysis and are equal

to $\gamma_s = 10000$ and $\gamma_a = 1$. The inversion was run for 10,000 iterations after which the cost function had converged.

Here we invert the model to estimate $C$ using a commonly-used sliding law (Eq. A10) that relates the basal traction $\boldsymbol{t}_b$ to the horizontal components of the bed tangential basal sliding velocity $\boldsymbol{v}_b$. This was proposed by Asay-Davis et al. (2016) (Eq. 11 in that paper) and used in a recent inter-comparison experiment (Cornford et al., 2020). In our notation it reads

$$\boldsymbol{t}_b = -\frac{\mu_k N \, \mathcal{G} \, \beta^2 \|\boldsymbol{v}_b\|}{[(\mu_k N)^m + (\mathcal{G}\,\beta^2 \|\boldsymbol{v}_b\|)^m]^{1/m}} \frac{\boldsymbol{v}_b}{\|\boldsymbol{v}_b\|} \tag{A10}$$

where $\mu_k$ is a coefficient of kinetic friction, and $\mathcal{G}$ is the grounding/floating mask, with $\mathcal{G} = 1$ where the ice is grounded and $\mathcal{G} = 0$ otherwise. Here $\beta^2$ is defined as,

$$\beta^2 = C^{-1/m} \|\boldsymbol{v}_b\|^{1/m-1}, \tag{A11}$$

where $C$ is the slipperiness coefficient and we set $m = 3$. When calculating the effective pressure, $N$, we assume a perfect hydrological connection with the ocean, i.e. $N = \mathcal{G} \rho g (h - h_f) = \mathcal{G} \rho g h_{\mathrm{af}}$ where $h_f$ the flotation ice thickness (maximum ice

thickness possible for a given ocean water column thickness, $H$, where $H = S - B$ and $S$ is the ocean surface and $B$ is the bedrock) or $h_f = H \rho_w / \rho$. This ensures that the basal drag approaches zeros as the grounding line is approached from above, i.e., $\boldsymbol{t}_b = \boldsymbol{0}$ at the grounding line.

The sliding law (Eq. A10) combines basal drag as calculated separately by the Coulomb and Weertman sliding laws through reciprocal weighting and, thus, represents a combination of those two sliding laws. Equation (A10) gives the limits,

$$\|\boldsymbol{t}_b\| = \begin{cases} -\mathcal{G}\beta^2 \|\boldsymbol{v}_b\|, & \text{for } N \to +\infty \text{ or } \|\boldsymbol{v}_b\| \to 0 \quad \text{(Weertman)} \\ -\mu_k N, & \text{for } N \to 0 \text{ or } \|\boldsymbol{v}_b\| \to +\infty \quad \text{(Coulomb)} \end{cases} \tag{A12}$$

### A2.3 Relaxation

Similar to Elmer/Ice as described above (Sect. A1.3), Úa requires a period of relaxation after the inversion to dampen ice flux anomalies. Here, we use a two-step, semi-iterative approach to apply a correction to the mass balance term in Eq. (1). First,





we take rates of thickness change that are calculated at the end of the inversion $\frac{dh}{dt}\big|_{\text{inv}}$ and use these as input to Eq. (1). Then,

we relax the model forward in time for 1 year using this initial correction to $\dot{m}$, after which we take rates of thickness change $\frac{dh}{dt}\big|_{t_{\text{relax}}}$ and use these as second correction to the mass balance. This correction term is sufficient to bring the model into a steady state, in which ice volume changes are approximately zero. This correction term then remains fixed for all of the remaining control and perturbation simulations.

## A3   PISM

We here extend on the model description of PISM. For more details on the spin-up procedure, please see Reese et al. (2022).

### A3.1   Model description

The Parallel Ice Sheet Model (PISM; https://www.pism.io; Bueler and Brown, 2009; Winkelmann et al., 2011) is an open-source ice dynamics model developed at the University of Alaska, Fairbanks, and the Potsdam Institute for Climate Impact Research. In PISM, ice rheology is calculated taking into account the ice enthalpy and following the Glen–Paterson–Budd–

Lliboutry–Duval flow law (Lliboutry and Duval, 1985), assuming a Glen exponent of $n = 3$ and a SSA flow enhancement factor of $E_{SSA} = 1$. The SIA flow enhancement factor is varied in the parameter ensemble. For the run presented in this manuscript, we use a value of $E_{SIA} = 2$.

Basal sliding is parameterized in the form of a generalized power-law formulation (Schoof and Hindmarsh, 2010),

$$\boldsymbol{t}_b = -\tau_c \frac{\boldsymbol{v}_b}{u_0^q \|\boldsymbol{v}_b\|^{1-q}}, \tag{A13}$$

where $\boldsymbol{t}_b$ is the basal shear stress, $\tau_c$ is the yield stress for basal till (see below), $\boldsymbol{v}_b$ is the SSA basal sliding velocity, and $u_0$ is a threshold velocity. The sliding-law exponent $q = 1/m$ can vary between 0 (purely-plastic Coulomb sliding) and 1 (linear relationship between basal velocity and shear stress with coefficient $\tau_c/u_0$). For the experiments presented here we adopt values of $u_0 = 100$ m/a and $q = 0.75$, respectively.

Basal shear stress in the vicinity of the grounding line is linearly interpolated on a sub-grid scale between adjacent grounded

and floating grid cells according to the height above buoyancy (Feldmann et al., 2014), which allows the grounding line to evolve freely. Note that sub-shelf melt is not interpolated across the grounding line and not applied in partially floating grid cells, as usually done in PISM. The yield stress $\tau_c$ is a function of parameterized till material properties (heuristic till friction angle $\phi$) and effective till pressure $N_{till}$ ("Mohr-Coulomb criterion"; Bueler and van Pelt, 2015):

$$\tau_c = \tan(\phi)\, N_{till}, \tag{A14}$$

where $N_{till}$ is a function of the ice overburden pressure and the modelled effective amount of water in the till layer. No connection to the ocean is assumed in the calculation of the till water content, however, the till is assumed to be fully saturated when in contact with the ocean (in grid cells with floating ice or ice-free ocean) which means that freshly grounded grid cells are usually slippery.



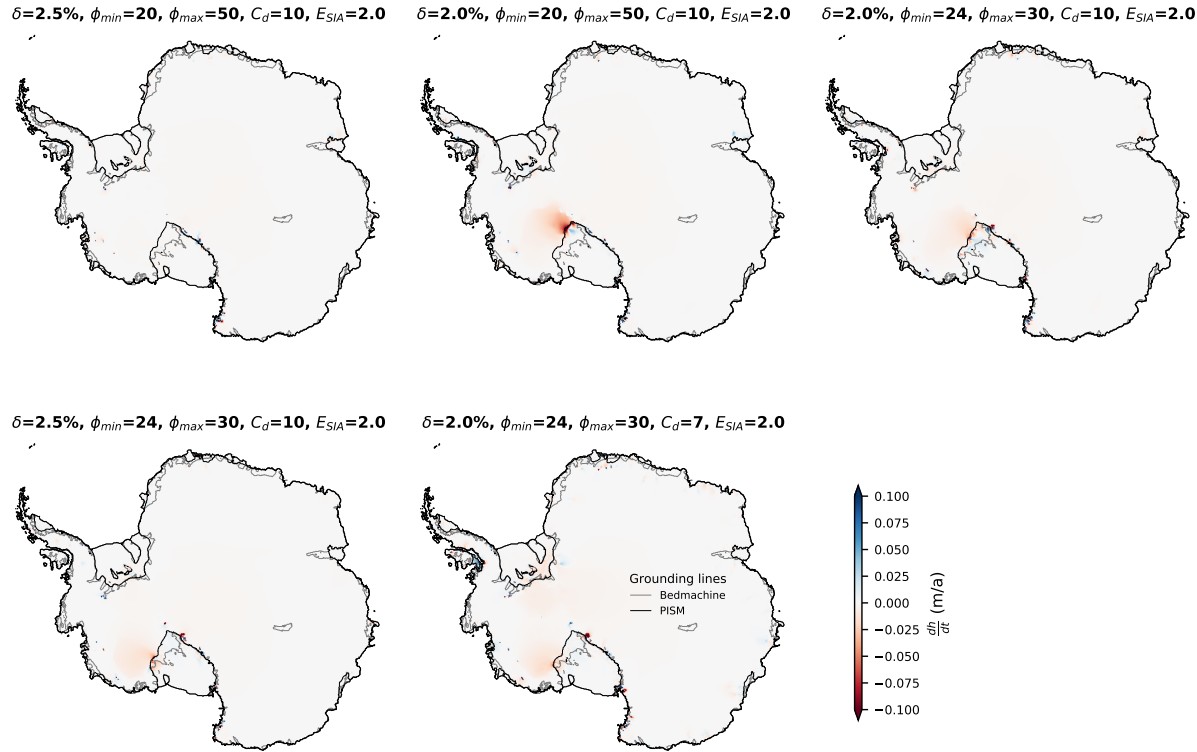

**Figure A1. Steady state simulations with PISM that are close to current grounding lines under present-day forcing.** Panels show members of the PISM present-day ensemble from Reese et al. (2022) with grounding-line positions that are close to present-day observations. The rates of ice thickness change are very low which indicates that these members are close to steady state. Note that these steady states are by construction stable.

### A3.2 Stable steady states with PISM for present-day climate conditions

Figure A1 shows steady states created with PISM under current climate conditions. Due to the spin-up approach, these states are by construction stable. Here we only show members of the long-term ensemble done with PISM in Reese et al. (2022) which remain close to present-day. This is independent evidence that indeed stable steady states can be found close to present-day grounding lines. For PISM, we do not adjust the surface or basal mass balance with a mass balance correction as done in Úa or Elmer/Ice. This means that grounding lines deviate from the exact present-day locations, for example they are upstream 720 in the Siple Coast of Ross Ice Shelf.

*Code availability.* Elmer/Ice code is publicly available through GitHub (https://github.com/ElmerCSC/elmerfem; Gagliardini et al., 2013). All the simulations were performed with version 9.0 (Rev: 242e4bb) of Elmer/Ice. PISM code is publicly available at https://github.com/



pism/pism. The Úa code is publicly available through Github (https://github.com/GHilmarG/UaSource/), and an archived version of the model can be found at https://doi.org/10.5281/zenodo.3706623 (Gudmundsson, 2020).

*Data availability.* Model output data will be made publicly available on Zenodo. DOI links to the repositories will be provided upon publication.

*Author contributions.* The experiments presented in this paper were collectively designed by members of the TiPACCs work package 2. BU, RR and EH set up and ran the simulations for Elmer/Ice, PISM and Úa, respectively, and wrote the paper. JG created Figs. 1 and 2 and provided assistance in the creation of Fig. 4. All authors contributed to the writing and discussion of ideas.

*Competing interests.* All authors declare that they have no competing interests.

*Acknowledgements.* This work is part of the TiPACCs project, which receives funding from the European Union's Horizon 2020 research and innovation programme under grant agreement no. 820575. The Elmer/Ice computations presented in this paper were performed using HPC resources of CINES under the allocation 2021-A0060106066 made by GENCI. We further acknowledge the European Regional Development Fund (ERDF), the German Federal Ministry of Education and Research (BMBF) and the Land Brandenburg for supporting this project by
providing resources on the high-performance computer system at the Potsdam Institute for Climate Impact Research. The development of PISM is supported by NSF grants PLR-1644277 and PLR-1914668 and NASA grants NNX17AG65G and 20-CRYO2020-0052. We acknowledge the use of the Northumbria University HPC facility Oswald used to perform the Úa simulations. We like to thank Christian Schoof for providing helpful input on stability analysis.



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
