# Peer review of "The stability of present-day Antarctic grounding lines — Part A: No indication of marine ice sheet instability in the current geometry"

_The Cryosphere, 2022_

## Referee Comment (RC1)

**Review of a manuscript "The stability of present-day Antarctic grounding lines — Part A: No indication of marine ice sheet instability in the current geometry" by Urruty et al.**

The manuscript describes results of a multi-model study aiming to assess sensitivity of the current Antarctic grounding line positions to short-term perturbations in the submarine melting. The authors demonstrate that simulated grounding line position reverts to its initial state after the perturbations are removed. The study represents a substantial group effort, and simulations produced many results that can be useful for other kinds of analyses. The current version of the manuscript, however, requires significant modifications, because it has several misconceptions and self-contradictions.

**General comments**

The authors conclude "This suggests that present-day grounding-line retreat is driven by external climate forcing alone. Hence, if the currently observed mass imbalance were to be removed, the grounding-line retreat would likely stop." The first part of these conclusions – the "present-day grounding-line retreat is driven by external climate forcing"– is absolutely correct. However, this is an observational evidence, rather than conclusions of this study. As the authors indicate on line 60 "... it is not realistic to assume the Antarctic Ice Sheet is in a steady state today", it is equally unrealistic to assume the Antarctic Ice Sheet has ever been or will be in a steady state. A state of an ice sheet cannot be considered in isolation from its environmental conditions – atmospheric and oceanic, at least. An ice sheet can be in a steady state only with respect to a specific set of these conditions that are constant in time. If these conditions change, the ice-sheet configuration, and its grounding line position change as well. The Earth atmospheric and oceanic conditions always vary, and they do so on a wide range of temporal scales (e.g. Jouzel, et al. (2007); Thomas et al. (2013)). Consequently, it is the variability of these conditions (that are in their turn are affected by the ice-sheet conditions) that drive variability in the ice sheets.

The abstracts starts with the sentence "Theoretical and numerical work has firmly established that grounding lines of marine-type ice sheets can enter phases of irreversible advance and retreat driven by the marine ice sheet instability (MISI).". The marine ice-sheet instability hypothesis was proposed by Weertman (1974) who was interested in "the steady-state size of a two-dimensional [unconfined] ice sheet ... that rests on a flat bed (flat before the ice sheet was placed on it) situated below sea-level". No part of the Antarctic (or Greenland) Ice Sheet has this configuration. While the authors acknowledge that the presence of the lateral confinement complicates the original Weertman's (1974) and later Schoof's (2007, 2012) conclusions that the bed slopes determine stability of the grounding line (lines 30-35), they seem to continue to use the bed slope as an indicator of stability throughout the introduction section. In addition to the ice-shelf lateral confinement, non-negligible bed topography found, for instance, under Thwaites Glacier, and very weak beds, such as under Siple Coast ice streams complicate stability conditions (Sergienko and Wingham, 2019, 2022). In the presence of feedbacks between the ice sheet characteristics (*e.g.*, the surface elevation) and environmental conditions (*e.g.*, surface accumulation) there are no general stability conditions at all (Sergienko, 2022).

Although it is true that the marine ice-sheet instability hypothesis is widely used to interpret the observed grounding line retreat, there is no need to promote this misconception, or more accurately, misapplication of a concept of stability (or instability). These concepts can be applied only to a steady state. Even though the authors point out (line 75) that "Although stability cannot strictly be defined for non-steady states" they continue "we find that even with transient forcing included, the current grounding-line positions do not show self-sustained retreat." These kinds of statements are confusing, and to some extent, self-contradictory. A possible interpretation of the finding that the grounding line positions do not show self-sustained retreat is that the grounding-line advance and retreat does not correlate with the immediate forcing it is experiencing. For instance, Robel et al. (2022) point out that in places with the appreciable bed topography, the grounding lines can persist at bed peaks under substantial changes in the environmental conditions.

It may appear that it is just a matter of semantics whether to call the present-day ice-sheet configuration "stable" or "close to stable" (either with or without quotation marks) or to have sentences like "Note that in this manuscript we refer to grounding lines as 'unstable' if they are engaged in MISI-driven retreat, and 'stable' otherwise, even if the grounding lines are not in steady state." However, the words matter (especially those in the abstract and conclusion sections), and the basic assumptions of the study should not contradict observational evidences. As currently framed, the manuscript gives a strong impression of misapprehension, despite already mentioned "bracketing sentences".

**Technical Comments**
Presumably the calving front position was held constant during all simulations. It is unlikely that during 100 years no icebergs would calve. As Haseloff and Sergienko (2022) show, the calving front conditions have stronger effects on the grounding line stability than melting.

Lines 103 and 167-168 state that PISM uses 8 km horizontal resolution (no need to repeat that twice). Seroussi et al. (2014) find that the accurate representation of the grounding line dynamics requires the horizontal resolution of 2 km or higher. I appreciate that running a spin-up for 400 kyr with such a high uniform resolution is computationally expensive, but at least some comments have to be made in that regard.

Lines 117-119 PICO is hardly a realistic representation of submarine melting. As studies that used ocean circulation models to compute melt rates show over and over again, melt rates do not correlate with the ice-shelf thickness or the ocean depth (e.g .Goldberg and Holland, 2022). Once again some justification is needed for the use of PICO (an alternative could have been melt rates inferred from observations, *e.g.* by Adusumill et al., 2020).

Lines 150-165 This section about mass-balance correction is opaque. It seems that the results whether the grounding line retreats or not depend on this correction. It also appears that the obtained 'steady-state' is contrived, *i.e.* the configuration is essentially artificially held in this state by means of this correction. If this is not the case then tthis should be illustrated. With the used approach, making any connections to the actual ice-sheet state is a stretch.

Lines 223-247 What is used as the surface mass balance during the perturbation experiment? Is the mass-balance correction applied as well?

Line 253 "An increase in ice shelf melt, and thus reduced buttressing, will lead to an increased ice flux. "This is an unsupported statement. It could potentially be verified by computing buttressing and demonstrating that it was reduced. If the calving front position is indeed fixed, then the retreat of the grounding line caused by increased melting, leads to the increase of the horizontal extent of the ice shelves, and hence increase in buttressing (Haseloff and Sergienko, 2018, 2022).

Lines 258-259 "When the grounding line does not retreat further, it means that it has found a new stable position very close of the previous one."As already mentioned, an alternative could be a non-linear response of the grounding line to the applied forcing (Robel et al., 2022).

Lines 262-265 and eqn(2) It is unclear what is meant by "the recovery of the ice flux" $\Delta Q$, and how it is computed. Presumably the $e-$ folding time could be estimated for the grounding line as well ($e.g.$, Sergienko and Wingham, 2019).

**Summary**
The manuscript requires significant modifications in terms of the presentation and of the framing of the problem as well. Also, the main conclusion that the observed grounding line migration is driven by the atmospheric and oceanic (and lithostatic) forcing is based on observations, and not an outcome of this study. Taken at its face value, the study contradicts the observational evidence that the Antarctic Ice Sheet is not in a steady state. This is hardly a good starting point if modeling studies are to be taken seriously. As mentioned above, this study is a collaborative effort of a large group of people, undoubtedly they can find less equivocal ways to use these results.

**References**

Adusumilli, S., Fricker, H. A., Medley, B., Padman, L., and Siegfried, M. R.(2020). Interannual variations in meltwater input to the southern ocean from antarctic ice shelves. *Nature Geoscience*, 13 (9), 616–620. doi:10.1038/s41561-020-0616-z

Goldberg, D. N., and Holland, P. R. (2022). The relative impacts of initialization and climate forcing in coupled ice sheet-ocean modeling: Application to Pope, Smith, and Kohler glaciers. *Journal of Geophysical Research: Earth Surface*, 127, doi:10.1029/2021JF006570

Haseloff, M., and Sergienko, O. (2022). Effects of calving and submarine melting on steady states and stability of buttressed marine ice sheets. *Journal of Glaciology*, 1-18. doi:10.1017/jog.2022.29

Jouzel, J. et al. Orbital and millennial antarctic climate variability over the past 800,000 years.Science 317, 793–796 (2007).

Robel, A., Pegler, S., Catania, G., Felikson, D., and Simkins, L. (2022). Ambiguous stability of glaciers at bed peaks. *Journal of Glaciology*, 1-8. doi:10.1017/jog.2022.31

Schoof, C (2007) Marine ice-sheet dynamics. Part 1. The case of rapid sliding. *Journal of Fluid Mechanics* 573, 27–55. doi: 10.1017/S0022112006003570

Schoof, C (2012) Marine ice sheet stability. *Journal of Fluid Mechanics* 698, 62–72. doi: 10.1017/jfm.2012.43

Sergienko O. and Wingham D. 2019. Grounding line stability in a regime of low driving and basal stresses. *Journal of Glaciology*, 65(253), 833–849 (doi: 10.1017/jog.2019.53)

Sergienko, O., and Wingham, D. (2022). Bed topography and marine ice-sheet stability. *Journal of Glaciology*, 68(267), 124-138. doi:10.1017/jog.2021.79

Sergienko, O.V. No general stability conditions for marine ice-sheet grounding lines in the presence of feedbacks. *Nat Commun* 13, 2265 (2022). https://doi.org/10.1038/s41467-022-29892-3

Seroussi, H., Morlighem, M., Larour, E., Rignot, E., and Khazendar, A.: Hydrostatic grounding line parameterization in ice sheet models, *The Cryosphere*, 8, 2075–2087, https://doi.org/10.5194/tc-8-2075-2014, 2014.

Thomas, E. R., Bracegirdle, T. J., Turner, J. and Wolff, E. W. A 308 year record of climate variability in West Antarctica. *Geophysical Research Letters* 40, 5492–5496 (2013)

Weertman, J (1974) Stability of the junction of an ice sheet and an ice shelf. *Journal of Glaciology* 13(67), 3–11. doi: 10.3189/S0022143000023327

---

## Author Response (AR1)

Comments to the author:
Dear Benoît Urruty and co-authors,

I want to truly thank you and your co-authors for the sincere and thorough answers that you forwarded to address many of the review comments.

Both reviewers acknowledge the significance of your study to the community and appreciate the substantial group effort. Yet the reviewers disagree on the level of required revisions. While reviewer #2 mostly forwarded technical comments, reviewer #1 raised more severe concerns and on two of them I want to pick up here:

1. The first major comment relates to the concept of 'stability' for ice-sheets, which is strictly only defined for 'steady states'. As I understand you, you plan to moderate your usage of the stability term and rather speak about the ice sheet showing a steady or non-steady evolution. Your effort in concise terminology is most appreciated. To me, however, the term 'unsteady' seems to invoke another notion so please confirm its usage in the literature. An alternative might be to use 'equilibrated' and 'not/non-equilibrated'.

2. A second major comment, still somewhat unresolved, relates to the surface mass balance correction. In your reply, you explain that you will add some further description/details, which is certainly beneficial. Yet you might remember that, in my initial comments I also had concerns about this correction approach. The low-resolution PISM experiments do not fully alleviate my concerns. I therefore repeat my request that you should provide additional figures showing 2D maps of this correction for all of Antarctica as well as for some specific areas of interest (presented in a supplement). These maps will then require some additional discussion in the main manuscript.

On the basis of your point-by-point answers, I invite you to submit a revised version of your manuscript but urge you to consider the two comments I raised above.

In summary, I suggest that your revised article will then enter a second review round.

Best,

Johannes Fürst

Response to the Editor:
Dear Johannes Fürst,

Thank you for inviting us to submit a revised manuscript.

Below we include a response to each of the major points you outlined. On the following pages we include the responses (in blue) to reviewers comments (in black), where **in bold** we now refer to line numbers in the revised manuscript where we have amended the text. At the end of this document we provide the track-changes manuscript.

1.  We have now indeed modified our usage of the term 'stability' throughout the manuscript. In particular we now use 'reversible' or 'self-sustained retreat'. We agree that 'unsteady' could be somewhat confusing. We have refrained from using 'unsteady' in the manuscript and instead use 'transient' to refer to the PISM state. See examples on lines 87 and 438 in the revised manuscript.

2.  We have now added two figures to the supplement (Figs. S14/S15) that show the correction terms used by Úa and Elmer/Ice to achieve a steady state (see Figure 1 below). We have also shown this correction for some specific areas of interest; the Amundsen Sea Embayment, Filchner Ice Shelf and ice streams feeding the Ross Ice Shelf. In the text we have added some additional description of these mass balance correction fields. We note that the correction terms used in Elmer/Ice and Úa are highly variable compared to one another, both in spatial pattern and amplitude. Given that we find no signs of instability in either model despite these differences, we can be confident that our results are not dependent upon this model choice.

We also want to highlight that during the review process we found that the temperature corrections applied during the PICO parameter selection were too weak (see Part B manuscript: https://tc.copernicus.org/preprints/tc-2022-105/), i.e., the present-day melt rates were underestimated. We redid the parameter selection. The parameters are only marginally affected, but the temperature corrections change. Therefore the perturbation experiments for PISM were re-run. This does not qualitatively affect our results or the main messages of the manuscript, but we have adjusted the manuscript accordingly for the results of these new simulations. We did not need to re-run the simulations for Úa and Elmer/Ice because in both models we only apply the anomaly in PICO to our background melt rates: since the sensitivity of the melt rates to ocean temperature changes depends on the input temperatures in an almost linear way in PICO, the

effect of the changes in the temperature corrections on the melt rates cancels out when using the anomaly fields in Elmer/Ice and Úa.

We thank you again for the handling of our manuscript.

Yours sincerely,

Benoît Urruty and co-authors

[Figure]

**Figure 1:** Mass balance correction terms used in Elmer/Ice and Úa to achieve a steady state. Regional maps of the mass balance correction terms (indicated by the black boxes) are provided in the supplement to the manuscript.

**Author's response: Review of the manuscript "The stability of present-day Antarctic grounding lines — Part A: No indication of marine ice sheet instability in the current geometry" by Urruty et al.**

*Dear Reviewer,*
*Thank you for reviewing our manuscript. Your comments are helpful and we are glad to respond to them. Please find our responses to your comments below and we will address all comments in a revised version of the manuscript. In order to facilitate the reading of this document, our responses are given in blue and italic compared to your comments which are given in* black without italic font.

The manuscript describes results of a multi-model study aiming to assess sensitivity of the current Antarctic grounding line positions to short-term perturbations in the submarine melting. The authors demonstrate that simulated grounding line position reverts to its initial state after the perturbations are removed. The study represents a substantial group effort, and simulations produced many results that can be useful for other kinds of analyses. The current version of the manuscript, however, requires significant modifications, because it has several misconceptions and self-contradictions.

*We are grateful for this feedback on our manuscript and we hope we will clarify the misconceptions and self-contradictions that you highlight.*

**General comments**

The authors conclude "This suggests that present-day grounding-line retreat is driven by external climate forcing alone. Hence, if the currently observed mass imbalance were to be removed, the grounding-line retreat would likely stop." The first part of these conclusions – the "present-day grounding-line retreat is driven by external climate forcing"– is absolutely correct. However, this is an observational evidence, rather than conclusions of this study. As the authors indicate on line 60 "... it is not realistic to assume the Antarctic Ice Sheet is in a steady state today", it is equally unrealistic to assume the Antarctic Ice Sheet has ever been or will be in a steady state. A state of an ice sheet cannot be considered in isolation from its environmental conditions – atmospheric and oceanic, at least. An ice sheet can be in a steady state only with respect to a specific set of these conditions that are constant in time. If these conditions change, the ice-sheet configuration, and its grounding line position change as well. The Earth atmospheric and oceanic conditions always vary, and they do so on a wide range of temporal scales (e.g. Jouzel, et al. (2007); Thomas et al. (2013)). Consequently, it is the variability of these conditions (that are in their turn are affected by the ice-sheet conditions) that drive variability in the ice sheets.

*We do not clearly understand the argument of the reviewer and presumably only partly agree. Owing to the long-term response of ice sheets (decadal to multi-millenia), it is obvious that they cannot be in a steady state in an always varying climate. We agree that, ultimately, any change in the ice sheet is triggered by variations in the external conditions. However, ice sheets can undergo hysteresis and thus respond irreversibly, driven by internal dynamics, to such changes. Observing a mass loss and a retreat of the grounding line alone cannot discriminate whether these changes are only a limited and reversible response to the forcing, or a self-sustained collapse (initially triggered by climate forcing). Modeling studies appear fundamental to make such a discrimination and our work is an attempt in that direction. **We have made changes to the manuscript in the introduction to make it clear that the real ice sheet is not in steady state today, and further justification for requiring a steady-state configuration for our modeling analysis (lines 72 to 81 in the revised manuscript).** With regards to the sentence in the abstract, we agree it could be confusing to the reader that the statement "present-day grounding line retreat is driven by external climate forcing" could be concluded from observations. Importantly, what we can conclude here from our model experiments is that the current retreat is only driven by external climate forcing, and not yet driven by MISI alongside this. **To alleviate confusion, we have removed this sentence from the abstract.***

The abstracts starts with the sentence "Theoretical and numerical work has firmly established that grounding lines of marine-type ice sheets can enter phases of irreversible advance and retreat driven by the marine ice sheet instability (MISI).". The marine ice-sheet instability hypothesis was proposed by Weertman (1974) who was interested in "the steady-state size of a two-dimensional [unconfined] ice sheet ... that rests on a flat bed (flat before the ice sheet was placed on it) situated below sea-level". No part of the Antarctic (or Greenland) Ice Sheet has this configuration. While the authors acknowledge that the presence of the lateral confinement complicates the original Weertman's (1974) and later Schoof's (2007, 2012) conclusions that the bed slopes determine stability of the grounding line (lines 30-35), they seem to continue to use the bed slope as an indicator of stability throughout the introduction section. In addition to the ice-shelf lateral confinement, non-negligible bed topography found, for instance, under Thwaites Glacier, and very weak beds, such as under Siple Coast ice streams complicate stability conditions (Sergienko and Wingham, 2019, 2022). In the presence of feedbacks between the ice sheet characteristics (e.g., the surface elevation) and environmental conditions (e.g., surface accumulation) there are no general stability conditions at all (Sergienko, 2022).

*We fully agree with the above arguments. We adjusted the manuscript carefully to avoid the confusion that retrograde bed slope is the only determinant of marine ice-sheet instability (i.e., bed slope is a sufficient and necessary condition for MISI), however recognizing that it is a necessary condition. We are grateful for the additional sentences and references that **we have now incorporated into the paragraph in the introduction where we explain the grounding line stability conditions (line 33 to 36).***

*With regards to the comment where we "continue to use bed slope as an indicator of stability throughout the introduction" we have found two instances where we think the reviewer is referring to (line numbers in the original manuscript):*

  *1)  line 43 "Large parts of the Antarctic Ice Sheet have been identified as susceptible to MISI due to their deep inland sloping topography", which is taken from figure S60 in Morlighem et al. 2020*
  *2)  lines 52-56, beginning with "Parts of the East Antarctic Ice Sheet could also be susceptible to MISI due to their deep marine basins"*

*In both cases we had used the word susceptible to suggest that these regions could be prone to MISI due to their topography, i.e., fulfilling the necessary condition. To make it completely clear that we are not suggesting that they are prone to MISI based on bed-slope alone, **we have added 'satisfying the necessary condition for MISI' on line 47 (for instance 1), and 'also fulfill the necessary condition for MISI' on line 53 (for instance 2).***

Although it is true that the marine ice-sheet instability hypothesis is widely used to interpret the observed grounding line retreat, there is no need to promote this misconception, or more accurately, misapplication of a concept of stability (or instability). These concepts can be applied only to a steady state. Even though the authors point out (line 75) that "Although stability cannot strictly be defined for non-steady states" they continue "we find that even with transient forcing included, the current grounding-line positions do not show self-sustained retreat." These kinds of statements are confusing, and to some extent, self-contradictory. A possible interpretation of the finding that the grounding line positions do not show self-sustained retreat is that the grounding-line advance and retreat does not correlate with the immediate forcing it is experiencing. For instance, Robel et al. (2022) point out that in places with the appreciable bed topography, the grounding lines can persist at bed peaks under substantial changes in the environmental conditions.

It may appear that it is just a matter of semantics whether to call the present-day ice-sheet configuration "stable" or "close to stable" (either with or without quotation marks) or to have sentences like "Note that in this manuscript we refer to grounding lines as 'unstable' if they are engaged in MISI-driven retreat, and 'stable' otherwise, even if the grounding lines are not in steady state." However, the words matter (especially those in the abstract and conclusion sections), and the basic assumptions of the study should not contradict observational evidence. As currently framed, the manuscript gives a strong impression of misapprehension, despite already mentioned "bracketing sentences".

*As mentioned by the reviewer the concept of "stable" or "unstable" can only be applied to a steady-state. In particular, we agree that there are some confusing statements in the manuscript with respect to how we refer to the results of the PISM experiments (conducted using a transient state) alongside the experiments conducted with Úa and Elmer/Ice, both of which are in steady*

state. **We have removed both of the sentences that you have quoted above to alleviate confusion.** *Furthermore, we now also make use of "unsteady" and distinguish it from "unstable" as we agree with the reviewer that there is some confusion of these words found in the literature.* **We have now made sure not to use the words unstable and stable when referring to the transient "unsteady" ice sheet state used by PISM (examples on lines 323 and 437).** *We note that we use control simulations alongside the perturbations so that the impact of the transient forcing in PISM is removed.* **We then refer to "reversible" in the sense that it arrives at the location in the control simulation (see example in the Results on line 397-98).**

*In the PISM experiments that include present-day forcing, it is possible that the grounding lines have stopped at a bed peak. In our Part B experiments, applying the present-day forcing for an extended period allows us to test whether the grounding lines will remain at these positions in the long-term (millennial timescales). Indeed we find that in a number of cases, in certain regions, they do not. Importantly, however, we find that reversing the forcing in Part B from present-day (under which grounding lines show large-scale, irreversible retreat in Thwaites) to historic conditions shows that grounding lines remain close to their current position (over 10,000 years). We take this as an additional indication that no irreversible retreat has begun yet.*

*We want to stress that the basic assumptions of our study are not contradicting observational evidence. The study is designed to be able to perform stability experiments and at the same time draw conclusions about the current state of Antarctic grounding lines, as we explain in the following: To be able to apply the concept of "stable" and "unstable", we adjusted the surface mass balance for the Úa and Elmer/ice initial states in our study. With perturbation experiments (the outlet glaciers are able to recover after small-amplitude perturbations of the sub-shelf melt), we show that a stable steady state with the current geometry of the Antarctic Ice Sheet can be built (with ad-hoc adjustment of the SMB). This means that these grounding line positions are not inherently unstable.*

*Importantly, we think that these results can be applied to understand the current state of the Antarctic Ice Sheet as follows: that the current grounding line positions are not inherently unstable means that the positive feedback related to MISI is not necessarily at play for grounding lines located in their current position. Inversing the argument, this means that an observed retreat of grounding lines in their current position does not imply that there is a self-sustained component to it. Statements such as "The bed is sloping down towards the interior and the grounding line is retreating, this means that MISI has already begun / WAIS is collapsing" are wrong for the current grounding line positions. As a next step, since we find similar results for both models and also reversibility of the transient state of PISM, we think that our results rather support that the currently observed grounding line retreat has no self-reinforcing, positive feedback component to it at all.*

*Identifying the possibility for large-scale, internally driven retreat of the present-day Antarctic grounding line is of high interest as pointed out by reviewer 2. Our results drive us to the conclusion that none of the outlet glaciers are obviously engaged in an internally driven retreat.* **Of course our approach has some limitations that we hope are properly described in this new version of the manuscript (see lines 554-58 in the Discussion as examples)**. *In this version of the manuscript we have been more careful on the semantics, added more explanation of the interpretation of the results and hope to reduce confusion for the reader.*

**Technical Comments**

Presumably the calving front position was held constant during all simulations. It is unlikely that during 100 years no icebergs would calve. As Haseloff and Sergienko (2022) show, the calving front conditions have stronger effects on the grounding line stability than melting.

*The calving front position was indeed held constant, but this does not mean nothing is calved, because in order for the position to remain fixed, we are imposing a calving rate which is represented by the flux of ice at the front. When the melt perturbation is applied and the ice shelves thin, we indeed do not impose calving, but instead the ice thickness is set to a minimum value, small enough to apply only a very limited buttressing. This minimum ice thickness could have an impact on rebuilding the ice shelf, and we address this limitation in our discussion.*
*It would have been challenging, and is currently not possible in all models, to impose a calving law during the perturbation.* **We agree that this is a limitation and we have added a sentence to the discussion to reflect this, including a reference to Haseloff and Sergienko (2022) on lines 554 to 558.**

Lines 103 and 167-168 state that PISM uses 8 km horizontal resolution (no need to repeat that twice). Seroussi et al. (2014) find that the accurate representation of the grounding line dynamics requires the horizontal resolution of 2 km or higher. I appreciate that running a spin-up for 400 kyr with such a high uniform resolution is computationally expensive, but at least some comments have to be made in that regard.

**We have removed the first mention of 8km resolution.** *We agree that a higher resolution would be preferable, but as you mention, it is not possible to increase the resolution of PISM further. However, the results for all three models are consistent, despite these differences in resolution.* **We added: "Seroussi et al. (2014) report that a horizontal resolution of 2km is required to accurately represent grounding line dynamics, Feldmann et al. (2014) find that using a subgrid interpolation of friction, grounding line reversibility in PISM is also captured at coarser (x > 10km) resolution. While a higher horizontal resolution would be wishful, we here employ this interpolation to be able to run PISM over millennial time scales. We find that**

*PISM results are in line with results from Elmer/ice and Úa that employ finer resolution around the grounding lines." (lines 194 to 199)*

Lines 117-119 PICO is hardly a realistic representation of submarine melting. As studies that used ocean circulation models to compute melt rates show over and over again, melt rates do not correlate with the ice-shelf thickness or the ocean depth (e.g .Goldberg and Holland, 2022). Once again some justification is needed for the use of PICO (an alternative could have been melt rates inferred from observations, e.g. by Adusumill et al., 2020).

*We agree with the reviewer that PICO may not be the most realistic way to impose submarine melting in our models. However, conducting perturbation experiments using observations directly may not have been a good solution. Firstly, these observations are a snapshot of the current situation and we want to perturb the current situation, and therefore move away from observations a little bit. One option would have been to apply a factor or an offset to these observations, but we are not sure this would be better than a full parameterisation like the one used here. Secondly, this would have also had the issue of how to apply melt in cells that become afloat during the experiments, where observations are not present. A parameterisation of the melt does not have this issue. We want to note that ultimately the nature of the perturbation itself is not important; we could have chosen to perturb the grounding lines using a number of different parameters, and indeed we did tests perturbing the basal slipperiness field, and found the same results. In addition, all models impose slightly different spatial melt distributions; Úa and Elmer/Ice have a background "balanced melt rate field" due to the correction approach, and apply only the anomaly in PICO on top of this, whereas PISM uses the PICO melt rates directly. Despite these differences in melt, no model shows any sign of MISI driven grounding line retreat, supporting the notion that the melt rate distribution itself does not affect the results.*
***We have added "While PICO is not a perfect representation of present-day melt rates, it can track the grounding line movement and provides melting for newly ungrounded regions." to discuss this (lines 126 to 127).***
*Furthermore, we want to note that PICO includes more physics than a simple, depth-dependent parameterisation that is only based on ice draft depth / water column depth as it parameterises the vertical overturning circulation in the ice shelf cavity.*

Lines 150-165 This section about mass-balance correction is opaque. It seems that the results whether the grounding line retreats or not depend on this correction. It also appears that the obtained 'steady-state' is contrived, i.e. the configuration is essentially artificially held in this state by means of this correction. If this is not the case then this should be illustrated. With the used approach, making any connections to the actual ice-sheet state is a stretch.

*As mentioned above, stability is a property of steady states, and therefore our methodology requires a steady state. To obtain a steady state in the current geometry of the ice sheet we had to*

*apply a correction to the mass balance term. However, creating this steady state configuration in Elmer/Ice and Úa does not mean by definition that this steady state is stable. This was indeed the purpose of our experiments, to determine whether this steady state configuration of the ice sheet is stable or unstable. By excluding the effect of transient external forcing (starting from a steady state) we are able to understand if the grounding lines in their current geometry are undergoing MISI or not as outlined in our reply to the general comments above.*

*We appreciate that this balanced approach is a limitation, in the sense that we are artificially shifting the state of the hysteresis curve (see Fig.1 Part B manuscript) and varying the spatial gradients in surface mass balance. However, they are two reasons we believe this does not affect our conclusions: 1) the correction we apply to the mass balance is small (as we mention in the discussion) and therefore we have not shifted the critical thresholds in the real ice sheet, 2) PISM does not use the mass balance correction approach and is not in steady state, but also shows that the grounding lines retreat when perturbed and re-advance to the control run positions when the perturbation is removed. Thus, our results are consistent despite the individual choices made in each model.* **We have added some additional sentences to the end of Section 2.2.1 to address these issues (lines 173 to 183). On request of the editor we have also added a Figure to the supplement that shows the corrected mass balance fields of the two models. In the discussion on lines 459 to 466 we expand on this caveat.**

Lines 223-247 What is used as the surface mass balance during the perturbation experiment? Is the mass-balance correction applied as well?

*Yes, we use the corrected mass balance field throughout all (control and perturbed) simulations in Elmer/Ice and Úa, and the RACMO surface mass balance field is used in PISM.* **We have added a sentence on lines 265 to 266 to this section to clarify this:**
**"In all of our control and perturbed simulations the surface mass balance remains fixed, which in Elmer/Ice and Úa is the corrected mass balance field $m$ described in Section 2.2.1, and RACMO surface mass balance in PISM."**

Line 253 "An increase in ice shelf melt, and thus reduced buttressing, will lead to an increased ice flux. "This is an unsupported statement. It could potentially be verified by computing buttressing and demonstrating that it was reduced. If the calving front position is indeed fixed, then the retreat of the grounding line caused by increased melting, leads to the increase of the horizontal extent of the ice shelves, and hence increase in buttressing (Haseloff and Sergienko, 2018, 2022).

*Agreed, without computing the buttressing, we do not know that increased ice shelf melting has reduced the buttressing. However, our simulations clearly show that when we apply the melt perturbation, we get a sharp increase in ice flux, which is assumed to be due to reduced ice*

*thickness having a larger effect than the increased length of the ice shelf. Indeed, our profile plots show the ice shelves thin substantially during the perturbation.* **We have reworded this sentence on lines 281 to 282 to read:**
**"Increased ice shelf melting in our simulations leads to a sharp increase in ice flux across the grounding line, which is assumed to be due to a loss of buttressing as a result of ice shelf thinning."**

Lines 258-259 "When the grounding line does not retreat further, it means that it has found a new stable position very close of the previous one."As already mentioned, an alternative could be a non-linear response of the grounding line to the applied forcing (Robel et al., 2022).

*We agree with the reviewer that we cannot exclude that the GL will temporarily stop at a position for some time and then start to retreat again as suggested by Robel et al. (2022). Robel et al. (2022) suggest that "[..] the utility of 'stability' as a tool for categorizing observed glacier changes is limited without the critical context of multi-centennial (or millennial) glacier changes, [...]" They appear to decide if a grounding line "stabilizes" by running the model for 1000 years forward in time and testing if further retreat occurs during that time. We here capture almost half of that period, i.e., the centennial time scales, with the 480 year relaxation period for the entire Antarctic Ice Sheet using three different models (and therefore three different bed topographies). Since we find no indication of further retreat during this time we are relatively confident that the non-linear response mentioned by the reviewer is not at play. There are instances where the grounding lines have stopped at bed peaks when the perturbation is removed (e.g. Cook, Ronne, Thwaites), but there are also cases where they do not stop at a bed peak (e.g. Dotson). We further test the millennial timescales in PISM, as discussed in the reply to the main comment where this concern was also raised.* **We added the study to the first paragraph of our discussion on lines 444 to 448.**

Lines 262-265 and eqn(2) It is unclear what is meant by "the recovery of the ice flux" ΔQ, and how it is computed. Presumably the e− folding time could be estimated for the grounding line as well (e.g., Sergienko and Wingham, 2019).

**We have rephrased lines 291 to 293 (in the revised manuscript) to better explain the justification for calculating the e-folding time.** *We could have also calculated the e-folding time for the grounding line position, but as explained on lines 248-253 (in the original manuscript), we choose to use ice flux as our 'metric of interest'. As stated in the manuscript we chose this because the grounding line/grounded area recovery time is much longer because it first relies on the regrowth of the grounded part of the ice sheet, to recover the ice volume lost during the perturbation.*

**Summary**

The manuscript requires significant modifications in terms of the presentation and of the framing of the problem as well. Also, the main conclusion that the observed grounding line migration is driven by the atmospheric and oceanic (and lithostatic) forcing is based on observations, and not an outcome of this study. Taken at its face value, the study contradicts the observational evidence that the Antarctic Ice Sheet is not in a steady state. This is hardly a good starting point if modeling studies are to be taken seriously. As mentioned above, this study is a collaborative effort of a large group of people, undoubtedly they can find less equivocal ways to use these results.

*We don't agree that from observations alone it can be concluded that grounding line retreat is only due to external forcing. Positive feedback mechanisms (such as MISI) can lead to a non-linear, irreversible response of the system that is sustained even when the external trigger is removed. In fact that is the entire objective of this modeling paper, to clearly determine whether the present-day observed retreat of the grounding lines is only due to external atmospheric/oceanic forcing, or is it also supplemented by an internal instability (MISI) that would continue even if the external forcing was reduced. If such a process is at play cannot be concluded from observations.*

**We agree that this could be clearer in the manuscript and we have made changes to the abstract (removed the sentence on 'retreat driven by external climate forcing') and conclusion to alleviate any confusion on this point (removed the sentence 'likely that present-day observed retreat is purely driven by external forcing').**

*We completely agree that the current ice sheet is not in a steady state right now, and we hope that we never stated otherwise, even in the previous version of the manuscript. As we discussed above, we carefully designed the numerical experiments using steady states such that they allow us to make certain conclusions about the present-day state of the unsteady, real Antarctic Ice Sheet. Creating a steady state of the ice sheet in its current geometry is a prerequisite for conducting a stability analysis. In order to obtain a steady state for the models Elmer/Ice and Úa we had to apply a correction to the surface mass balance, which we describe in Section 2.2.1. However, we do not believe that our results are obtained because we used this approach. A strong indication of this is that we find no signs of irreversible retreat in our PISM simulations, for which the initial state was not generated using a correction to the surface mass balance. Also the correction fields applied to both Elmer/Ice and Úa are different from one another. Given that we have repeated our experiments with three different models and found our results are consistent across all models, we can be confident that our results are not dependent upon any particular choices made in each model.*

**We have added some additional sentences to the introduction to make it clear that: 1) we do not assume that the (real) ice sheet is in steady state, and 2) despite observational evidence that suggests present-day grounding line retreat is driven by external climate forcing, these**

*observations alone are not sufficient to conclude that MISI is not also underway, hence our steady state numerical simulations (lines 72 to 81 in the revised manuscript).*

**Author's response: Review of the manuscript "The stability of present-day Antarctic grounding lines — Part A: No indication of marine ice sheet instability in the current geometry" by Urruty et al.**

*Dear Alexander Robinson,*
*Thank you for reviewing our manuscript. Your comments are helpful and we are glad to respond to them. Please find our responses to your comments below and we will address all comments in a revised version of the manuscript. In order to facilitate the reading of this document, our responses are given in blue and italic compared to your comments which are given in* black without italic font.

This study presents a comprehensive evaluation of the possibility for large-scale, internally driven retreat of present-day Antarctic grounding lines. Two different ice sheet models were spun-up to approximate a steady-state with the present-day geometry of Antarctica, while a third model was spun-up to approximate the ice sheet after transient historical forcing since 1850. Perturbation analysis was then used to determine whether a temporal increase in basal melting over 20 years could cause the ice sheet to undergo strong grounding-line retreat that would continue after the forcing was removed. In all experiments, the present-day geometry was found to be a stable configuration, in the sense that all major grounding lines essentially returned to their original position.

This study is very interesting, timely and well done. The experiments are designed to test a specific hypothesis, and the results are convincing. Furthermore, overall the authors do a good job of discussing the various caveats to their methods and using the complementary strengths of the different models and experimental setups to confirm their findings. Particularly, I think the value of the study comes across quite well in the discussion section.

In contrast to the first reviewer, I find no major impediments to publication. I do agree that some of the framing in the Introduction and Methods could be more precise, with a few comments noted below. But I would recommend publication after only minor revisions.

*Thank you for your positive words on our work and we are glad you understood the main objectives of the study. Following the suggestions of reviewer 1 we have made improvements to the introduction and methods. **In particular we have clarified the need for a steady-state configuration of the ice sheet for our numerical model experiments (lines 72 to 81 in the revised manuscript). We have also modified the introduction to make sure there is no confusion relating to bed slope as the only criterion for grounding line stability (see response to Reviewer 1). We have additionally added some extra justification for our mass balance correction in Section 2.2.1 in the methods (lines 178 to 183 in the revised manuscript).***

**Specific comments:**

L46: Delete "In the future," as it doesn't seem to fit. Maybe instead add an "also" to become "have also shown potential"

*Sentence now removed*

L51: larger event => larger one

*Done (line 49)*

L52: marine basins => marine basins,

*Done (line 54)*

L57: "The aim of this paper is to determine if stable grounding-line positions exist in the current geometry of the ice sheet." <= Rephrase here. The current geometry has been stable for several thousand years now.

*We have rephrased the aim to: "The aim of this paper is to determine if the currently observed grounding line retreat in any Antarctic region is due to MISI, and therefore irreversible." (lines 59 to 60)*

L63: "control parameter that satisfies the steady state condition" <= It should be a control variable that satisfies the steady-state condition, right? The perturbation is applied to a parameter and the control variable (e.g., grounding-line position) is allowed to evolve. Please revise.

*Agreed, this sentence is confusing as it is currently written. We have rephrased to: "We can apply a small-amplitude perturbation to a control variable that satisfies the steady state condition, in this case the position of the grounding line. We perturb the grounding line position indirectly, by increasing the sub-shelf melt rates." (lines 63 to 65)*

L71-73: "The existence of such stable steady states is also strong indication that the currently observed retreat of Antarctic grounding lines is purely driven by changes in the external drivers such as oceanic forcing." <= It could also be an indication that the ice-sheet continues to evolve due to past climatic forcing, since as mentioned, in reality it is not in steady state. Consider adding some nuance here, which would flow better into the next paragraph which treats this point.

*We have now added that this could also be in response to past climate forcing (line 80).*

L74: ice sheet state => ice-sheet state

*Done **(line 82)***

L100: set-ups => setups

*Done **(line 108)***

L205: What is the motivation for this formula for error in grounding-line position? Add a sentence or two, as there could be many ways to define this error.

*Integrated grounded area is easy to obtain for all models rather than calculating the movement of individual grounding line segments. Grounded area is also an appropriate proxy for grounding line position change, but we wanted to convert this to a length (rather than area), and so normalised this by the grounding line length. This is also the same metric we use in our results, for grounding line position change (grounded area normalised by grounding line length). **We have adjusted the text to: "We calculate the error in the grounded ice area (a proxy for grounding line change), by differencing the simulated and observed grounded ice areas. To obtain a relative displacement of the grounding line itself, we normalise this area change by the simulated length of the grounding line." We updated the values for this error value (there was a mistake for Ua and Elmer/Ice and PISM has a new initial state) on lines 228 to 230.***

L287: 500 years => 480 years [right?]

*Done **(line 271)***

L415: "The spin-up procedure..." <= Fragment, please revise.

*Moved to the methods now and changed to "Due to the spin-up procedure…" **(line 243)***

L440: firstly => first

*Done **(line 475)***

L536: committing => producing

*Changed 'committing' to 'causing' **(line 579)***

---

## Referee Report (RR1)

**Review of a manuscript "The stability of present-day Antarctic grounding lines Part A: No indication of marine ice sheet instability in the current geometry" by Urruty et al.**

This is a revised version of an earlier submitted manuscript. I thank the authors for engaging with my comments. Some of them have been answered. However, there are several issues that still have to be resolved and clarified on both conceptual and presentation levels before the manuscript can be published. In general, the manuscript is still written in terms of "stable/unstable" ice-sheet geometry irrespective whether the climate conditions (surface accumulation and submarine melting) are changing with time or not; and the general thrust of the paper is still focused on the ice-sheet geometry with unclear and somewhat confusing description what has been done to the climate conditions and modifications to them. To this reader, there is a disconnect between what has been done in the study and how the study steps and the results have been interpreted and described. What follows below, first addresses conceptual aspects, and then the presentation aspects (in some places they're mixed together).

In very broad terms, my understanding of what has been done is the following:

1. The observed present-day geometry of the Antarctic Ice Sheet is assumed to be a steady-state one or close to a steady state. This geometry and observed velocities are used in inversion procedures in Elmer/Ice and Úa to construct basal conditions.

2. The climate conditions (the RACMO surface accumulation and PICO submarine melting)[1] are modified in such a way that when the constructed field is used in the mass-balance equation, the resulting changes of the ice thickness with time $h_t$ are fairly small or close to zero (depending on the model). Let's call this created field $\dot{A}_{ar}$ – "alternative reality" climate conditions.

3. In $\dot{A}_{ar}$, the submarine melting component is modified again by applying "small perturbations" for 20 years; after that these "small perturbations" are removed and models run for another 80 years.

4. The temporal evolution of the ice flux through the grounding line and the grounding line position simulated during these 100 years are used to establish whether the constructed steady state is stable or unstable.

If this is incorrect then clearly the manuscript does not convey what steps have been taken and it needs to be completely rewritten. Assuming that it is indeed what has been done, the text needs to modified to accurately describe these steps, the assumptions that were made at the outset, and interpretations of the results.

Among these steps, the central one is step 2 – the construction of the field $\dot{A}_{ar}$. This field is such that, if it is held constant in time, the present-day ice-sheet geometry is in (or close to) a steady state with respect to it (at this point, it does not matter whether this steady state a stable or unstable).
* * *
[1] I appreciate that there are inconsistencies between the observed geometry and ice velocities, and that RACMO and PICO are not the actual surface accumulation and submarine melting, and the need for what the authors call the "relaxation", which also contributes to the field $\dot{A}_{ar}$.

Observations show that the present-day surface elevation, ice thickness and the grounding line positions change with time, hence, the present-day geometry is not in steady state with respect to the present climate conditions. If it were, then according to the mass balance $\vec{\nabla} \cdot (uh) = \dot{A}_{pd}$, and $h_t = 0$, where $\vec{v}$ is the ice velocity, and $\dot{A}_{pd}$ is the present-day ice-sheet mass balance. Because the observed $h_t \neq 0$, the present-day conditions differ from the constructed $\dot{A}_{ar}$. Consequently, the authors need to a) articulate this point that they have constructed an "alternative reality" climate conditions; and that each model has its own "alternative reality" climate (as figs. S14-S15 indicate); b) clearly describe how $\dot{A}_{ar}$ have been constructed for each model; and preferably c) discuss how different they are from the present-day climate. Although, supplemental figures S14 and S15 show the mass-balance correction terms for Elemer/Ice and Úa, they do not address point c). These plots do not show similar fields for PISM; the colors in panels showing Úa results are oversaturated suggesting that these corrections are much larger than the colorbar limits of $\pm 2$ m/yr.

With regards to point b), the current description (lines 156-177) is not clear, especially with both RACMO, which is the surface accumulation/ablation, and PICO which is submarine melting denoted by the same variable $\dot{b}$. It would be beneficial for the manuscript to have figures showing $\dot{A}_{ar}$ for each model in the main text and either absolute or relative differences between $\dot{A}_{ar}$ and RACMO and PICO fields (essentially $h_t$ terms). Although the authors point out that the imposed mass-balance corrections have small magnitudes and are a small fraction of the total present-day mass balance (line 458), it is not only their total value, but the spatial distributions of those corrections that matters. This is because, as fig. S14 shows, these corrections are both positive and negative, and when integrated over the whole ice sheet, their contributions cancel each other. So it seems to me that the constructed $\dot{A}_{ar}$ is indeed quite "alternative reality" climate.

The description of how perturbations for PICO fields constructed for the step 3 (lines 257-275) is confusing, especially with inline formulas and too many terms having very similar notation $\dot{b}_{PICO}$. Throughout the text these perturbations are called "small", however the changes of the ocean temperature 5°C or even 1°C are hardly could be described as "small". The extra energy supplied to the ice shelves due to such an increase in the ocean temperature is $\Delta Q = C_p m \Delta T$, where $C_p$ is the sea-water heat capacity. The change in the air temperature that would correspond to this amount of extra heat $\Delta Q$ would be by about four times larger (assuming the same unit mass of air and water). This is because the air heat capacity is about four times smaller than the sea-water heat capacity. Thus, the corresponding air-temperature changes would be of the order of 4-20°C. This is well above any high-end projections of the climate warming. I appreciate that the magnitudes of perturbations have to be large enough to cause the grounding lines to move, and that they were also applied for a short time of 20 years, but still, they have to be physically reasonable. Some re-wording or clarifications for these values are needed. Perhaps, it might be better to cast these applied perturbations in terms of enhanced submarine melt-rates expressed in m/yr, rather than the ocean temperature changes. The need for such large changes in the ocean temperature suggests to me either low sensitivity of the models to changes in the submarine melting (if the changes in melt-rates that correspond to these changes in the ocean temperatures are large) or issues with PICO parameterizations. At least something needs to be said about the "smallness" of these perturbations.

It is also not clear how these perturbations have been applied. I suspect that for the ice

shelves the mass balance was

$$h_t + \vec{\nabla} \cdot (\vec{v}h) = \dot{A}_{ar} - \Delta \dot{b}_{PICO}, \tag{1}$$

and for the grounded parts it was

$$h_t + \vec{\nabla} \cdot (\vec{v}h) = \dot{A}_{ar}. \tag{2}$$

However, this is not clear from the description.

With regards to the presentation and description of the results (step 4), I am not sure that the detailed regional analysis adds value to the main results of the study which are a) there exist "alternative reality" climate conditions specific to each model, for which there are steady-state modeled geometries that are close to the present-day ice-sheet geometry, and b) these modeled steady states appear to be stable. I leave it up to the authors to decide whether to keep or remove it, but it'd be easier to read the manuscript if these parts of the text would be expressed more succinctly.

Moving on the presentational aspects, it appears that there is still a confusion between the effects of time-variant and steady-state climates on the grounding lines. The abstract starts with the sentence "Theoretical and numerical work has firmly established that grounding lines of marine-type ice sheets can enter phases of irreversible advance and retreat driven by the marine ice sheet instability (MISI). " However, theoretical and numerical work has firmly established that grounding line can equally not enter phases of irreversible advance and retreat (e.i., be stable to small perturbations). Without this second sentence, the first sentence gives an impression that the irreversibility is the only option for the grounding lines. The last two sentences "...his suggests that if the currently observed mass imbalance (external climate forcing) were to be removed, the grounding-line retreat would likely stop. However, under present-day climate forcing, further grounding-line retreat is expected, and our accompanying paper (Part B, Reese et al., 2022) shows that this could eventually lead to a collapse of some marine regions of West Antarctica." indicate that the authors view only the mass imbalance as the climate forcing, and not the temporal variability of the surface accumulation/ablation and submarine melting. Were the authors to use their constructed "alternative reality" climate conditions $\dot{A}_{ar}$ and apply temporal variations to them (even without long-term trends), they might find the grounding line behavior significantly different from the one they obtained in this study. It is unclear what is the basis of the statement that "under present-day climate forcing, further grounding-line retreat is expected". What is the reason for such an expectation?

Despite changes made to the introduction section, it continues to rely on the bed slope as an indicator of stability. The authors added a statement that "the retrograde sloping bed is a necessary conditions". This statement is incorrect. None of the stability conditions derived by Haseloff and Sergienko (2022) and Sergienko and Wingham (2019, 2022) has a necessary condition for an unstable steady state configuration to have a retrograde sloping bed. If the authors disagree, they need to support their statement with mathematical derivations demostrating that these stability conditions indeed have such a necessary condition. The authors' statement about the retrograde slopes is a widely held misconception that stems from the stability condition for a configuration with very specific conditions derived by Schoof (2012). These conditions are the absence of the lateral confinement, very smooth beds with negligible bed slopes and the smallness of the accumulation/ablation rate at the grounding lines. Only if these conditions are

simultaneously satisfied than indeed the stability condition is reduced to a condition on the sign of the bed slope (Haseloff and Sergienko, 2022; Sergienko and Wingham, 2022). However, there are no locations in Antarctica or Greenland for which all these conditions are satisfied. In addition to that, in the presence of feedbacks between the ice sheet and climate conditions there are no general stability conditions that can be related to steady-state properties (Sergienko, 2022). There is no need to promote this misconception that the retrograde beds are the necessary condition for instability; so the rest of the introduction needs to be modified accordingly.

The provided reason why the present-day Antarctic Ice Sheet is not in a steady state (lines 72-73) is incorrect as well. The ice sheet is not in a steady state not because of its response to varying climate forcing is long-term, but because the climate forcing itself varies in time, and because this climate forcing is never in steady state. In their experiments, the authors keep their constructed "alternative reality" climate conditions constant to maintain steady-state configurations. This never happens with the real climate conditions, they always vary on a wide range of temporal scales. It appears that the authors confuse steady states with equilibrium states. In the latter ones, the climate can vary with time, and if ice sheets vary in such a way that their mass gains balance their mass losses (both controlled by these time-varying climate conditions), then the ice sheets are in equilibrium with their climate. However, such an equilibrium is not a steady state, because the climate conditions vary with time. Weertman's (1974) analysis, Schoof's (2012), Haseloff and Sergienko (2022) and Sergienko and Wingham (2022) stability conditions apply to steady states only, and are not valid to equilibrium states. Currently, there is no theoretical analysis of the ice sheets which are in equilibrium with their time-varying climate conditions.

In the discussion section, comparisons with the results of other studies appear as attempts to reconcile the conclusion of this study that the grounding line is stable with conclusions of those studies that it is unstable. Because the climate conditions used in this study are vastly different from climate conditions in other studies comparison of the results is similar to comparing apples and oranges.

In summary: the manuscript needs a very clear description what has been done in the study; description of the constructed climate conditions and discussion how different they are from the present-day ones, and not only in the integral magnitude expressed in Gt/yr, but the spatial patterns as well; clarifications of the concepts of steady states and their stability that reflect the present level of knowledge; and description of the results in the context of the design of the study.

I would like to reiterate that the study has produced interesting results and it would be a pity if the manuscript describing them would misinterpret and misrepresent them.

References:
Haseloff, M. & Sergienko, O. V. (2022). Effects of calving and submarine melting on steady states and stability of buttressed marine ice sheets. Journal of Glaciology, 118. doi:10.1017/jog.2022.29

Sergienko O. V. & Wingham, D. J.(2022). Grounding line stability in a regime of low driving and basal stresses. Journal of Glaciology 65, 833849. doi:10.1017/jog.2019.53

Sergienko, O. V. & Wingham, D. J. (2022). Bed topography and marine ice-sheet stability. Journal of Glaciology 68, 124138. doi:10.1017/jog.2021.79

Sergienko, O. V. (2022). No general stability conditions for marine ice-sheet grounding lines in the presence of feedbacks. Nature Communications 13, 2265. doi:10.1038/s41467-022-29892-3

---

## Referee Report (RR2)

**Review of a manuscript "The stability of present-day Antarctic grounding lines Part A: No indication of marine ice sheet instability in the current geometry" by Hill et al.**

This manuscript had two rounds of reviews. I would like to thank the authors for their genuine efforts to address my comments. The manuscript is substantially improved. Considering its high-profile, it would be better to make it clear as possible. It appears that the authors continue to treat the ice-sheet geometry as the only characteristic of the ice sheet behaviour (*e.g.*, line 439 "As our experiments aimed to assess the stability of the observed grounding lines in their current geometry..."), yet the state of an ice sheet and its behaviour depends on the forcing in the equal measure.

My comments are of editorial nature and their purpose is to clarify remaining inconsistencies, and not to criticize the authors. The comments are not separated in "major" or "minor"; they just follow the text. By no means I want to put words in the authors' mouth. They should take the suggested text as suggestions and modify it in a way they find appropriate. On the other hand, I have no objections if they decide to use it verbatim.

**Title**: it'd be clearer if you add "... under steady-state climate conditions..." so it reads

"The stability of present-day Antarctic grounding lines *under steady-state [present-day] climate conditions* Part A: ".

My suggestion would be either to remove "marine ice-sheet instability" from the title; if your prefer to keep it, then have something like

"The stability of present-day Antarctic grounding lines Part A: No indication of marine ice sheet instability in the current geometry *under steady-state [present-day] climate conditions*"

because that's the experiments your performed. Though in your response you state that there are different versions of interpretation of marine ice-sheet instability, my understanding is that in your study you consider stability of a steady state, or something close to it, because for PISM simulations the forcing is the same - RACMO and PICO fields that do not change in time, apart from the periods when perturbations are applied and removed.

**Abstract** requires more clarifications and a brief but clear description what has been done in a study it could be something along the lines:

"Theoretical and numerical work has shown that under certain circumstances grounding lines of marine-type ice sheets can enter phases of irreversible advance and retreat – a process typically described as the marine ice sheet instability (MISI). Instances of such irreversible retreat have been found in several simulations of the Antarctic Ice Sheet under atmospheric and oceanic forcing that were meant to represent past or future climate conditions. However, it has not been assessed whether the currently observed ice-sheet geometry can maintain a steady state configuration, if the climate forcing similar to the present-day would remain constant, and whether such a steady state is stable. Here, we conduct a systematic numerical stability analysis of the present-day Antarctic Ice Sheet configuration using climate conditions that are similar to the observed ones with three state-of-the-art ice-sheet models, Úa, Elmer/Ice, and PISM. For the first two models, we construct steady-state configurations by modifying the surface accumulation simulated with RACMO and sub-ice-shelf melt rates parametrized with PICO and in such a way that if the modified fields kept constant in time, the simulated grounding lines remain at the observed present-day positions. The third model, PISM, uses spin-up procedure and historical forcing such that its transient state is close to the observed one. To assess the stability of these simulated states, we apply short-term perturbations to submarine melting that causes the grounding-line migration, which is reversed after the perturbations are removed. Our results indicate that were these climate conditions remained constant in time the grounding line positions would remain at or very close to their present-day positions. Our results suggest that the simulated grounding-line retreat under drastically different climate conditions used in previous studies is not an indication of the marine ice-sheet instability.".

The last sentence of the current version – "However, our accompanying paper (Part B, Reese et al., 2022) shows that if the grounding-lines retreat further inland, under present-day climate forcing, it may lead to the eventual irreversible collapse of some marine regions of West Antarctica."– is confusing because it contradicts the rest of the abstract. In my view, it is unnecessary, and the two parts of your study, A and B, could stand on their own.

**Introduction**:
Though the rebuttal states that "We however only use "stable/unstable" when referring to a steady state…" the first sentence of Introduction states "Retreat of the Antarctic grounding lines, i.e. the zones where the grounded ice sheet becomes so thin that it floats, could destabilise large marine regions of the ice sheet…" (emphasis is mine). Both sentences in the first paragraph need to be modified. Also references to Weertman (1974) and Schoof (2007) in the context of these sentences are incorrect – they considered idealized configurations, and had nothing (apart from a motivation) to do with Antarctic grounding lines and their retreats.

The first sentence of the second paragraph (line 18) is incorrect, because " instability" is not a mechanism, it's a property either of a steady state or a time-variant trajectory (*i.e.,* a limit cycle). Weertman (1974) and Schoof (2012) discussed it specifically as a property of a steady state, and not in "the sense of having a self-enhancing, irreversible grounding line retreat due to a positive feedback mechanism related to the ice dynamics " as you describe in the rebuttal. (It is not until line 36 that you mention the "positive feedback"sense.) The Introduction sentences up to one in line 23 (starting with "For marine, laterally uniform ice sheets with constant conditions… ") could say that the results of numerical studies with climate forcings (either varying in time or constant) that are substantially different than observed today show irreversible retreat of the grounding lines.

Line 27: the sentence "However, in the case of laterally confined ice shelves that buttress the inland grounded ice, the MISI mechanism becomes more complex." is incorrect for the same reason as above – the MISI is not a mechanism. A simple modification for it could be to replace "the MISI mechanism" with "the situation" or something similar.

Line 33: the sentensce "Critically, this means. . ." is disconnected from the sentences above it, because they describe steady-state studies, and this sentence refers to "observed retreat"– an unsteady behaviour.

Line 36: It would be helpful if you describe how this "self-reinforcing, positive feedback" operates. I suspect that it is based on the Schoof's (2007) formula of the ice flux at grounding line. If that is the case, then that refers back to the use of the MISI in the sense of stability of steady states. Also, studies referenced in lines 28-33 show that this formula is valid only for laterally unconfined marine ice sheets.

Lines 39: the sentence "For example Rosier et al. (2021). . ." is very similar to those in lines 20-23. It is better to remove/modify those and keep this one.

Lines 44-54: the second sentences of these two "However, to date there has not been a systematic analysis to determine whether the currently observed changes in grounding line position are reversible. In this paper we use a systematic modelling approach to assess whether the current retreat of the Antarctic grounding lines is due to an ongoing positive feedback mechanism related to MISI." is incorrect. This is because you do not assess whether the observed changes are reversible or not: you have created alternative, steady, states and assessed the reversibility of small perturbations from them. As you correctly point out later (lines 140-143), the present day ice sheet (and not only its geometry) is not in a steady state; and its current behaviour is not a perturbation from the steady state, or at the very least, you do not have evidence that indicates that it is a perturbation. The last sentence of this paragraph is incorrect as well for the same reason. This paragraph could state something like:

"Previous studies have argued that sections of the West Antarctic Ice Sheet may already be undergoing self-sustained unstable retreat (Joughin et al., 2014; Favier et al., 2014; Rignot et al., 2014). However, to date there has not been a systematic analysis whether, if the climate conditions close to the present-day ones kept constant, (1) the present day ice-sheet configuration could attain a steady state and (2) is stable or unstable. Our modelling approach is outlined at the beginning of Sect. 2. Briefly, we perform numerical experiments using three state-of-the-art ice sheet models, Elmer/Ice (Gagliardini et al., 2013), òa (Gudmundsson, 2020) and the Parallel Ice Sheet Model (PISM; Bueler and Brown, 2009; Winkelmann et al., 2011), by applying a perturbation to our initial model forcing, that all closely replicate the current geometry and velocity of the Antarctic Ice Sheet. Our results indicate that (1) there exist climate conditions, such that were they held constant in time, the present-day ice-sheet geometry is in (or close to) a steady state with respect to them; and (2) this steady state is stable with respect to small perturbations in the model forcing."

Line 55: the first sentence is factually incorrect for the reasons described above. This paragraph can mention the companion paper and a different approach/forcing used in it.

**Methods**
Lines 92-103: your results are still conditioned on steady states: in the case of the PISM simulations, it is forcing that is steady.

Line 158: My suggestion would be to move figs S1-S3 to the main text. Fig. S1 does not have units.

**Discussion**:
Lines 404-415: The last sentence of the first paragraph is over interpretation of the results for the reasons discussed in the comments to Introduction section. In contrast to the first sentence of the second paragraph stating that the obtained results are surprising, in my view they are not. This is because the mentioned studies use very different forcings compared to those used in your study, and also their experiments were very different from yours. Rosier et al. (2021) used forcing which is equivalent to extremely large submarine meltrates. Though you are right that conclusions of other studies that current retreat of the Pine Island Glacier is an indication of the MISI are incorrect, this is not a direct conclusion of your study.

Line 429: the first sentence of this paragraph is not correct. The use of three models does not eliminate structural uncertainties of the individual models. Because you have created steady state or initial conditions that are very specific to these models. The only common procedure is application of submarine melt perturbations. Considering that all the models use the same vertically integrated momentum balance, solve the same set of equations (more or less) and use the same perturbations, it is not surprising that they produce fairly similar behaviour. But once again, that does not imply that their structural uncertainties are eliminated.

Lines 448-452: The comparison with idealized simulations of Robel et al (2022) is not relevant for many reasons, including their experiment design and a specific choice of the bed topography.

Lines 453-470: Your assessment that the dynamic calving front would not change your results (I paraphrase) is most likely overstatement. Acknowledging that your results are valid in this particular configuration with a fixed calving front and that simulations with a moving calving front might produce different results would be better.

An additional limitation of your study that you do not consider any feedbacks between the ice-sheet geometry, *e.g.*, surface elevation, the ice-shelf draft) and the climate forcing,*e.g.*, surface accumulation and submarine melting (Sergienko, 2022; https://doi.org/10.1038/s41467-022-29892-3).

**Conclusions**: The first sentence is absolutely correct. The following sentences unfortunately not so for the reasons already mentioned above – a state of an ice sheet (its geometry and surface velocity) and its forcing cannot be considered separately. As you demonstrated, it is possible to have almost identical steady-state configurations of the Antarctic Ice Sheet with slightly different climate forcings (one for Elmer/Ice another for Úa), and a transient state, which is close to the observed one with PISM. However, your results do not tell what is the cause of the present-day grounding line migration. They tell that it is not the MISI that caused them, which is a slightly different conclusion. Lines 480-481 state "Here, we have argued that the currently observed changes of the Antarctic Ice Sheet are not a manifestation of an ongoing positive-feedback related to MISI." However, the manuscript does not have a clear description

of this positive feedback and, consequently a clear demonstration that this feedback does not operate in both your simulations and the real ice sheet. Here is a suggestion of Conclusions based on the results of your study:

"In this study, we have investigated the existence of a steady-state configuration of the Antarctic Ice Sheet which closely resembles the observed present-day configuration and the climate conditions that can maintain this steady state if kept constant in time. Our results show that it is indeed possible to construct such climate conditions for two different ice-sheet models Elmer/Ice and Úa that produce the geometry and surface velocities that are very close to those of the present-day Antarctic Ice Sheet. A transient state, which is close to the present-day ice sheet was obtained with PISM model. We find that the in all models the grounding line retreat caused by increasing sub-ice-shelf melting for a short period of time of twenty years is reversible. These results indicate that if the climate conditions would be the same as we have constructed and remain constant in time, the present-day Antarctic Ice Sheet would be in steady state, and this steady state would be stable. They also suggest that the grounding-line retreat obtained in simulations with drastically different climate conditions are not indications of the marine ice-sheet instability, and is externally forced retreat."

---

## Author Response (AR2)

Dear Benoît Urruty and co-authors,

First of all, I want to thank you and your co-authors for revising your manuscript according to the review request. Both reviewers appreciate the efforts you took in adopting their comments. Yet their general assessments diverge. Reviewer #2 requires further clarifications on the experimental setup (e.g. the SMB correction), discussion of specific setup choices, as well as an improvement in the usage of terminology with regard to steady and equilibrium states. Moreover, an explanation is requested for a possible contradiction with the partner study. In my view, these comments are constructive and will serve to make the modelling strategies more easy to follow as well as to better convey nuances in modelling choices as well as in terminology.

In summary, I continue to consider your article draft as highly interesting for publication in The Cryopshere and I therefore invite you to address these new review comments in a second revision round.

Best,

Johannes Fürst

Dear Johannes Fürst,

We want to thank you again for taking on our review process and for the handling of our manuscript up to this point.

We have made considerable efforts to revise the manuscript to address the points outlined above and to address the comments of the reviewers. This has included substantial rewriting and restructuring of some sections. Please see the response to reviewers included below.

In particular we have provided a new detailed Methods section of the manuscript in this latest version, which summarises and clarifies the experimental set-up at the beginning, and closely follows the suggestions of the reviewer. We have also added a section of the methods which clearly outlines the process to modify our mass balance field, and how these fields compare between models and to the output of regional climate models.

We have made a conscious effort to remove or amend any locations in the manuscript where there may have been confusion regarding the terminology of steady states and equilibrium.

We are unsure where you are referring to when you mention the possible contradiction with the companion paper. This is perhaps in reference to a couple of sentences at the end of the abstract as pointed out by the reviewer. We have amended these sentences for clarity.

We want to thank you and the reviewers again for their constructive comments, it has greatly improved the manuscript over the previous versions, and we appreciate the time taken to complete the reviews.

At this point we note that there has been a minor change to the order of the three first authors as it appears in the author list of the paper for this re-submission.

With best wishes,

Emily Hill, Benoît Urruty, Ronja Reese and co-authors

**Reviewer #1:**

The revised manuscript is in very good shape, and I believe it will be ready for publication after some small revisions as suggested below. The authors may also carefully read the manuscript again, as it is clear that there were several small grammatical mistakes.

Many thanks for reviewing our manuscript. Based on the comments by the second reviewer, we have made substantial changes to the manuscript and during this process we carefully checked for grammatical mistakes.

L35: a necessary conditions => a necessary condition

On suggestion from reviewer #1, the phrase "necessary condition" has been removed from the manuscript.

L59: grounding line retreat => grounding-line retreat

Done

L91: if the => as to whether the

Done

L179: As aforementioned, => We note again that

This sentence has now been removed from the manuscript.

L183: conclude on => learn about

Done

L197: would be wishful => is desirable

Done

L369: start slightly retreated of those => start in slightly retreated positions compared to those

Done

L446: 480 year => 480-year

This sentence has now been removed from the manuscript.

L487: aforementioned => mentioned earlier

We were not exactly sure where this was referring to, "aforementioned" (on previous line 455) has now been removed from the manuscript.

**Reviewer #2: Review of a manuscript "The stability of present-day Antarctic grounding lines Part A: No indication of marine ice sheet instability in the current geometry" by Urruty et al.** This is a revised version of an earlier submitted manuscript. I thank the authors for engaging with my comments. Some of them have been answered. However, there are several issues that still have to be resolved and clarified on both conceptual and presentation levels before the manuscript can be published. In general, the manuscript is still written in terms of "stable/unstable" ice-sheet geometry irrespective whether the climate conditions (surface accumulation and submarine melting) are changing with time or not; and the general thrust of the paper is still focused on the ice-sheet geometry with unclear and somewhat confusing description what has been done to the climate conditions and modifications to them. To this reader, there is a disconnect between what has been done in the study and how the study steps and the results have been interpreted and described. What follows below, first addresses conceptual aspects, and then the presentation aspects (in some places they're mixed together).

Many thanks for reviewing our manuscript, we appreciate the effort the reviewer has put into this that we think have helped us to improve the manuscript substantially. We have addressed all comments below and revised the manuscript based on them. We want to point out two more general comments:

(1) We have the impression that the reviewer thinks of "Marine Ice Sheet Instability" in a mathematically rigorous way that is only applicable to steady states and has in that sense no application to the real Antarctic ice sheet (as the reviewer mentions, this can never be in a steady state). However, importantly, the concept has in the past been used in the glaciological community in the sense of "having a self-enhancing, irreversible grounding line retreat due to a positive feedback mechanism related to the ice dynamics" (e.g., IPCC AR6 WG1 report, Pattyn & Morlighem 2020). Since this is how "MISI" is usually understood and used in publications and stakeholder dialogue, we decided to keep the terminology "MISI" to make the context of our study clear. We added a sentence to explain this in the introduction (lines 36-38). We however only use "stable/unstable" when referring to a steady state, otherwise we use "reversible/irreversible" to not propagate this any further.

(2) While the Antarctic Ice Sheet is not in a steady state as we construct some of our model configurations to be, we argue that our results can still inform about the state of the current, realistic Antarctic Ice Sheet: (a) as any numerical study, our experiments cannot perfectly reproduce the real Antarctic Ice Sheet. However, we want to stress that we here performed several experiments with a range of ice sheet models that we carefully initialise, which makes our results more robust than studies relying on only one numerical model. (b) we here show that stable, steady states can exist in the geometry of the current ice sheet, i.e., for the current grounding line positions and ice thickness distribution, with the modified surface mass balance fields. It hence means that at the current grounding line positions of the Antarctic Ice Sheet, a positive feedback mechanism causing self-sustaining retreat is not automatically at play (if it was, the steady states we construct must have been unstable). We did further experiments where the grounding lines are drifting through time, and no modification to the mass balance field was applied, and found no self-reinforcing retreat in any of these additional simulations. Taken together, our experiments indicate that in the current geometry no self-reinforcing retreat occurs (note that we do not claim this to be an implication, however, this is our best understanding until we are able to construct a state in the current geometry that shows self-reinforcing, positive grounding line retreat or "MISI" as defined in 1 and claimed in previous studies).

In very broad terms, my understanding of what has been done is the following:

1. The observed present-day geometry of the Antarctic Ice Sheet is assumed to be a steady state one or close to a steady state. This geometry and observed velocities are used in inversion procedures in Elmer/Ice and Úa to construct basal conditions.

2. The climate conditions (the RACMO surface accumulation and PICO submarine melting)[1] are modified in such a way that when the constructed field is used in the mass balance equation, the resulting changes of the ice thickness with time $h_t$ are fairly small or close to zero (depending on the model). Let's call this created field $A_{ar}$ – "alternative reality" climate conditions.

3. In $A_{ar}$, the submarine melting component is modified again by applying "small perturbations" for 20 years; after that these "small perturbations" are removed and models run for another 80 years.

4. The temporal evolution of the ice flux through the grounding line and the grounding line position simulated during these 100 years are used to establish whether the constructed steady state is stable or unstable.

If this is incorrect then clearly the manuscript does not convey what steps have been taken and it needs to be completely rewritten. Assuming that it is indeed what has been done, the text needs to modified to accurately describe these steps, the assumptions that were made at the outset, and interpretations of the results.

We thank the reviewer for clearly itemising the steps that we have carried out in the methodology of this paper. As you have written them, they are indeed correct. We appreciate that this could be better explained in the manuscript. To address this comment we have made substantial modifications to the Introduction and the Methods section. The changes that have been made are summarised below:

1. We have shortened and streamlined the Introduction to make the approach clearer.

2. At the beginning of the Methods section, we have provided a detailed explanation and justification for the methodological approaches chosen in this study. Namely, we perform two sets of perturbation experiments to 1) steady states, 2) transient states.

3. At the beginning of the Methods, we also provide a summary of how each set of experiments were performed, which in the steady state case, closely follow the bullet points 1-4 outlined in the review. This means that it is clear to the reader early on that a modification is made to the mass balance to create a steady state

4. We have substantially restructured the entire Methods section to provide a clear and logical flow and to clearly distinguish between the two types of model states for which we perform our numerical experiments. It is now split up so that "steady states" and "transient states" are presented separately.

5. As part of restructuring the Methods we have included a dedicated subsection on the modification made to the mass balance for Elmer/Ice and Úa, which both show how the corrected mass balance fields were created but also provides discussion on how these fields look with respect to present day, i.e. RACMO.

One thing to note on your description for Step 1, it is correct that the commonly adopted inversion methodology used in several ice sheet models is assuming that the geometry is in steady state at that snapshot in time. We do state in the manuscript that "Both models also
* * *
[1] I appreciate that there are inconsistencies between the observed geometry and ice velocities, and that RACMO and PICO are not the actual surface accumulation and submarine melting, and the need for what the authors call the "relaxation", which also contributes to the field $A_{ar}$.

apply an additional penalty on the rates of thickness change, to reduce nonphysical ice flux divergence anomalies" (Line 135 in the revised manuscript). However, it is important to note that using dh/dt in the inversion is designed to avoid unrealistically large flux divergences and is not a strong enough constraint to actually bring our models into steady state, because there is a notable drift in our models when we run them forward-in-time after the inversion. We have now clarified this in the Methods section, stating "…the penalty we apply to rates of thickness change, is not sufficient to bring the model into a steady-state, and when the model is run forward-in-time it diverges away from the present-day geometry and ice velocity" (Line 144 in the revised manuscript) as justification for our necessary modification to the mass balance in Step 2.

Among these steps, the central one is step 2 – the construction of the field $A_{ar}$. This field is such that, if it is held constant in time, the present-day ice-sheet geometry is in (or close to) a steady state with respect to it (at this point, it does not matter whether this steady state a stable or unstable).

This is a good explanation of what we have achieved in Step 2 of our methodology. We have now incorporated this wording into Step 2 in our summary at the beginning of our revised Methods section on lines 86-87 (in the revised manuscript).

Observations show that the present-day surface elevation, ice thickness and the grounding line positions change with time, hence, the present-day geometry is not in steady state with respect to the present climate conditions. If it were, then according to the mass balance $\nabla \cdot (vh) = A_{pd}$, and $h_t = 0$, where $v$ is the ice velocity, and $A_{pd}$ is the present-day ice-sheet mass balance. Because the observed $h_t \neq 0$, the present-day conditions differ from the constructed $A_{ar}$. Consequently, the authors need to a) articulate this point that they have constructed an "alternative reality" climate conditions; and that each model has its own "alternative reality" climate (as figs. S14-S15 indicate); b) clearly describe how $A_{ar}$ have been constructed for each model; and preferably c) discuss how different they are from the present-day climate. Although, supplemental figures S14 and S15 show the mass-balance correction terms for Elmer/Ice and Úa, they do not address point c). These plots do not show similar fields for PISM; the colors in panels showing Úa results are oversaturated suggesting that these corrections are much larger than the colorbar limits of ±2 m/yr.

We have now created a dedicated section of the Methods to present how and why we went about creating our modified mass balance field. Details are included in the list below.

1. We have used the information provided above to explain why we needed to create our modified mass balance field and that each model has a different modified mass balance field (see Section 2.2.2 Mass balance modification in the Methods)
2. We have included further explanation as to how this was done for Elmer/Ice and Úa. There is a summary in the main text and further information included in the respective appendices. There is no modification made to the mass balance field for PISM as we analyse a state in PISM that includes the present-day trend in mass losses. We have restructured the Methods to make clear the differences between Elmer/Ice / Úa and PISM in terms of the initial ice-sheet model states that are created (former two models are in steady state, PISM uses RACMO for the surface mass balance).
3. In subsection 2.2.2 we provide additional explanation and discussion of the modified mass balance fields and we have included new figures in the Supplement (Figures S1 and S2). We have also modified the colourbar limits and

plotted them on a log scale to account for the higher modification values in Úa compared with Elmer/Ice.

We prefer not to refer to this modified mass balance field as an "alternative reality" as we believe this is misleading, as we don't want the whole ice sheet model set-up to be considered an alternative reality, as the geometry is very close to the real one. Instead, we refer to it as our modified mass balance field, because this accurately describes what we have done; we have made a modification to the mass balanced field to bring the models into a steady state.

With regards to point b), the current description (lines 156-177) is not clear, especially with both RACMO, which is the surface accumulation/ablation, and PICO which is submarine melting denoted by the same variable b. It would be beneficial for the manuscript to have figures showing $A_{ar}$ for each model in the main text and either absolute or relative differences between $A_{ar}$ and RACMO and PICO fields (essentially $h_t$ terms). Although the authors point out that the imposed mass-balance corrections have small magnitudes and are a small fraction of the total present-day mass balance (line 458), it is not only their total value, but the spatial distributions of those corrections that matters. This is because, as fig. S14 shows, these corrections are both positive and negative, and when integrated over the whole ice sheet, their contributions cancel each other. So it seems to me that the constructed $A_{ar}$ is indeed quite "alternative reality" climate.

In the first round of revisions the Editor suggested we used the mass balance notation from Cogley's 'Glossary of glacier mass balance' for the climatic surface mass balance and then use RACMO/PICO as subscripts. However, now that we have removed inline formulas (see following comment below) we hope that it is not too confusing to leave this notation as it is.

We have however modified the figures to now include all three terms in Equation 1 (Figures S1 and S2), the modified mass balance fields we use, the modification applied to obtain those fields, and the present-day climate field (RACMO+PICO). We have also added discussion to the Methods as to how the modified mass balance fields compare to present-day climatic conditions.

The description of how perturbations for PICO fields constructed for the step 3 (lines 257-275) is confusing, especially with inline formulas and too many terms having very similar notation $b_{PICO}$. Throughout the text these perturbations are called "small", however the changes of the ocean temperature 5°C or even 1°C are hardly could be described as "small". The extra energy supplied to the ice shelves due to such an increase in the ocean temperature is $\Delta Q = C_p m \Delta T$, where $C_p$ is the sea-water heat capacity. The change in the air temperature that would correspond to this amount of extra heat $\Delta Q$ would be by about four times larger (assuming the same unit mass of air and water). This is because the air heat capacity is about four times smaller than the sea-water heat capacity. Thus, the corresponding air-temperature changes would be of the order of 4-20°C. This is well above any high-end projections of the climate warming. I appreciate that the magnitudes of perturbations have to be large enough to cause the grounding lines to move, and that they were also applied for a short time of 20 years, but still, they have to be physically reasonable. Some re-wording or clarifications for these values are needed. Perhaps, it might be better to cast these applied perturbations in terms of enhanced submarine melt-rates expressed in m/yr, rather than the ocean temperature changes. The need for such large changes in the ocean temperature suggests to me either low sensitivity of the models to changes in the submarine melting (if the changes in melt-rates that correspond to these

changes in the ocean temperatures are large) or issues with PICO parameterizations. At least something needs to be said about the "smallness" of these perturbations.

We have modified the perturbations section of the Methods to remove the inline formulas and updated the notation to follow equations (1) and (2) that are included below.

We do not agree that the perturbations have to be physically reasonable, and we never state so in our manuscript. The point is not how realistic the perturbation is, but as you state, just that the perturbation is sufficient to create a small deviation in the position of the grounding line. We could have chosen a number of different model parameters to perturb in order to achieve this. It is of course true that there could be some reasons in the models that a 5°C perturbation is needed to see a significant grounding line retreat, such as:

- mesh resolution at the grounding line,
- not applying adaptive remeshing at the grounding line
- melt not applied to cells crossing the grounding line
- PICO parameterization, where in the AMSE in particular it does not capture high melt at the grounding line, so higher temperatures are needed

However, we do not feel this discussion is needed in the text as it would make this section of the Methods too long. However, we have carefully been through the manuscript to make sure that all references to small perturbation are referring to a small retreat of the grounding line position and not to the forcing applied to create this perturbation. This includes revising sentences in the conclusions and abstract. We have also added a sentence to the Methods (Section 2.4) that reads "While +5 °C appears to be unrealistically high magnitude change, we want to stress that this perturbation is not designed to be realistic and is only applied over a few decades. Instead, we can think of our small perturbation, as a small movement of the grounding line away from its current position, on the order of a few grid squares or ice thicknesses at the grounding line." (Lines 242-243 in the revised manuscript).

It is also not clear how these perturbations have been applied. I suspect that for the ice shelves the mass balance was

$$h_t + \nabla \cdot (vh) = A_{ar} - \Delta b_{PICO}, \qquad (1)$$

and for the grounded parts it was

$$h_t + \nabla \cdot (vh) = A_{ar}. \qquad (2)$$

However, this is not clear from the description.

We have now made it clear in the manuscript how the perturbations were applied following the equations outlined above.

With regards to the presentation and description of the results (step 4), I am not sure that the detailed regional analysis adds value to the main results of the study which are a) there exist "alternative reality" climate conditions specific to each model, for which there are steady-state modeled geometries that are close to the present-day ice-sheet geometry, and b) these modeled steady states appear to be stable. I leave it up to the authors to decide whether to keep or remove it, but it'd be easier to read the manuscript if these parts of the text would be expressed more succinctly.

We appreciate the comments in relation to the regional analysis, but we would like to keep it in, because we believe that it is important to stress that the results we find are both

for the general Antarctic wide signal but also for any individual regions in Antarctica (including those where MISI is often discussed, e.g. Amundsen Sea). This makes it clear to the reader that there are no compensating effects in the Antarctic wide result, i.e., the behaviour of one region does not cancel out the behaviour in another. However, we have carefully been through the results sections and made efforts to make it more succinct and concise.

We have also made changes to the Discussion section to present the results of our experiments more clearly and succinctly. The key message here is that we have performed a number of experiments both on steady and unsteady states and in none of these experiments did we find any indication of self-sustained, unstable retreat. We can take this to mean that it is unlikely that there is a positive feedback mechanism (related to MISI) is at play right now in Antarctica.

Moving on the presentational aspects, it appears that there is still a confusion between the effects of time-variant and steady-state climates on the grounding lines. The abstract starts with the sentence "Theoretical and numerical work has firmly established that grounding lines of marine-type ice sheets can enter phases of irreversible advance and retreat driven by the marine ice sheet instability (MISI). " However, theoretical and numerical work has firmly established that grounding line can equally not enter phases of irreversible advance and retreat (e.i., be stable to small perturbations). Without this second sentence, the first sentence gives an impression that the irreversibility is the only option for the grounding lines. The last two sentences ". . . his suggests that if the currently observed mass imbalance (external climate forcing) were to be removed, the grounding-line retreat would likely stop. However, under present-day climate forcing, further grounding-line retreat is expected, and our accompanying paper (Part B, Reese et al., 2022) shows that this could eventually lead to a collapse of some marine regions of West Antarctica." indicate that the authors view only the mass imbalance as the climate forcing, and not the temporal variability of the surface accumulation/ablation and submarine melting. Were the authors to use their constructed "alternative reality" climate conditions $A_{ar}$ and apply temporal variations to them (even without long-term trends), they might find the grounding line behavior significantly different from the one they obtained in this study. It is unclear what is the basis of the statement that "under present-day climate forcing, further grounding-line retreat is expected". What is the reason for such an expectation?

We had hoped that the use of the word "can" in the first sentence of the abstract already suggested that grounding lines equally "cannot" enter phases of irreversible retreat, but we appreciate that this was not entirely clear. We have now modified this sentence to read "Theoretical and numerical work has shown that under certain circumstances grounding lines of marine-type ice sheets can enter phases of irreversible advance and retreat driven by the marine ice sheet instability (MISI)."

We appreciate the confusion with the statement about "observed mass imbalance" and that we have not included any assessment of the temporal variability of surface accumulation/ablation and submarine melting on the stability of the grounding lines. For simplicity we have simply removed the second to last sentence from the abstract as it was not necessary. We have also modified the final sentence to make a clearer statement about our accompanying paper and to remove the word expected which we agree was unfounded. It now reads as "However, our accompanying paper (Part B, Reese et al., 2022) shows that if the grounding-lines retreat further inland, under present-day climate forcing, it may lead to the eventual collapse of some marine regions of West Antarctica."

Despite changes made to the introduction section, it continues to rely on the bed slope as an indicator of stability. The authors added a statement that "the retrograde sloping bed is a necessary conditions". This statement is incorrect. None of the stability conditions derived by Haseloff and Sergienko (2022) and Sergienko and Wingham (2019, 2022) has a necessary condition for an unstable steady state configuration to have a retrograde sloping bed. If the authors disagree, they need to support their statement with mathematical derivations demostrating that these stability conditions indeed have such a necessary condition. The authors' statement about the retrograde slopes is a widely held misconception that stems from the stability condition for a configuration with very specific conditions derived by Schoof (2012). These conditions are the absence of the lateral confinement, very smooth beds with negligible bed slopes and the smallness of the accumulation/ablation rate at the grounding lines. Only if these conditions are simultaneously satisfied than indeed the stability condition is reduced to a condition on the sign of the bed slope (Haseloff and Sergienko, 2022; Sergienko and Wingham, 2022). However, there are no locations in Antarctica or Greenland for which all these conditions are satisfied. In addition to that, in the presence of feedbacks between the ice sheet and climate conditions there are no general stability conditions that can be related to steady-state properties (Sergienko, 2022). There is no need to promote this misconception that the retrograde beds are the necessary condition for instability; so the rest of the introduction needs to be modified accordingly.

We are grateful for your explanation of the relevant studies and for the clarification on our statements about the retrograde bed slope as a necessary condition. We of course do not want to perpetrate the misconception that bed slope is a necessary condition for instability. We have now revised the introduction substantially compared to the previous version and we no longer include the paragraphs that refer to marine basins of the ice sheet and there is no longer any mention that the retrograde bed slope is a necessary condition for MISI to occur.

The provided reason why the present-day Antarctic Ice Sheet is not in a steady state (lines 72-73) is incorrect as well. The ice sheet is not in a steady state not because of its response to varying climate forcing is long-term, but because the climate forcing itself varies in time, and because this climate forcing is never in steady state. In their experiments, the authors keep their constructed "alternative reality" climate conditions constant to maintain steady-state configurations. This never happens with the real climate conditions, they always vary on a wide range of temporal scales. It appears that the authors confuse steady states with equilibrium states. In the latter ones, the climate can vary with time, and if ice sheets vary in such a way that their mass gains balance their mass losses (both controlled by these time-varying climate conditions), then the ice sheets are in equilibrium with their climate. However, such an equilibrium is not a steady state, because the climate conditions vary with time. Weertman's (1974) analysis, Schoof's (2012), Haseloff and Sergienko (2022) and Sergienko and Wingham (2022) stability conditions apply to steady states only, and are not valid to equilibrium states. Currently, there is no theoretical analysis of the ice sheets which are in equilibrium with their time-varying climate conditions.

We have made substantial modifications to the introduction in this latest version of the manuscript such that the sentence that was previously on lines 72-73 has now been removed. We have also made sure that there are no longer any confusing statements with regards to steady-state and equilibrium. We refer to steady state for the initial states created in Elmer/Ice and Úa, where dh/dt=0 (or numerically close). We only use the term equilibrium to describe the initialisation of PISM and define this as the integrated ice sheet mass balance is close to zero.

In the discussion section, comparisons with the results of other studies appear as attempts to reconcile the conclusion of this study that the grounding line is stable with conclusions of those studies that it is unstable. Because the climate conditions used in this study are vastly different from climate conditions in other studies comparison of the results is similar to comparing apples and oranges.

We appreciate the concerns about making comparisons between studies that use different approaches to draw their conclusions. It is true of course that the climate conditions used in this study may be different to others. A few thoughts on this: 1) we refer to papers that have "suggested" that MISI might be underway, using observations of grounding line retreat and bed topography alone, and do not consider the changing climate. 2) we compare our results to papers that have run modelling simulations, but importantly some of these studies also keep climate conditions constant through time, and secondly, do not perform a numerical stability analysis, i.e., applying and reversing a small amplitude perturbation. Therefore, while the climate conditions are different, those studies were not able to conclude on the stability of the current grounding lines. Here, we have shown that we can't assume that the current grounding lines are retreating due to a positive feedback mechanism related to MISI, and that (albeit using our modified mass balance) we can find stable grounding line positions in the current geometry of the ice sheet. Given that this is the key conclusion of our work, we feel these comparisons to previous studies are necessary. However, we have looked back through the Discussion and made modifications to the text to make the comparisons to previous studies clearer.

- We have removed the sentence "This is also supported by observations that show grounding-line retreat to have recently stagnated, suggesting the current position is indeed stable (Konrad et al., 2018)." As it was not necessary.
- We have made the sentence in which we compare the results of the companion paper to future modelling experiments at Thwaites Glacier clearer by stating that they applied increases in ocean forcing, but kept present-day surface mass balance conditions fixed through time. This sentence now reads "Several modelling studies have also shown that once the grounding lines retreat further inland under future increases in ocean forcing (and in the absence of any increases in surface accumulation/ablation from present day), it is possible that they will enter phrases of accelerated retreat (Joughin et al., 2014; Seroussi et al., 2017)"

A somewhat side note is that it is our impression that it is rare that any two modelling studies (with the exception of model intercomparison exercises) use the exact same climate conditions, and therefore comparing results between studies would become near on impossible. In our opinion, the comparison to previous studies both highlights similarities and puts our results in a wider scientific context, with the purpose of advancing scientific knowledge.

In summary: the manuscript needs a very clear description what has been done in the study; description of the constructed climate conditions and discussion how different they are from the present-day ones, and not only in the integral magnitude expressed in Gt/yr, but the spatial patterns as well; clarifications of the concepts of steady states and their stability that reflect the present level of knowledge; and description of the results in the context of the design of the study.

Following the suggestions of the reviewer, in summary, in this latest version of the manuscript we have:

- Provided a clear description of the methodology and what has been done at the beginning of the methods.
- Included a dedicated section of the methods to how the mass balance fields ("constructed climate conditions") were created for each model and how they differ from one another and from the present-day climate conditions. We have focused on the spatial patterns, have provided new figures and removed the focus on the integral magnitude.
- We have removed any misconceptions around steady states and equilibrium states from the manuscript.
- We have left the regional analysis of our results as they were, but we have reframed the key messages/results presented in the abstract, discussion, and conclusions section to better reflect our key results in the context of the design of the study.

I would like to reiterate that the study has produced interesting results and it would be a pity if the manuscript describing them would misinterpret and misrepresent them.

We thank the reviewer for their interest in our work and for reviewing the manuscript. The reviewer's comments have greatly helped to improve the clarity of the manuscript over the previous versions.

References:

Haseloff, M. & Sergienko, O. V. (2022). Effects of calving and submarine melting on steady states and stability of buttressed marine ice sheets. Journal of Glaciology, 118. doi:10.1017/jog.2022.29

Sergienko O. V. & Wingham, D. J.(2022). Grounding line stability in a regime of low driving and basal stresses. Journal of Glaciology 65, 833849. doi:10.1017/jog.2019.53

Sergienko, O. V. & Wingham, D. J. (2022). Bed topography and marine ice-sheet stability. Journal of Glaciology 68, 124138. doi:10.1017/jog.2021.79 4

Sergienko, O. V. (2022). No general stability conditions for marine ice-sheet grounding lines in the presence of feedbacks. Nature Communications 13, 2265. doi:10.1038/s41467-022-29892-3

---

## Author Response (AR3)

Dear Emily Hill and co-authors,

I am most delighted to inform you that the pending review report is in and that the reviewer very much appreciates your 'genuine efforts to address [the] comments'. At this stage, the reviewer mostly asks for editorial changes. Comments appear constructive and many of them come with pre-formulated suggestions for text modifications.

In light of this positive report, I invite you to address these comments during this step of minor revisions. I expect a fast iteration.

Yours Sincerely,

Johannes Fürst

Dear Johannes Fürst,

Thank you for inviting us to address these minor revisions. We have made a conscious and thorough effort to implement all of these reviewer comments, including taking on board pre-formulated text modifications.

Please find included below the reviewer comments in black and our response in blue.

With best wishes,
Emily Hill, Benoît Urruty, Ronja Reese and co-authors

**Review of a manuscript "The stability of present-day Antarctic grounding lines Part A: No indication of marine ice sheet instability in the current geometry" by Hill et al.**

This manuscript had two rounds of reviews. I would like to thank the authors for their genuine efforts to address my comments. The manuscript is substantially improved. Considering its high-profile, it would be better to make it clear as possible. It appears that the authors continue to treat the ice-sheet geometry as the only characteristic of the ice sheet behaviour (e.g., line 439 "As our experiments aimed to assess the stability of the observed grounding lines in their current geometry. . . "), yet the state of an ice sheet and its behaviour depends on the forcing in the equal measure. My comments are of editorial nature and their purpose is to clarify remaining inconsistencies, and not to criticize the authors. The comments are not separated in "major" or "minor"; they just follow the text. By no means I want to put words in the authors' mouth. They should take the suggested text as suggestions and modify it in a way they find appropriate. On the other hand, I have no objections if they decide to use it verbatim.

Thanks for the additional review comments. We have made a conscious effort to address all of your comments. See our responses and changes that have been made in the manuscript below.

**Title:** it'd be clearer if you add "... under steady-state climate conditions..." so it reads

"The stability of present-day Antarctic grounding lines under steady-state [present-day] climate conditions Part A: No indication of marine ice sheet instability in the current geometry".

My suggestion would be either to remove "marine ice-sheet instability" from the title; if your prefer to keep it, then have something like

"The stability of present-day Antarctic grounding lines Part A: No indication of marine ice sheet instability in the current geometry under steady-state [present-day] climate conditions"

because that's the experiments your performed. Though in your response you state that there are different versions of interpretation of marine ice-sheet instability, my understanding is that in your study you consider stability of a steady state, or something close to it, because for PISM simulations the forcing is the same - RACMO and PICO fields that do not change in time, apart from the periods when perturbations are applied and removed.

It is true that in all our experiments the climate conditions are steady, i.e. fixed through time, but our PISM ice sheet state is not in a steady state, which is a key difference between this and the other models. In fact, Part B shows that the steady state the PISM configuration evolves toward under steady climate conditions has a substantially retreated grounding line in WAIS. Also the climate conditions used between the three models are also different so we feel that adding this to the title could cause confusion to the reader. Also the title already spans two lines, and adding more words makes it incomprehensible.

We do however acknowledge that having a steady climate means that we do not consider so-called noise-induced tipping that the reviewer appears to indirectly refer to in their comments on the steady versus variable climate conditions (here and below). We think that adding this directly as a

caveat to the discussion is easier for the reader to understand than indirectly implying it by adding "steady climate conditions" in the title and other places. We added the following text "Another point to make is that in our study here we consider only steady climate conditions in our experiments. Studies have shown that variability in the climate can cause noise-induced tipping which causes a system to transgress towards a qualitatively different state before the actual tipping point is crossed (e.g., Ashwin et al., 2012). Future work would benefit from incorporating time varying climate conditions to explore the possibility of noise-induced tipping in the real Antarctic Ice Sheet."

**Abstract** requires more clarifications and a brief but clear description what has been done in a study it could be something along the lines:

"Theoretical and numerical work has shown that under certain circumstances grounding lines of marine-type ice sheets can enter phases of irreversible advance and retreat – a process typically described as the marine ice sheet instability (MISI). Instances of such irreversible retreat have been found in several simulations of the Antarctic Ice Sheet under atmospheric and oceanic forcing that were meant to represent past or future climate conditions. However, it has not been assessed whether the currently observed ice-sheet geometry can maintain a steady state configuration, if the climate forcing similar to the present-day would remain constant, and whether such a steady state is stable. Here, we conduct a systematic numerical stability analysis of the present-day Antarctic Ice Sheet configuration using climate conditions that are similar to the observed ones with three state-of-the-art ice-sheet models, Ua, Elmer/Ice, and PISM. For the first two models, we construct steady-state configurations by modifying the surface ac- cumulation simulated with RACMO and sub-ice-shelf melt rates parametrized with PICO and in such a way that if the modified fields kept constant in time, the simulated grounding lines remain at the observed present-day positions. The third model, PISM, uses spin-up procedure and historical forcing such that its transient state is close to the observed one. To assess the stability of these simulated states, we apply short-term perturbations to submarine melting that causes the grounding-line migration, which is reversed after the perturbations are removed. Our results indicate that were these climate conditions remained constant in time the grounding line positions would remain at or very close to their present-day positions. Our results suggest that the simulated grounding-line retreat under drastically different climate conditions used in previous studies is not an indication of the marine ice-sheet instability."

We have not taken this new abstract verbatim but agree that the model set-up and initial states used could have been explained more clearly. We have added some sentences to that effect and the new abstract is included below. We did not want to include the second to last sentence of your proposed rewrite of the Abstract because stating "if climate conditions remained constant in time..positions would remain close to present-day positions" (paraphrased), because this contradicts the findings of our Part B experiments, which clearly show (as mentioned above), that under steady (present-day) climate conditions, the ice sheet eventually evolves towards a 'retreated' state. We are not sure what is meant by 'drastically different climate conditions', and because this is a qualitative description we feel that it would be confusing to the reader, especially given that we do not make comparisons to climate conditions used in other studies elsewhere in the manuscript.

The last sentence of the current version – "However, our accompanying paper (Part B, Reese et al., 2022) shows that if the grounding-lines retreat further inland, under present-day climate forcing, it may lead to the eventual irreversible collapse of some marine regions of West Antarctica."– is

confusing because it contradicts the rest of the abstract. In my view, it is unnecessary, and the two parts of your study, A and B, could stand on their own.

Perhaps it is true that they could stand alone, but at present they are referenced to one another with Part A and B and so it seems appropriate to make a reference to Part B in the abstract. This statement is also not intended to be contradictory to the rest of the abstract. It would however, contradict this sentence suggested by the reviewer "Our results indicate that were these climate conditions remained constant in time the grounding line positions would remain at or very close to their present-day positions". As we discuss in the reply above, this statement is not what our experiments are showing and we do not claim this as a conclusion anywhere in the manuscript either. We have made an effort to reformulate the sentence about our Part B results to make this clearer to the reader. See our revised abstract below.

"Theoretical and numerical work has shown that under certain circumstances grounding lines of marine-type ice sheets can enter phases of irreversible advance and retreat driven by the marine ice sheet instability (MISI). Instances of such irreversible retreat have been found in several simulations of the Antarctic Ice Sheet. However, it has not been assessed whether the Antarctic grounding lines are already undergoing MISI in their current position. Here, we conduct a systematic numerical stability analysis using three state-of-the-art ice-sheet models, Ua, Elmer/Ice, and PISM. For the first two models, we construct steady-state initial configurations whereby the simulated grounding lines remain at the observed present-day positions through time. The third model, PISM, uses a spin-up procedure and historical forcing such that its transient state is close to the observed one. To assess the stability of these simulated states, we apply short-term perturbations to submarine melting. Our results show that the grounding lines around Antarctica migrate slightly away from their initial position while the perturbation is applied, and then revert once the perturbation is removed. This indicates that present-day retreat of Antarctic grounding lines is not yet irreversible or self-sustained. However, our accompanying paper (Part B, Reese et al., 2022) shows that if the grounding-lines retreat further inland, under present-day climate forcing, it may lead to the eventual irreversible collapse of some marine regions of West Antarctica."

**Introduction:**

Though the rebuttal states that "We however only use "stable/unstable" when referring to a steady state. . . " the first sentence of Introduction states "Retreat of the Antarctic ground- ing lines, i.e. the zones where the grounded ice sheet becomes so thin that it floats, could destabilise large marine regions of the ice sheet. . . " (emphasis is mine). Both sentences in the first paragraph need to be modified. Also references to Weertman (1974) and Schoof (2007) in the context of these sentences are incorrect – they considered idealized configurations, and had nothing (apart from a motivation) to do with Antarctic grounding lines and their retreats.

The first paragraph of the introduction has been revised. References to Weertman and Schoof have been removed.

The first sentence of the second paragraph (line 18) is incorrect, because " instability" is not a mechanism, it's a property either of a steady state or a time-variant trajectory (i.e., a limit cycle).

Weertman (1974) and Schoof (2012) discussed it specifically as a property of a steady state, and not in "the sense of having a self-enhancing, irreversible grounding line retreat due to a positive feedback mechanism related to the ice dynamics " as you describe in the rebuttal. (It is not until line 36 that you mention the "positive feedback"sense.) The Introduction sentences up to one in line 23 (starting with "For marine, laterally uniform ice sheets with constant conditions. . . ") could say that the results of numerical studies with climate forcings (either varying in time or constant) that are substantially different than observed today show irreversible retreat of the grounding lines.

We agree with the reviewer that technically MISI is not a mechanism, but a property of a steady state or a time-variant trajectory. However, and importantly, across the literature it is referred to as a mechanism and we want to stay consistent with this. To align with this, and to address your comment further down about 'describing how this self-reinforcing positive-feedback' operates, we have moved the sentences from Line 36 up to the start of this second paragraph in the introduction. This now makes it clear from the outset what we are referring to when we say "positive self-reinforcing mechanism"

We have also changed the second sentence of the second paragraph to "The existence of MISI means that a shift in the position of the grounding line can cause it to cross a critical threshold (or `tipping point'), beyond which the system is driven towards a different steady state"

Line 27: the sentence "However, in the case of laterally confined ice shelves that buttress the inland grounded ice, the MISI mechanism becomes more complex." is incorrect for the same reason as above – the MISI is not a mechanism. A simple modification for it could be to replace "the MISI mechanism" with "the situation" or something similar.

We changed the sentence to now refer to 'feedback mechanism' instead of MISI.

Line 33: the sentence "Critically, this means. . . "is disconnected from the sentences above it, because they describe steady-state studies, and this sentence refers to "observed retreat"– an unsteady behaviour.

For simplicity, we have removed these two sentences.

Line 36: It would be helpful if you describe how this "self-reinforcing, positive feedback" operates. I suspect that it is based on the Schoof's (2007) formula of the ice flux at grounding line. If that is the case, then that refers back to the use of the MISI in the sense of stability of steady states. Also, studies referenced in lines 28-33 show that this formula is valid only for laterally unconfined marine ice sheets.

See comment above, we have moved these sentences earlier in the introduction to make it clear what we mean by "self-reinforcing, positive feedback". It is also discussed on line 25 in the previous version of the manuscript. In our understanding, this "self-reinforcing, positive feedback" is exactly how Schoof 2007 links to the "transient" MISI understanding of today.

Lines 39: the sentence "For example Rosier et al. (2021). . . " is very similar to those in lines 20-23. It is better to remove/modify those and keep this one.

Sentences from lines 20-23 have now been incorporated at line 39.

Lines 44-54: the second sentences of these two "However, to date there has not been a systematic analysis to determine whether the currently observed changes in grounding line position are reversible. In this paper we use a systematic modelling approach to assess whether the current retreat of the Antarctic grounding lines is due to an ongoing positive feedback mechanism related to MISI." is incorrect. This is because you do not assess whether the observed changes are reversible or not: you have created alternative, steady, states and assessed the reversibility of small perturbations from them. As you correctly point out later (lines 140-143), the present day ice sheet (and not only its geometry) is not in a steady state; and its current behaviour is not a perturbation from the steady state, or at the very least, you do not have evidence that indicates that it is a perturbation. The last sentence of this paragraph is incorrect as well for the same reason. This paragraph could state something like:

"Previous studies have argued that sections of the West Antarctic Ice Sheet may already be undergoing self-sustained unstable retreat (Joughin et al., 2014; Favier et al., 2014; Rignot et al., 2014). However, to date there has not been a systematic analysis whether, if the climate conditions close to the present-day ones kept constant, (1) the present day ice-sheet configu- ration could attain a steady state and (2) is stable or unstable. Our modelling approach is outlined at the beginning of Sect. 2. Briefly, we perform numerical experiments using three state-of-the-art ice sheet models, Elmer/Ice (Gagliardini et al., 2013), o`a (Gudmundsson, 2020) and the Parallel Ice Sheet Model (PISM; Bueler and Brown, 2009; Winkelmann et al., 2011), by applying a perturbation to our initial model forcing, that all closely replicate the current geometry and velocity of the Antarctic Ice Sheet. Our results indicate that (1) there exist climate conditions, such that were they held constant in time, the present-day ice-sheet geometry is in (or close to) a steady state with respect to them; and (2) this steady state is stable with respect to small perturbations in the model forcing."

We appreciate the suggested change to the text. Crucially though, changing the text in this manner would lead to inconsistencies in the manuscript. The detailed explanation of our methodology at the beginning of Section 2 would no longer make sense if we were to reformulate the paragraph in this way. This would require an entire reformulation of our approach throughout the entire manuscript.

However, you are right that we cannot assess whether the observed changes are reversible or not, but we can only make inferences about how the present-day retreat of the grounding lines may not be due to MISI. To that effect we have rephrased the section of the paragraph to remove the sentence "In this paper we use a systematic modelling approach to assess whether the current retreat of the Antarctic grounding lines is due to an ongoing positive feedback mechanism related to MISI". The new paragraph now reads as below:

"Previous studies have suggested that present-day retreat in regions of the West Antarctic Ice Sheet could mean that irreversible retreat has begun (Joughin et al., 2014; Rignot et al., 2014). However, to date there has not yet been a systematic analysis to assess whether irreversible retreat of Antarctic grounding lines is already underway. In this paper we use a systematic modelling approach to assess whether, under steady climate conditions, the grounding line positions of the Antarctic Ice Sheet are reversible with respect to a small-amplitude perturbation away from their current positions. Our

modelling approach is outlined in detail at the beginning of Sect. 2. Briefly, we perform numerical experiments using three state-of-the-art ice sheet models, Elmer/Ice (Gagliardini et al., 2013), Ua (Gudmundsson, 2020) and the Parallel Ice Sheet Model (PISM; Bueler and Brown, 2009; Winkelmann et al., 2011) by applying a small but numerically significant perturbation to our initial model states, that all closely replicate the current geometry and velocity of the Antarctic Ice Sheet. If we find the grounding lines to either revert back to their former position (if the state is steady), or stay within the vicinity (if drifting through time), then this would indicate that current retreat of Antarctic grounding lines is unlikely to be due to an ongoing positive feedback mechanism, i.e. related to MISI."

Line 55: the first sentence is factually incorrect for the reasons described above. This paragraph can mention the companion paper and a different approach/forcing used in it.

The first sentence has been rephrased to "A follow-on question to the stability of the Antarctic grounding lines is: could the currently observed retreat driven by present-day climate conditions, eventually commit the grounding lines to undergo irreversible retreat?"

**Methods**

Lines 92-103: your results are still conditioned on steady states: in the case of the PISM simulations, it is forcing that is steady.

We agree that the climate forcing in PISM is steady, but the ice sheet state is not. We feel that this is a key distinction between the experiments conducted with Elmer/Ice and Ua and the experiments conducted with PISM. To make this clear throughout the manuscript we have stated where the climate forcing is steady but the ice sheet state is not. For example in Section 2.3.2 (describing the initial state in PISM) we state "During the control run the present-day climate conditions (2015) are held constant and so the climate forcing itself is steady, but the ice sheet state itself is not in a steady-state (ice thickness changes through time are not equal to zero)."

Line 158: My suggestion would be to move figs S1-S3 to the main text. Fig. S1 does not have units.

We do not think adding an extra three figures into the main text is justified, it will make it too long, and detract from the main results of the paper. The figures are the first to appear in the supplementary document, which is easy to access for all readers. Units have been added to Fig. S1.

**Discussion:**

Lines 404-415: The last sentence of the first paragraph is over interpretation of the results for the reasons discussed in the comments to Introduction section. In contrast to the first sentence of the second paragraph stating that the obtained results are surprising, in my view they are not. This is because the mentioned studies use very different forcings compared to those used in your study, and also their experiments were very different from yours. Rosier et al. (2021) used forcing which is equivalent to extremely large submarine meltrates. Though you are right that conclusions of other studies that current retreat of the Pine Island Glacier is an indication of the MISI are incorrect, this is not a direct conclusion of your study.

For simplicity we have removed the last sentence of the first paragraph of the Discussion.

The second paragraph has been reformulated to remove the statement "it is perhaps surprising" and the sentence "...no indication of ongoing retreat in this sector of the ice sheet" as we agree this cannot be a direct conclusion of our study, but only something we can infer from our experiments. It now reads "Our results instead suggest that the ASE sector of West Antarctica has reversible grounding-lines in response to a small deviation from their current position".

Line 429: the first sentence of this paragraph is not correct. The use of three models does not eliminate structural uncertainties of the individual models. Because you have created steady state or initial conditions that are very specific to these models. The only common procedure is application of submarine melt perturbations. Considering that all the models use the same vertically integrated momentum balance, solve the same set of equations (more or less) and use the same perturbations, it is not surprising that they produce fairly similar behaviour.  But once again, that does not imply that their structural uncertainties are eliminated.

Changed the first sentence to "While our experiments, performed with three different ice sheet models, have shown consistent results, in the following paragraphs we note several caveats and potential sources of uncertainty."

Lines 448-452: The comparison with idealized simulations of Robel et al (2022) is not relevant for many reasons, including their experiment design and a specific choice of the bed topography.

It was a suggestion in the previous rounds of review to add this reference. We are happy to remove it and have done so.

Lines 453-470: Your assessment that the dynamic calving front would not change your results (I paraphrase) is most likely overstatement. Acknowledging that your results are valid in this particular configuration with a fixed calving front and that simulations with a moving calving front might produce different results would be better.

We have changed one of the sentences to read "We acknowledge that an evolving calving front may produce different results, especially as recent work has shown that (in the presence of buttressing) iceberg calving could impact the stability regime of grounding lines (Haseloff and Sergienko, 2022)"

An additional limitation of your study that you do not consider any feedbacks between the ice- sheet geometry, e.g., surface elevation, the ice-shelf draft) and the climate forcing,e.g., surface accumulation and submarine melting (Sergienko, 2022; https://doi.org/10.1038/s41467-022-29892-3).

Added a sentence to the end of the discussion stating we do not account for the surface melt elevation feedback. We include the negative feedback related to melting causing the ice shelf to thin, which increases the in-situ pressure melting point and thereby reduces melting as this is covered in the PICO melt calculation. However, we do not consider feedbacks on ocean circulation that would require a coupled ice-ocean model configuration.

**Conclusions:** The first sentence is absolutely correct. The following sentences unfortunately not so for the reasons already mentioned above – a state of an ice sheet (its geometry and surface velocity) and its forcing cannot be considered separately. As you demonstrated, it is possible to have almost identical steady-state configurations of the Antarctic Ice Sheet with slightly different climate forcings (one for Elmer/Ice and Ua), and a transient state, which is close to the observed one with PISM. However, your results do not tell what is the cause of the present-day grounding line migration. They tell that it is not the MISI that caused them, which is a slightly different conclusion. Lines 480-481 state "Here, we have argued that the currently observed changes of the Antarctic Ice Sheet are not a manifestation of an ongoing positive-feedback related to MISI." However, the manuscript does not have a clear description of this positive feedback and, consequently a clear demonstration that this feedback does not operate in both your simulations and the real ice sheet. Here is a suggestion of Conclusions based on the results of your study:

"In this study, we have investigated the existence of a steady-state configuration of the Antarctic Ice Sheet which closely resembles the observed present-day configuration and the climate conditions that can maintain this steady state if kept constant in time. Our results show that it is indeed possible to construct such climate conditions for two different ice-sheet models Elmer/Ice and Ua that produce the geometry and surface velocities that are very close to those of the present-day Antarctic Ice Sheet. A transient state, which is close to the present- day ice sheet was obtained with PISM model.  We find that the in all models the grounding line retreat caused by increasing sub-ice-shelf melting for a short period of time of twenty years is reversible. These results indicate that if the climate conditions would be the same as we have constructed and remain constant in time, the present-day Antarctic Ice Sheet would be in steady state, and this steady state would be stable. They also suggest that the grounding-line retreat obtained in simulations with drastically different climate conditions are not indications of the marine ice-sheet instability, and is externally forced retreat."

We have modified the second sentences of the conclusions to remove "(its geometry and surface velocity)". With regards to your second point, nowhere in the conclusions do we state what the cause of present-day grounding line migration is (in the previous versions of the manuscript we said "externally forced" but this is no longer in there now). Instead we have the sentence "we have argued that the currently observed changes of the Antarctic Ice Sheet are not a manifestation of an ongoing positive-feedback related to MISI", which is exactly as you state "...that MISI has not caused them". So we are not sure where this refers to. We have made the description of the positive feedback clearer above and our reversibility experiments show that it does not operate. We have revised the conclusions. See below for the new version:

"Our key finding is that the grounding lines of Antarctica, including Pine Island and Thwaites glaciers, show no indication of marine ice sheet instability (MISI) in their current state. We arrive at this conclusion by showing the existence of stable steady-state model configurations of the Antarctic Ice Sheet which closely resemble the observed present-day configuration as well as by showing reversibility of transient model configurations of the Antarctic Ice Sheet, under steady climate conditions. Our results indicate that MISI is not causing the present-day grounding line migration.

There is a general consensus within the ice-sheet modelling community that the West Antarctic Ice Sheet is susceptible to MISI. Here, we have argued that the currently observed changes of the Antarctic Ice Sheet are not a manifestation of an ongoing positive-feedback related to MISI. While

our experiments suggest that an internal instability threshold has not yet been crossed in Antarctica, future retreat driven by changes in climate conditions could force the grounding lines to cross a tipping point, after which retreat becomes driven by MISI. Further work is needed to quantify the amplitude and duration of forcing required for the Antarctic Ice Sheet to enter a phase of a large-scale irreversible collapse involving grounding-line retreat over hundreds of kilometers and a concomitant sea-level rise of potentially several meters."